# The Unreasonable Effectiveness of Gaussian Score Approximation for Diffusion Models and its Applications

**Binxu Wang**                                                    *binxu_wang@hms.harvard.edu*
*Kempner Institute for the Study of Natural and Artificial Intelligence*
*Harvard University*
*Boston, MA 02134, USA*

**John J. Vastola**                                               *john_vastola@hms.harvard.edu*
*Department of Neurobiology*
*Harvard Medical School*
*Boston, MA 02115, USA*

**Reviewed on OpenReview:** *https://openreview.net/forum?id=IOuknSHM2j*

## Abstract

Diffusion models have achieved remarkable results in multiple domains of generative modeling. By learning the gradient of smoothed data distributions, they can iteratively generate samples from complex distributions, e.g., of natural images. The learned score function enables their generalization capabilities, but how the learned score relates to the score of the underlying data manifold remains largely unclear. Here, we aim to elucidate this relationship by comparing the learned scores of neural-network-based models to the scores of two kinds of analytically tractable distributions: Gaussians and Gaussian mixtures. The simplicity of the Gaussian model makes it particularly attractive from a theoretical point of view, and we show that it admits a closed-form solution and predicts many qualitative aspects of sample generation dynamics. We claim that the learned neural score is dominated by its linear (Gaussian) approximation for moderate to high noise scales, and supply both theoretical and empirical arguments to support this claim. Moreover, the Gaussian approximation empirically works for a larger range of noise scales than naive theory suggests it should, and is preferentially learned by networks early in training. At smaller noise scales, we observe that learned scores are better described by a coarse-grained (Gaussian mixture) approximation of training data than by the score of the training distribution, a finding consistent with generalization. Our findings enable us to precisely predict the initial phase of trained models' sampling trajectories through their Gaussian approximations. We show that this allows one to leverage the Gaussian analytical solution to skip the first 15-30% of sampling steps while maintaining high sample quality (with a near state-of-the-art FID score of 1.93 on CIFAR-10 unconditional generation). This forms the foundation of a novel hybrid sampling method, termed *analytical teleportation*, which can seamlessly integrate with and accelerate existing samplers, including DPM-Solver-v3 and UniPC. Our findings strengthen the field's theoretical understanding of how diffusion models work and suggest ways to improve the design and training of diffusion models.

## 1 Introduction

Diffusion models (Sohl-Dickstein et al., 2015; Song & Ermon, 2019; Ho et al., 2020; Song et al., 2021) have revolutionized the field of generative modeling by achieving remarkable performance across diverse domains, including image (Rombach et al., 2022), audio (Kong et al., 2021; Chen et al., 2021; Popov et al., 2021), and video (Harvey et al., 2022; Ho et al., 2022; Blattmann et al., 2023) generation. Despite these successes, *why* diffusion models perform as well as they do is poorly understood. Two major open questions are as follows.

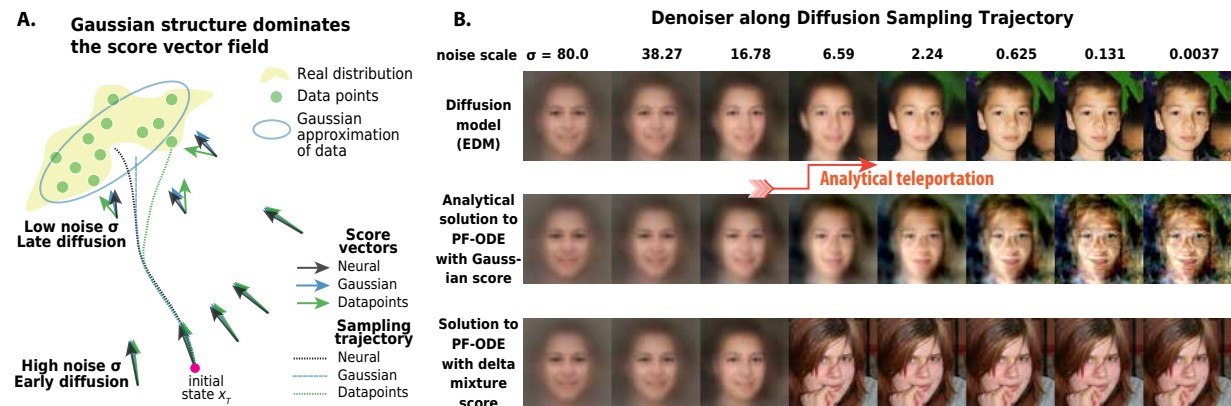

Figure 1: **Gaussian score well-approximates the learned neural score**. **A.** Schematic illustrating our work's main claim. In the high noise regime, the neural score is well-approximated by both the Gaussian/linear score and the score of the training set; in the low noise regime, neural scores are better described by the Gaussian model. **B.** Visual demonstration of the effect. Denoiser outputs along PF-ODE trajectories were similar at high noise, regardless of the score model (neural score, Gaussian score, or delta mixture score); at small noise scales, neural denoiser outputs more closely resemble those of the Gaussian model.

First, given that samples are generated via a dynamic process from noise, *when* do different sample features emerge, and *what* controls which features appear in the final sample? Second, given the empirical fact that diffusion models often generalize beyond their training data, what distribution do they learn instead?

Central to understanding these models is characterizing the score function, the dynamic vector field learned by a neural network during training. By modeling the gradient of the smoothed data distribution, the score function guides the iterative sample generation process. Curiously, it has been observed to deviate from its theoretically expected behavior: it may not be the gradient of a mixture of data points, and it may not even be a gradient field at all (Wenliang & Moran, 2023). Such deviations raise fundamental questions about the nature of what networks learn and how they generalize training data.

How does the learned score function compare to that of the underlying data manifold, especially given that neural networks only have access to a finite set of training examples in practice? Critically, a network that learns the exact score function of the training set—i.e., a mixture of delta functions centered on training data—can only reproduce training examples, and hence cannot generalize. This observation suggests that neural networks optimize for a balance between learning the score function of the training set and capturing aspects of the underlying data manifold. However, the precise characteristics of the learned score function, and how the development of this balance proceeds over the course of training, remain largely unexplored.

One simple possibility is that diffusion models generalize in part by learning the score function associated with certain *summary statistics* of the traning set, like its mean and covariance matrix. If this were true, one would expect the score function to be that of a Gaussian model, i.e., linear in its state features. While it is decidedly *not* true that learned neural scores are Gaussian, we find that this is much *closer* to being true than one might expect. To show this, we first mathematically characterize the properties of the Gaussian model. Next, we compare the Gaussian model's score function to real score functions, and find that the Gaussian model well-approximates the behavior of neural network models for moderate to high noise levels (Fig. 1A). Finally, we show how this insight can be applied to accelerate sampling from diffusion models (Fig. 1B).

## 1.1 Main contributions

Our main contributions are as follows:

- We thoroughly analyze the Gaussian score model, including characterizing its sampling trajectories by exactly solving the associated probability flow ODE (PF-ODE). (Sec. 3.1-3.2)

- We show how the Gaussian model recapitulates nontrivial features of 'real' sample generation, including the low dimensionality of sampling trajectories, the time course of feature emergence, and the effect of perturbations during sampling. (Sec. 3.3)

- We theoretically (Sec. 4.1) and empirically (Sec. 4.2-4.3) support our claim that, in the high-noise regime, the score field of real diffusion models is dominated by its Gaussian/linear approximation.

- We find that learned neural scores align more closely with those of low-rank Gaussian mixtures than the (delta mixture) score of the training distribution. (Sec. 4.4)

- We characterize the learning dynamics of neural score functions, and in particular find that Gaussian/linear structure is preferentially learned early in training. (Sec. 5)

- We apply our findings by using the Gaussian model solution to accelerate sampling. (Sec. 6)

## 2 Mathematical formulation

In this section, we review the mathematics of diffusion models and define the idealized score models of interest. To streamline notation and match popular implementations of diffusion models, we use the "EDM" framework of Karras et al. (2022). This choice implies no loss of generality, as a simple reparametrization allows one to map to other formalisms (Song et al., 2021; Ho et al., 2020). See Appendix D.2 for more details on this point.

### 2.1 Basics of score-based modeling

Let $p_{data}(\mathbf{x})$ be a data distribution in $\mathbb{R}^D$. The core idea of diffusion models is to corrupt this distribution with (usually Gaussian) noise, and to learn to undo this corruption; this way, one can sample from the generally complex distribution $p(\mathbf{x})$ by first sampling from a Gaussian, and then iteratively removing noise. The noise-corrupted distribution is defined as

$$p(\mathbf{x}; \sigma) := \int_{\mathbb{R}^D} d\mathbf{x}' \, p(\mathbf{x}|\mathbf{x}', \sigma) \, p_{data}(\mathbf{x}') = \int_{\mathbb{R}^D} d\mathbf{x}' \, \mathcal{N}(\mathbf{x}; \mathbf{x}', \sigma^2 \mathbf{I}) \, p_{data}(\mathbf{x}') \tag{1}$$

where $\sigma \geq 0$ is the noise scale. There are many ways to remove noise, but a popular method introduced by Song et al. (2021) is to utilize the PF-ODE, which in the Karras et al. (2022) formulation has the form

$$d\mathbf{x} = -\dot{\sigma}_t \sigma_t \, \mathbf{s}(\mathbf{x}, \sigma_t) \, dt \tag{2}$$

where $t$ denotes time and $\sigma_t$ denotes the noise scale at time $t$ (with $\dot{\sigma}_t$ its derivative)[1]. The key ingredient of this process is the score function $\mathbf{s}(\mathbf{x}, \sigma) := \nabla_{\mathbf{x}} \log p(\mathbf{x}; \sigma)$, i.e., the gradient of the noise-corrupted data distribution. To generate a sample, one samples $\mathbf{x}_T \sim \mathcal{N}(\mathbf{x}; \sigma_T^2 \mathbf{I})$ with a large $\sigma_T = \sigma_{max}$, and then integrates Eq. 2 backward in time until $\sigma_{t_{min}} = \sigma_{min} \approx 0$. Since we are interested in understanding these dynamics in detail, we must carefully study the dynamic vector field $\mathbf{s}(\mathbf{x}, \sigma)$.

There are various ways to learn a parameterized score approximator $\hat{\mathbf{s}}_{\boldsymbol{\theta}}(\mathbf{x}, \sigma)$ (Yang et al., 2023), including score matching (Hyvärinen, 2005) and sliced score matching (Song et al., 2020b), but the current most popular and performant approach is denoising score matching (Raphan & Simoncelli, 2006; Vincent, 2011; Song & Ermon, 2019). The corresponding objective, which is minimized by the score function of the training set, is

$$\mathcal{L}_{\boldsymbol{\theta}} = \mathbb{E}_{\mathbf{y} \sim p_{data}} \mathbb{E}_{\mathbf{n} \sim \mathcal{N}(0, \sigma^2 \mathbf{I})} \|\mathbf{D}_{\boldsymbol{\theta}}(\mathbf{y} + \mathbf{n}, \sigma) - \mathbf{y}\|_2^2 \tag{3}$$

where $\mathbf{y}$ denotes a sample from training data, and where the learned denoiser $\mathbf{D}_{\theta}(\mathbf{x}, \sigma)$ relates to the learned score function via

$$\hat{\mathbf{s}}_{\boldsymbol{\theta}}(\mathbf{x}, \sigma) := \frac{\mathbf{D}_{\boldsymbol{\theta}}(\mathbf{x}, \sigma) - \mathbf{x}}{\sigma^2} \ . \tag{4}$$

Sample generation dynamics are most usefully understood in terms of this denoiser, which supplies an estimate $\hat{\mathbf{x}}_0 = \mathbf{D}_{\boldsymbol{\theta}}(\mathbf{x}, \sigma)$ of the PF-ODE's endpoint given the current state $\mathbf{x}$ and noise scale $\sigma^2$[2]. The mathematical question we are most interested in becomes: *how does $\hat{\mathbf{x}}_0$ evolve throughout sample generation?*

---

[1]Although EDM conventionally chooses $\sigma_t = t$, we do not assume this to keep our results slightly more general.

[2]We will use the terms 'denoiser output' $\mathbf{D}$ and 'endpoint estimate' $\hat{\mathbf{x}}_0$ interchangeably in what follows.

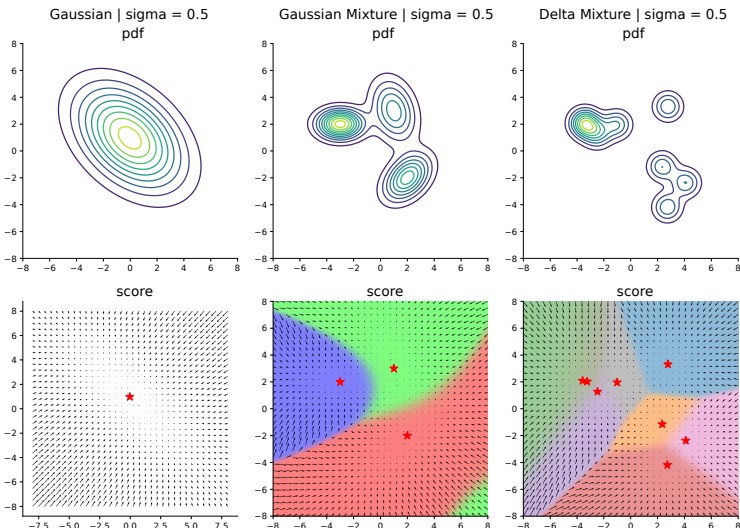

Figure 2: **Structure of Gaussian, Gaussian mixture, and delta mixture scores. Upper row:** Examples of 2D probability density functions for Gaussian, Gaussian mixture model (GMM), and delta mixture distributions. **Lower row:** The respective score vector fields. For mixture models, the space is colored based on the weighting function $w_i(\mathbf{x})$ from Eq. 7, with a unique color (red, blue, or green) assigned to each Gaussian component.

**Connection to alternative frameworks.** In Eq. 2, the norm of $\mathbf{x}_t$ changes significantly over time. Many alternative frameworks prefer a probability flow ODE or SDE where the norm of the state remains more stable, such as DDPM (Ho et al., 2020), DDIM (Song et al., 2020a), and VP-SDE (Song et al., 2021). As noted by Karras et al. (2022), these formulations produce dynamics equivalent to Eq. 2 when $\mathbf{x}_t$ is rescaled by a time-dependent factor $\mathbf{x}_t \to \tilde{\mathbf{x}}_t = \alpha_t \mathbf{x}_t$ and time is reparametrized, allowing their solutions to be directly mapped to one another. For simplicity, we present most of our results (except some in Sec. 3.3) without the time-dependent scaling factors; however, with minor modifications, these results are applicable to models with state scaling. Detailed results for alternative formulations are provided in Appendices D.3 and B.2.

## 2.2 Idealized score models of interest

In this paper, we will try to understand the score functions learned by real diffusion models by comparing them to the score functions of certain idealized distributions. We consider three such idealized score models: the Gaussian model, which assumes only the overall mean and covariance of the data are learned; the score of the delta mixture distribution, which is the *Exact* score of the training dataset, assuming training data are precisely memorized and that no generalization occurs; and the Gaussian mixture model (*GMM*), which lies somewhere between the previous two models. Below, we write down some basic but important properties of each of these idealized score models.

**Gaussian model.** Suppose the target distribution is a Gaussian distribution with mean $\boldsymbol{\mu} \in \mathbb{R}^D$ and covariance $\boldsymbol{\Sigma} \in \mathbb{R}^{D \times D}$. Since the effective dimensionality of data manifolds is often much lower than the dimensionality of the ambient space (Turk & Pentland, 1991; Hinton & Salakhutdinov, 2006; Camastra & Staiano, 2016) (consider, e.g., the dimensionality of image manifolds versus the dimensionality of pixel space), we allow $\boldsymbol{\Sigma}$ to have rank $r \leq D$. At noise scale $\sigma$, the score is that of $\mathcal{N}(\boldsymbol{\mu}, \boldsymbol{\Sigma} + \sigma^2 \mathbf{I})$, which reads

$$\mathbf{s}(\mathbf{x}, \sigma) = (\sigma^2 \mathbf{I} + \boldsymbol{\Sigma})^{-1}(\boldsymbol{\mu} - \mathbf{x}) . \tag{5}$$

The optimal denoiser reads

$$\mathbf{D}(\mathbf{x}, \sigma) = \frac{\sigma^2 \mathbf{I}}{\sigma^2 \mathbf{I} + \boldsymbol{\Sigma}} \boldsymbol{\mu} + \frac{\boldsymbol{\Sigma}}{\sigma^2 \mathbf{I} + \boldsymbol{\Sigma}} \mathbf{x} , \tag{6}$$

and can be interpreted as a certainty-weighted combination of the distribution mean and the current state[3]. Early in sample generation, when $\sigma^2$ is large, it dominates the signal variances, so the distribution mean is a

---

[3]We slightly abuse notation to write this expression more suggestively, since $\sigma^2 \mathbf{I} + \boldsymbol{\Sigma}$ is invertible and commutes with $\boldsymbol{\Sigma}$.

reasonable estimate. Later in sample generation, when $\sigma^2$ is smaller, the current state is more informative about the outcome. Notice that different components of $\mathbf{x}$ contribute differently to $\mathbf{D}(\mathbf{x}, \sigma)$ depending on their corresponding signal variances; we will elaborate on this point in the next section (see Eq. 16).

**Gaussian mixture model.** Suppose the target distribution is a Gaussian mixture with $K$ components. Denote the weight of mode $i$ by $\pi_i$, and the corresponding mean and covariance by $\boldsymbol{\mu}_i$ and $\boldsymbol{\Sigma}_i$. At noise scale $\sigma$, the score function and optimal denoiser of this Gaussian mixture model (GMM) read

$$
\begin{aligned}
\mathbf{s}(\mathbf{x}, \sigma) &= \sum_{i=1}^{K} w_i(\mathbf{x}, \sigma)(\sigma^2 \mathbf{I} + \boldsymbol{\Sigma}_i)^{-1}(\boldsymbol{\mu}_i - \mathbf{x}) \\
\mathbf{D}(\mathbf{x}, \sigma) &= \sum_{i=1}^{K} w_i(\mathbf{x}, \sigma) \left[ \frac{\sigma^2 \mathbf{I}}{\sigma^2 \mathbf{I} + \boldsymbol{\Sigma}_i} \boldsymbol{\mu}_i + \frac{\boldsymbol{\Sigma}_i}{\sigma^2 \mathbf{I} + \boldsymbol{\Sigma}_i} \mathbf{x} \right] .
\end{aligned}
\tag{7}
$$

Note that both the score and denoiser are simply weighted sums of the score/denoiser of each mode. The weighting function $w_i$ that determines each mode's contribution is

$$
w_i(\mathbf{x}, \sigma) := \frac{\pi_i \mathcal{N}(\mathbf{x}; \boldsymbol{\mu}_i, \sigma^2 \mathbf{I} + \boldsymbol{\Sigma}_i)}{\sum_j^K \pi_j \mathcal{N}(\mathbf{x}; \boldsymbol{\mu}_j, \sigma^2 \mathbf{I} + \boldsymbol{\Sigma}_j)} = \text{softmax}(\ \log \pi_i + \log \mathcal{N}(\mathbf{x}; \boldsymbol{\mu}_i, \sigma^2 \mathbf{I} + \boldsymbol{\Sigma}_i)\ ) .
\tag{8}
$$

When the noise scale is small, the function $w_i(\mathbf{x}, \sigma)$ becomes one-hot, and the GMM score is locally identical to the Gaussian score of the highlighted component (Fig. 2). Thus, the Gaussian mixture score can be interpreted as piecing together different Gaussian scores.

**Delta mixture (exact) score model.** Consider a dataset $\{\mathbf{y}_i\}_{i=1}^N$ of $N$ data points. The corresponding delta mixture distribution and its noise-corrupted version can be written

$$
p(\mathbf{x}) = \frac{1}{N} \sum_{i=1}^{N} \delta(\mathbf{x} - \mathbf{y}_i) \qquad\qquad p(\mathbf{x}; \sigma) = \frac{1}{N} \sum_{i=1}^{N} \mathcal{N}(\mathbf{x}; \mathbf{y}_i, \sigma^2 \mathbf{I}) .
\tag{9}
$$

The corresponding score function and optimal denoiser are

$$
\mathbf{s}(\mathbf{x}, \sigma) = \sum_{i=1}^{N} w_i(\mathbf{x}, \sigma) \frac{(\mathbf{y}_i - \mathbf{x})}{\sigma^2} \qquad\qquad \mathbf{D}(\mathbf{x}, \sigma) = \sum_{i=1}^{N} w_i(\mathbf{x}, \sigma)\ \mathbf{y}_i
\tag{10}
$$

$$
w_i(\mathbf{x}, \sigma) := \frac{\exp\left\{ -\frac{1}{2\sigma^2} \|\mathbf{x} - \boldsymbol{\mu}_i\|_2^2 \right\}}{\sum_j^K \exp\left\{ -\frac{1}{2\sigma^2} \|\mathbf{x} - \boldsymbol{\mu}_j\|_2^2 \right\}} = \text{softmax}(\ -\frac{1}{2\sigma^2} \|\mathbf{x} - \boldsymbol{\mu}_i\|_2^2\ ) .
\tag{11}
$$

This model's optimal denoiser is a weighted combination of training examples. The noise scale $\sigma$ acts like a temperature parameter: in the large $\sigma$ limit, the score points towards the data mean; in the $\sigma \to 0$ limit, the score pushes with infinitely strong 'force' towards the training example closest to the current state $\mathbf{x}$. Thus, generating samples using this score always (in the absence of numerical errors) reproduces training examples.

This score model is special, since it is the exact score of the finite training dataset (without augmentation), i.e., the score function minimizing the score matching objective (Eq. 3). Since we expect a sufficiently expressive score approximator to converge to this model in the absence of additional influences (e.g., regularization and early stopping), comparing learned scores to this model speaks to the question of generalization. If a score approximator learns something *other* than this, then the model has been implicitly regularized and generalizes beyond the training set to some extent.

## 3 Exact solution and interpretation of the Gaussian score model

In this section, we present the exact solution to the Gaussian model. The utility of this model is that it is simple enough that it can be solved exactly, and hence one can precisely quantify various aspects of sample generation dynamics. First, we briefly describe the solution method (Sec. 3.1-3.2), and then we discuss the qualitative insights we can obtain from it (Sec. 3.3). In Sec. 4-5, we will show how this admittedly simple model relates to real diffusion models.

### 3.1 Solution method: Exploiting a decomposition of the covariance matrix

Since the covariance $\mathbf{\Sigma}$ is symmetric and positive semidefinite, it has a compact singular value decomposition $\mathbf{\Sigma} = \mathbf{U}\mathbf{\Lambda}\mathbf{U}^T$, where $\mathbf{U} = [\mathbf{u}_1, ..., \mathbf{u}_r]$ is a $D \times r$ semi-orthogonal matrix and $\mathbf{\Lambda} \in \mathbb{R}^{r \times r}$ is diagonal with $\lambda_k := \Lambda_{kk} > 0$ for all $k$. The columns of $\mathbf{U}$, $\mathbf{u}_k$, are the principal component (PC) axes along which the Gaussian mode varies, and their span comprises the 'data manifold' of the Gaussian model.

This decomposition is useful since it allows us to write the score and denoiser of the Gaussian model in a more explicit form. Using the Woodbury identity (Woodbury, 1950),

$$(\sigma^2 \mathbf{I} + \mathbf{U}\mathbf{\Lambda}\mathbf{U}^T)^{-1} = \frac{1}{\sigma^2}\mathbf{I} - \frac{1}{\sigma^4}\mathbf{U}\left(\mathbf{\Lambda}^{-1} + \frac{1}{\sigma^2}\mathbf{I}\right)^{-1}\mathbf{U}^T = \frac{1}{\sigma^2}\left[\mathbf{I} - \mathbf{U}\,\mathrm{diag}\left[\frac{\lambda_k}{\lambda_k + \sigma^2}\right]\mathbf{U}^T\right].$$

For convenience, define the diagonal matrix $\tilde{\mathbf{\Lambda}}_\sigma := \mathrm{diag}[\frac{\lambda_k}{\lambda_k + \sigma^2}]$. We can now write

$$\mathbf{s}(\mathbf{x}, \sigma) = \frac{1}{\sigma^2}\left(\mathbf{I} - \mathbf{U}\tilde{\mathbf{\Lambda}}_\sigma \mathbf{U}^T\right)(\boldsymbol{\mu} - \mathbf{x}) \qquad \mathbf{D}(\mathbf{x}, \sigma) = \boldsymbol{\mu} + \mathbf{U}\tilde{\mathbf{\Lambda}}_\sigma \mathbf{U}^T(\mathbf{x} - \boldsymbol{\mu}) . \tag{12}$$

Slightly more explicitly, the optimal denoiser can be written

$$\mathbf{D}(\mathbf{x}, \sigma) = \boldsymbol{\mu} + \sum_{k=1}^{r} \frac{\lambda_k}{\lambda_k + \sigma^2}\left[\mathbf{u}_k \cdot (\mathbf{x} - \boldsymbol{\mu})\right]\mathbf{u}_k . \tag{13}$$

### 3.2 Closed-form solution of PF-ODE for Gaussian model

Using the rewritten score (Eq. 12), the PF-ODE (Eq. 2) takes a particularly simple form:

$$\dot{\mathbf{x}}_t = \frac{\dot{\sigma}}{\sigma}(\mathbf{I} - \mathbf{U}\tilde{\mathbf{\Lambda}}_\sigma \mathbf{U}^T)(\mathbf{x}_t - \boldsymbol{\mu}) . \tag{14}$$

The above ODE is linear, and its dynamics along each principal axis $\boldsymbol{u}_k$ are independent. Solving it in the usual way (see Appendix D.1), we find

$$\mathbf{x}_t = \boldsymbol{\mu} + \frac{\sigma_t}{\sigma_T}\,\mathbf{x}_T^\perp + \sum_{k=1}^{r}\psi(t, \lambda_k)c_k(T)\,\mathbf{u}_k \qquad\qquad \psi(t, \lambda) := \sqrt{\frac{\sigma_t^2 + \lambda}{\sigma_T^2 + \lambda}} \tag{15}$$

$$\mathbf{x}_T^\perp := (\mathbf{I} - \mathbf{U}\mathbf{U}^T)(\mathbf{x}_T - \boldsymbol{\mu}) \qquad\qquad c_k(T) := \mathbf{u}_k^T(\mathbf{x}_T - \boldsymbol{\mu}) .$$

The solution has three components: (i) the distribution mean, (ii) an off-manifold component, and (iii) an on-manifold component. The distribution mean term does not change throughout sample generation. The off-manifold component shrinks to zero as $t \to 0$. The on-manifold component, which is determined by the manifold-projected difference between $\mathbf{x}$ and $\boldsymbol{\mu}$, evolves independently according to $\psi(t, \lambda)$ along each PC direction.

This solution allows us to characterize the evolution of the denoiser output given the initial state $\mathbf{x}_T$:

$$\mathbf{D}(t) := \mathbf{D}(\mathbf{x}_t, \sigma_t) = \boldsymbol{\mu} + \sum_{k=1}^{r}\xi(t, \lambda_k)c_k(T)\,\mathbf{u}_k \qquad \xi(t, \lambda) := \frac{\lambda}{\sqrt{(\lambda + \sigma_t^2)(\lambda + \sigma_T^2)}} . \tag{16}$$

Finally, the exact solution also allows us to explicitly write the mapping from $\mathbf{x}_T$ to the final state $\mathbf{x}_0$:

$$\mathbf{x}_0 = \boldsymbol{\mu} + \sum_{k=1}^{r}\sqrt{\frac{\lambda_k}{\sigma_T^2 + \lambda_k}}\,\mathbf{u}_k\mathbf{u}_k^T(\mathbf{x}_T - \boldsymbol{\mu}) . \tag{17}$$

For the Gaussian model, the final sample is determined by the combined influence of the covariance structure of data, and the overlap between the vector $\mathbf{x}_T - \boldsymbol{\mu}$ and the data manifold.

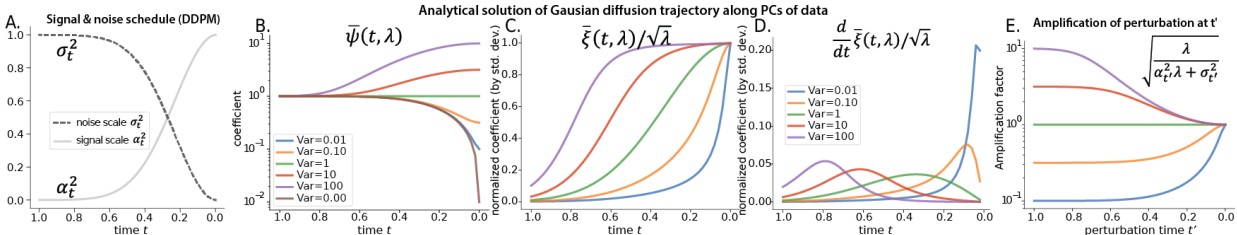

Figure 3: **Analytical solution to sample generation dynamics for Gaussian model. A.** The noise and signal schedule $\sigma_t^2$ and $\alpha_t^2$ from `ddpm-CIFAR-10`. **B.** $\bar{\psi}(t,\lambda)$ governs the dynamics of $\mathbf{x}_t$ along each principal axis $\mathbf{u}_k$. **C.** $\bar{\xi}(t,\lambda)$ governs the dynamics of the endpoint estimate $\hat{\mathbf{x}}_0(\mathbf{x}_t)$ along each PC, normalized by the standard deviation $\sqrt{\lambda}$. **D.** Time derivative of $\bar{\xi}(t,\lambda)/\sqrt{\lambda}$, highlighting the 'critical period' when each feature develops the most rapidly. **E.** $\sqrt{\lambda/(\sigma_{t'}^2 + \lambda\alpha_{t'}^2)}$, which quantifies the amplification effect of a perturbation along PC $\mathbf{u}_k$ at time $t'$ (Eq. 23).

**Solution with data scaling term.** The popular VP-SDE (Song et al., 2021) is an alternative to the EDM formulation, and its forward process is characterized by the transition probability $p(\mathbf{x}_t|\mathbf{x}_0) = \mathcal{N}(\alpha_t\mathbf{x}_0, \sigma_t^2\mathbf{I})$. Using the VP-SDE is equivalent to introducing a time-dependent scaling term $\alpha_t$ in Eq. 2. Thus, we can obtain the solution by substituting $\mathbf{x}_t \mapsto \mathbf{x}_t/\alpha_t$ and $\sigma_t \mapsto \sigma_t/\alpha_t$. The solution reads

$$\mathbf{x}_t = \alpha_t\mu + \frac{\sigma_t}{\sigma_T}\,\bar{\mathbf{x}}_T^{\perp} + \sum_{k=1}^{r} \bar{\psi}(t,\lambda_k)\bar{c}_k(T)\mathbf{u}_k \qquad \bar{\psi}(t,\lambda) := \sqrt{\frac{\sigma_t^2 + \lambda\alpha_t^2}{\sigma_T^2 + \lambda\alpha_T^2}} \tag{18}$$

$$\bar{\mathbf{x}}_T^{\perp} := (\mathbf{I} - \mathbf{U}^T\mathbf{U})(\mathbf{x}_T - \alpha_T\mu) \qquad \bar{c}_k(T) := \mathbf{u}_k^T(\mathbf{x}_T - \alpha_T\mu) . \tag{19}$$

$$\mathbf{D}(t) = \mu + \sum_{k=1}^{r} \bar{\xi}(t,\lambda_k)\bar{c}_k(T)\mathbf{u}_k \qquad \bar{\xi}(t,\lambda) := \frac{\alpha_t\lambda}{\sqrt{(\alpha_t^2\lambda + \sigma_t^2)(\alpha_T^2\lambda + \sigma_T^2)}} . \tag{20}$$

When $\alpha_t = 1$, these solutions reduce to Eq. 15 and 16 as expected.

## 3.3 Interpreting Gaussian generation dynamics

The closed-form solution of the Gaussian model provides the exact relationship between the covariance matrix and sample generation dynamics. Since the dynamics are separable along each PC, we only need to study the functions $\psi$ and $\xi$ to describe how the variance of the data distribution along each PC influences dynamics along that direction. Here, we highlight interesting consequences of the Gaussian model's exact solution related to four aspects of sample generation: 1) the determination of the final sample, 2) the geometry of trajectories, 3) the feature emergence order, and 4) the effect of perturbations.

In this subsection, we describe these consequences in the context of the VP-SDE formulation, which includes the additional $\alpha_t$ data-scaling factor. The signal scale $\alpha_t$ is assumed to decrease with time, the noise scale is assumed to increase with time, and we

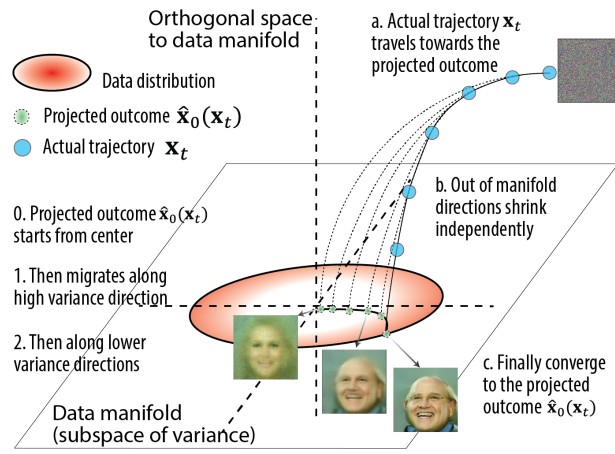

Figure 4: **Geometry of sample generation trajectories for the VP-SDE.**

also assume $\alpha_t^2 + \sigma_t^2 = 1$ at all times. We visualize $\alpha_t, \sigma_t$ along with the functions $\bar{\psi}(t,\lambda), \bar{\xi}(t,\lambda)$ to provide intuition about the Gaussian solution (Fig. 3).

**Mapping from initial noise $\mathbf{x}_T$ to final sample $\mathbf{x}_0$.** The explicit mapping between the initial noise pattern and final sample, which is given by

$$\mathbf{x}_0 = \boldsymbol{\mu} + \sum_{k=1}^{r} \bar{\psi}(0, \lambda_k) \, \mathbf{u}_k \mathbf{u}_k^T (\mathbf{x}_T - \alpha_T \boldsymbol{\mu}) \;, \tag{21}$$

is reminiscent of *linear filtering*. The final location of $\mathbf{x}_0$ along each axis $\mathbf{u}_k$ is determined by the projection of the (mean subtracted) initial noise onto that axis, amplified by the standard deviation $\bar{\psi}(0, \lambda_k) \approx \sqrt{\lambda_k}$. Thus, the subtle alignment between the initial noise pattern and the covariance of the data distribution determine what is generated. Note that even if the initial overlap is dominated by particular features, due to the amplification effect of $\bar{\psi}(0, \lambda_k)$ the final sample may be dominated by other features that vary more.

**Trajectory resembles 2D rotation.** The solution for $\mathbf{x}_t$ also implies that

$$\mathbf{x}_t \approx \alpha_t \mathbf{x}_0 + \sqrt{1 - \alpha_t^2} \, \mathbf{x}_T + \sum_{k=1}^{r} \left\{ \sqrt{\sigma_t^2 + \lambda_k \alpha_t^2} - \alpha_t \sqrt{\lambda_k} - \sigma_t \right\} c_k(T) \mathbf{u}_k \;,$$

i.e., that $\mathbf{x}_t$ dynamics tend to look like a rotation or a spherical interpolation within the 2D plane formed by $\mathbf{x}_0$ and $\mathbf{x}_T$, up to on-manifold correction terms (see Appendix D.4 for the derivation and more discussion). The correction terms tend to be small; assuming $r \ll D$, and that the typical overlap between the initial noise and any given eigendirection is roughly $1/\sqrt{D}$,

$$\left\| \mathbf{x}_t - \alpha_t \mathbf{x}_0 - \sqrt{1 - \alpha_t^2} \, \mathbf{x}_T \right\|_2^2 \leq \left( 1 - \frac{\sqrt{2}}{2} \right)^2 \frac{r}{D} \ll 1 \;. \tag{22}$$

**Feature emergence order.** The solution of the denoiser trajectory $\mathbf{D}(t)$ (Eq. 20) implies that the outputs of the denoiser remain on the data manifold throughout sample generation. This explains why the outputs of well-trained models resemble noise-free images, rather than images contaminated with noise.

Initially, as all $\bar{\xi}(T, \lambda) \approx 0$, the expected outcome $\mathbf{D}(T)$ starts close to the distribution mean $\boldsymbol{\mu}$. For a face dataset, this might resemble the so-called 'generic' face (Langlois & Roggman, 1990). The denoiser output moves along each PC direction according to the $\bar{\xi}$ function (Fig. 3C). The sigmoidal shape of $\bar{\xi}(t, \lambda)$ indicates that the feature associated with a given PC changes the most when $\sigma_t \approx \alpha_t \sqrt{\lambda}$, i.e., when the noise variance matches the scaled signal variance. After that point, the feature stabilizes, suggesting a "*critical period*" of development for each feature (Fig. 3D). Moreover, since the critical period happens when $\alpha_t \approx 1/\sqrt{1 + \lambda}$, features appear *in order of descending variance*. The highest-variance features appear earliest, and as generation proceeds more and more lower-variance features are unveiled.

This has interesting implications for image diffusion models. Natural images have more variance in low frequencies than high frequencies (Ruderman, 1994). Furthermore, for face images, 'semantic' features—like gender, head orientation, skin color, and backgrounds—have higher variance than subtle features such as glasses, facial, and hair textures (Wang & Ponce, 2021). Hence, facts about natural image statistics and sample generation dynamics together explain why features like scene layout are specified first in the endpoint estimate—or equivalently why generation is usually, outline-first, details later.

**Effect of perturbations.** Finally, we examined the effect of perturbations on feature commitment. Suppose at time $t' \in (0, T)$ the off-manifold directions are perturbed by $\delta \mathbf{x}^{\perp}$ and the on-manifold direction $\mathbf{u}_k$ is perturbed by an amount $\delta c_k$. The effect of the perturbation on the generated sample $\mathbf{x}_0$ is

$$\Delta \mathbf{x}_0 = \sum_{k=1}^{r} \frac{\bar{\psi}(0, \lambda_k)}{\bar{\psi}(t', \lambda_k)} \delta c_k \mathbf{u}_k = \sum_{k=1}^{r} \sqrt{\frac{\lambda_k}{\sigma_{t'}^2 + \lambda_k \alpha_{t'}^2}} \delta c_k \mathbf{u}_k \;. \tag{23}$$

Thanks to denoising, an off-manifold perturbation has no effect on the sample, while on-manifold perturbations have *varying effects depending on timing*. We visualized the amplification factor $\sqrt{\frac{\lambda}{\sigma_{t'}^2 + \lambda \alpha_{t'}^2}}$ of a perturbation as a function of perturbation time $t'$ and PC variance $\lambda$ (Fig. 3 E).

Perturbations along high variance axes ($\lambda > 1$) have an amplified effect when injected at the start and less amplification when injected later. Conversely, perturbations along low variance axes ($\lambda < 1$) have a diminished effect when injected early, and a less suppressed effect when injected later. This is consistent with the notion that features have a variance-dependent 'critical period' during which they are most easily affected by perturbations.

Even if a random perturbation is injected, depending on the timing $t'$, features with different variances will be most amplified. This time-dependent 'filtering' effect explains the classic finding (Ho et al., 2020) that early perturbations create variations of layout and semantic features, and late perturbations modulate high-frequency details.

**Summary.** The solution of the Gaussian model (Eq. 18) suggests a geometric picture of sample generation, which we depict in Fig. 4: the endpoint estimate $\hat{\mathbf{x}}_0$ begins at the center of the distribution and travels along the data manifold, moving first along the high variance axes and then along the lower variance axes; concurrently, the state $\mathbf{x}_t$ in the ambient space rotates towards the evolving endpoint estimate. Despite the Gaussian model's simplicity, various aspects of its behavior are qualitatively consistent with substantially more complex models like Stable Diffusion (Fig. 5).

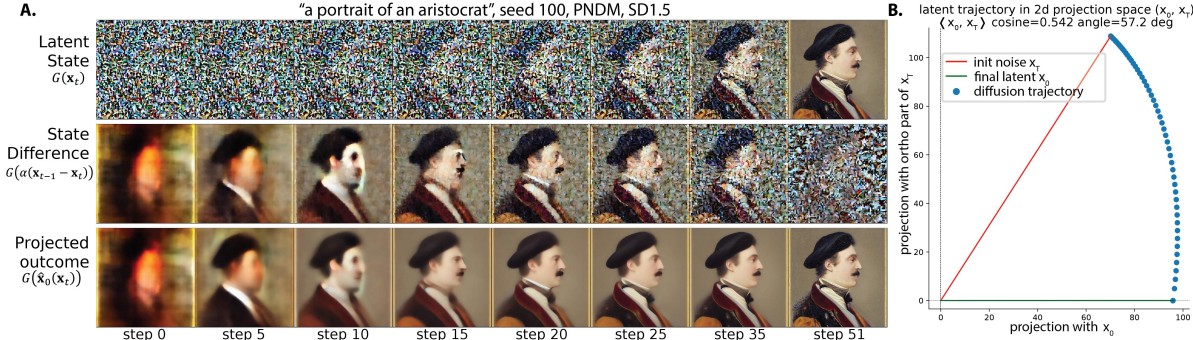

Figure 5: **Qualitative aspects of Stable Diffusion sample generation consistent with Gaussian theory**. **A**. Trajectories of states $G(\mathbf{x}_t)$ (top row), scaled differences between nearby states $G(k(\mathbf{x}_{t-1} - \mathbf{x}_t))$ (middle row), and denoiser / projected outcome $G(\hat{\mathbf{x}}_0(\mathbf{x}_t))$ (bottom row). $G$ denotes Stable Diffusion's decoder, which converts latent states to images. **B**. Trajectory of $\mathbf{x}_t$ projected onto the plane spanned by $\mathbf{x}_T$ and $\mathbf{x}_0$. Trajectories are effectively two-dimensional, and resemble a rotation from $\mathbf{x}_T$ to $\mathbf{x}_0$. Note that both feature emergence order and the low dimensionality of trajectories are consistent with the Gaussian model.

## 4 The far-field Gaussian structure of real diffusion models

Most distributions interesting enough to train generative models on are not Gaussian, so how is the exact solution of the Gaussian model related to real diffusion models trained on complex datasets? Our central claim is that, at moderate to high noise scales, the score function of an arbitrary bounded point cloud is nearly indistinguishable from that of a Gaussian with matching mean and covariance. We call such structure "*far-field*"; we borrow the term from physics, where it describes a region far enough from the source of (e.g., electromagnetic) waves that the size and shape of the source can be neglected.

First, we provide theoretical arguments to support this claim (Sec. 4.1). Second, we present two empirical analyses that support it. In Sec. 4.2 and 4.3, we study how well the Gaussian model approximates the score function, sampling trajectories, and samples generated by pre-trained diffusion models. In Sec. 4.4, we go beyond the Gaussian model and systematically compare the learned neural score with the scores of GMMs with varying numbers of components and covariance matrix ranks. In the following section (Sec. 5), we track the neural score field throughout training and examine when different types of score structure emerge.

### 4.1 Theoretical basis for far-field Gaussian score structure

Consider a point cloud $\{\mathbf{y}_i\}_{i=1}^N \subseteq \mathbb{R}^D$. When the noise level is high and/or the query point is far from the point cloud's support, we claim that the score of this distribution is nearly indistinguishable from that of a Gaussian with the same mean and covariance. Here, we provide a theoretical argument for this claim.

Let $\boldsymbol{\mu}$ and $\boldsymbol{\Sigma}$ denote the mean and covariance of the point cloud. Recall from Sec. 2.2 that the score function of the (noise-corrupted) point cloud is

$$
\begin{aligned}
\mathbf{s}(\mathbf{x}; \sigma) &= \sum_i w_i(\mathbf{x}) \frac{(\mathbf{y}_i - \mathbf{x})}{\sigma^2} = \frac{\boldsymbol{\mu} - \mathbf{x}}{\sigma^2} + \sum_i w_i(\mathbf{x}) \frac{(\mathbf{y}_i - \boldsymbol{\mu})}{\sigma^2} \\
w_i(\mathbf{x}) &= \frac{\exp\left(-\frac{1}{2\sigma^2}\|\mathbf{x} - \mathbf{y}_i\|_2^2\right)}{\sum_{j=1}^N \exp\left(-\frac{1}{2\sigma^2}\|\mathbf{x} - \mathbf{y}_j\|_2^2\right)} = \frac{\exp\left(-\frac{1}{2\sigma^2}\|\boldsymbol{\mu} - \mathbf{y}_i\|_2^2 + \frac{1}{\sigma^2}(\mathbf{x} - \boldsymbol{\mu})^T(\mathbf{y}_i - \boldsymbol{\mu})\right)}{\sum_{j=1}^N \exp\left(-\frac{1}{2\sigma^2}\|\boldsymbol{\mu} - \mathbf{y}_j\|_2^2 + \frac{1}{\sigma^2}(\mathbf{x} - \boldsymbol{\mu})^T(\mathbf{y}_j - \boldsymbol{\mu})\right)} \, .
\end{aligned}
\tag{24}
$$

Note that we have rearranged the score to be written in terms of distances *to the point cloud mean* $\boldsymbol{\mu}$. In the far-field regime, one expects this distance to account for most of the distance to each data point.

Next, note that $(\mathbf{x} - \boldsymbol{\mu})^T(\mu - \mathbf{y}_i)$ is on the order of $\sigma\sqrt{\mathrm{tr}\boldsymbol{\Sigma}}$ and $\|\boldsymbol{\mu} - \mathbf{y}_i\|_2^2$ is on the order of $\mathrm{tr}\boldsymbol{\Sigma}$ (see Appendix D.8 for a detailed derivation). Hence, when $\sigma \gg \sqrt{\mathrm{tr}\boldsymbol{\Sigma}}$—i.e., when the noise scale is substantially larger than the radius of the point cloud along each direction—the cross term will dominate. When this term dominates, to leading order we have

$$
w_i(\mathbf{x}) \approx \frac{1 + \frac{1}{\sigma^2}(\mathbf{x} - \boldsymbol{\mu})^T(\mathbf{y}_i - \boldsymbol{\mu})}{\sum_j^N \left[1 + \frac{1}{\sigma^2}(\mathbf{x} - \boldsymbol{\mu})^T(\mathbf{y}_j - \boldsymbol{\mu})\right]} \approx \frac{1 + \frac{1}{\sigma^2}(\mathbf{x} - \boldsymbol{\mu})^T(\mathbf{y}_i - \boldsymbol{\mu})}{N} + \mathcal{O}\left(\frac{\mathrm{tr}\boldsymbol{\Sigma}}{\sigma^2}\right) \, .
\tag{25}
$$

Substituting this result back into the point cloud score (Eq. 24), we find that

$$
\mathbf{s}(\mathbf{x}, \sigma) \approx \frac{\boldsymbol{\mu} - \mathbf{x}}{\sigma^2} + \frac{1}{\sigma^2}\frac{1}{N}\sum_i \left[1 + \frac{1}{\sigma^2}(\mathbf{y}_i - \boldsymbol{\mu})^T(\mathbf{x} - \boldsymbol{\mu})\right](\mathbf{y}_i - \boldsymbol{\mu}) = \frac{\boldsymbol{\mu} - \mathbf{x}}{\sigma^2} - \frac{1}{\sigma^4}\boldsymbol{\Sigma}(\boldsymbol{\mu} - \boldsymbol{x}) \, .
\tag{26}
$$

Finally, we observe that the score function of the Gaussian model (see Sec. 2.2) is identical to leading order:

$$
\frac{1}{\sigma^2}(\mathbf{I} + \frac{1}{\sigma^2}\boldsymbol{\Sigma})^{-1}(\boldsymbol{\mu} - \mathbf{x}) \approx \left[\frac{1}{\sigma^2}\mathbf{I} - \frac{1}{\sigma^4}\boldsymbol{\Sigma} + \mathcal{O}\left(\frac{\mathrm{tr}\boldsymbol{\Sigma}}{\sigma^4}\right)\right](\boldsymbol{\mu} - \mathbf{x}) \, .
\tag{27}
$$

This proves the claim. As a parenthetical comment, we note that our expansion strategy is similar in spirit to the multipole expansion used in (for example) electrodynamics (Jackson, 2012); in both cases, the key idea is to approximate a vector field by matching certain low-order statistics.

Is the Gaussian approximation *really* only accurate when $\sigma \gg \sqrt{\mathrm{tr}\boldsymbol{\Sigma}}$? In the next subsection, we empirically show that it works quite well even for somewhat lower noise scales—and in particular when the noise scale is smaller than the top eigenvalue of the covariance matrix (Fig. 6). The surprisingly broad range of noise scales for which the Gaussian approximation works well is what led us to note its '*unreasonable effectiveness*'.

### 4.2 Learned score vectors are empirically well-approximated by the Gaussian model

Motivated in part by the theoretical argument from the previous subsection, we empirically validated the claim that the score function of pre-trained diffusion models is well-described by the Gaussian model in the far-field/high-noise regime. To do this, we examined the score functions of three pre-trained models (of CIFAR-10, FFHQ64, and AFHQv2-64) from Karras et al. (2022). For each dataset, we computed the scores of several tractable approximations of the data:

- **Isotropic Gaussian score.** (*Iso*) This score simplifies the Gaussian model by removing covariance information, and has $\mathbf{s}(\mathbf{x}, \sigma) := (\boldsymbol{\mu} - \mathbf{x})/\sigma^2$.

- **Gaussian score.** (see Eq. 5) The Gaussian model with the mean and covariance of the dataset.

- **Exact point cloud score.** (*Exact delta*, see Eq. 10) The score of the training dataset.

- **Per-class Gaussian mixture score.** (*GMM*; see Eq. 7) For datasets with class labels, (e.g. CIFAR-10), we computed the score of a GMM with one Gaussian mode corresponding to each class, equipped with the mean and covariance of that class. See Appendix A.2 for more details.

For various noise scales $\sigma$, we evaluated the neural and analytical scores at random points sampled from $\mathcal{N}(\mathbf{0}, \sigma^2 \mathbf{I})$, and quantified their difference using the *fraction of unexplained variance*, defined via

$$\text{fraction of unexplained variance} := \frac{\|\mathbf{s}_{EDM}(\mathbf{x}, \sigma) - \mathbf{s}_{analy}(\mathbf{x}, \sigma)\|_2^2}{\|\mathbf{s}_{EDM}(\mathbf{x}, \sigma)\|_2^2} . \tag{28}$$

We characterized the average deviation as a function of noise scale for each type of idealized score (Fig. 6).

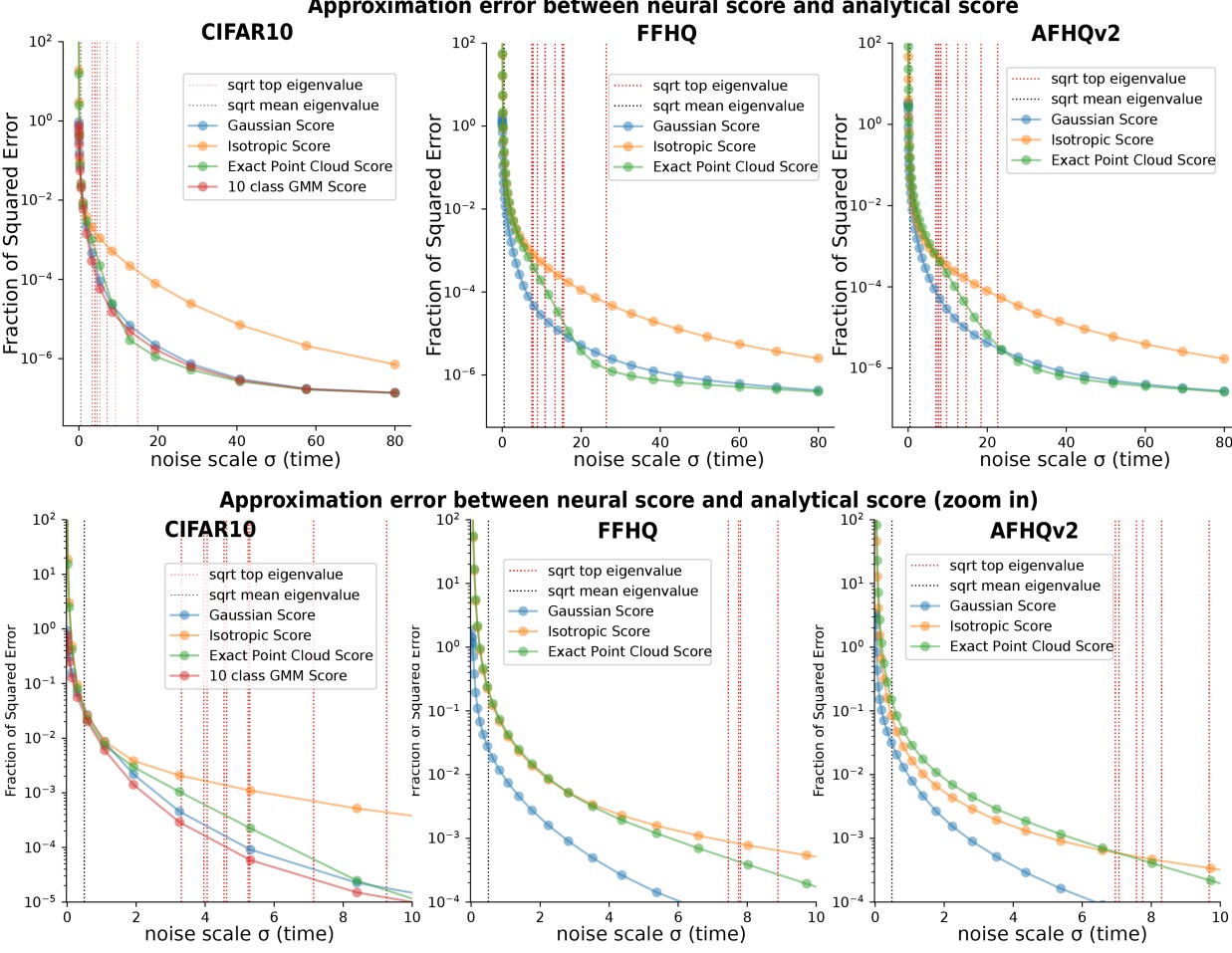

Figure 6: **Gaussian score dominates the learned neural score at high noise scales**. Fraction of squared error as a function of noise scale between neural score function and various analytical score approximations. Note the log scale on the y-axis. **Upper panel**, all noise levels. **Lower panel**, zoom in to low-noise regime. For reference, vertical lines show the square roots of the 10 largest covariance eigenvalues.

**Gaussian score predicts the learned score at high noise.** For all three datasets, at most noise levels ($\sigma > 1.08$), the Gaussian score explains almost all variance ($> 99\%$) of the neural score. As expected, it explains more variance than the isotropic model, which does not incorporate covariance information. Further, for CIFAR-10, at all levels, the 10-class Gaussian mixture score predicts the neural score slightly better than the Gaussian score, which shows that adding more modes indeed helps capture the details of the score field. Surprisingly, at most noise levels ($\sigma > 1.08$) the 10-mode model improved the fraction of explained variance

by less than $2.5 \times 10^{-3}$, showing that even a single Gaussian may be sufficient for explaining the learned neural score. To our surprise and against the intuition suggested by our earlier theoretical argument (see the red dashed line in Fig. 6), the Gaussian model well-approximates the neural score even when $\sigma < \sqrt{\lambda_{max}}$.

**Neural score deviates from 'exact delta' score at low noise.** Although the exact point cloud score slightly outperforms the Gaussian model in the high-noise regime, it deviates substantially from the learned score in the low-noise regime (Fig. 6), and in fact performs worse than the Gaussian and Gaussian mixture models for all three datasets. This suggests that, especially in the low noise regime, the models learned something substantially different from the 'exact' score, and more similar to the Gaussian or Gaussian mixture scores. As previously argued, deviations from the 'exact' score model can be viewed as a signature of generalization (Kadkhodaie et al., 2023; Yi et al., 2023). In Sec. 4.4, we study the effect of adding more modes to a Gaussian mixture approximation more comprehensively.

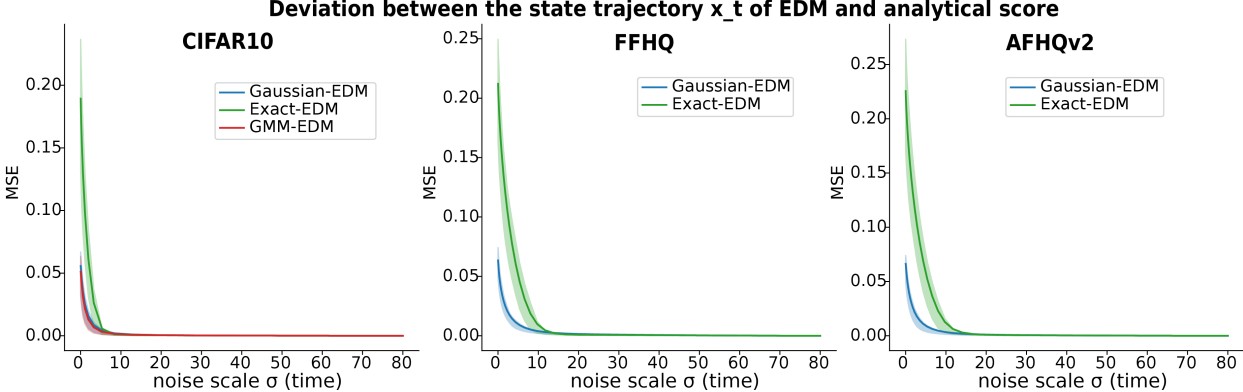

Figure 7: **Gaussian model solution predicts the early sampling trajectory of diffusion models**. Deviation between the $\mathbf{x}_t$ trajectory of the EDM neural score and the analytical score is plotted over time. Many initial conditions were used; the thick line denotes the ensemble average, and the shaded area denotes the 25%-75% quantile range.

### 4.3 Gaussian model empirically captures early sample generation dynamics

If the Gaussian model produces score vectors similar to those learned by neural networks, then their sampling trajectories may (at least initially) be similar too. On the other hand, it is also possible that initially small differences may accumulate over the course of sample generation, and hence that real sampling trajectories differ substantially from those predicted by the Gaussian model[4]. To test this, we compared the solutions of PF-ODE based on the idealized score models to the sampling trajectories of neural diffusion models with deterministic samplers. We used the exact solution for the Gaussian model (Eq. 15), Heun's 2nd-order method to integrate the neural scores, and an off-the-shelf RK4 integrator to integrate the mixture scores.

**Gaussian model predicts early diffusion trajectories.** We found that the early phase of reverse diffusion is well-predicted by the Gaussian analytical solution (**blue trace** in Fig. 7, 16), whose trajectory remained close to those of the trained diffusion models (MSE < 0.01) before diverging. For CIFAR-10, the point of divergence was around 9 sampling steps ($\sigma \approx 1.92$), and for FFHQ and AFHQ it was around 17 steps ($\sigma \approx 4.37$). This roughly corresponds to the scale where the Gaussian approximation of the score field breaks down, and the score needs to be approximated by that of a more complicated distribution (Fig. 6).

The exact score model predicts early sample generation dynamics slightly better than the Gaussian model (**green trace** in Fig. 7, 16), but diverges earlier ($\sigma < 15$) from the neural trajectory, and by a much larger amount than the Gaussian model, consistent with our earlier score function observations (Fig. 6).

---

[4]Theory suggests there is a limit to how much score errors can produce sample generation errors. In particular, Girsanov's theorem implies that such errors cannot compound exponentially (**?**).

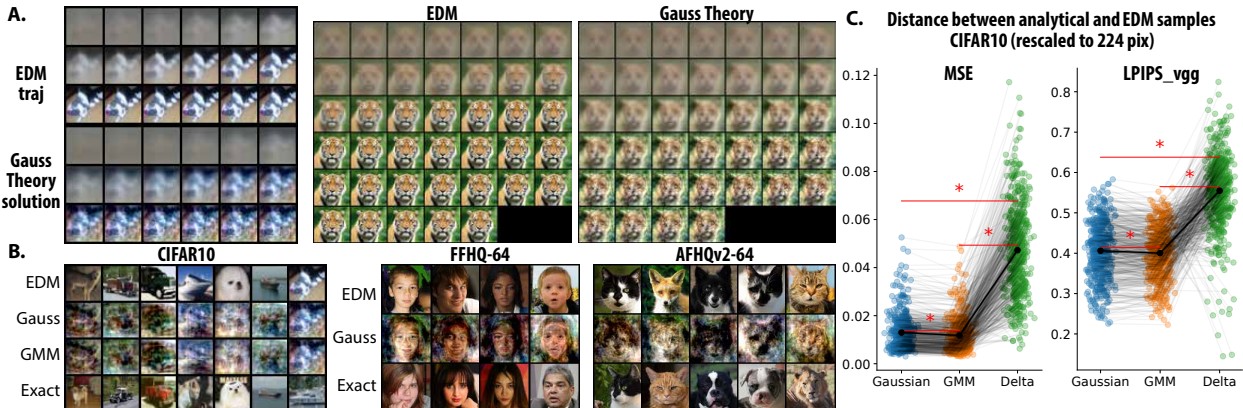

Figure 8: **Gaussian model predicts early denoiser trajectories and low-frequency features of samples**. **A.** The denoiser output $\mathbf{D}(\mathbf{x}_t, t)$ along a sampling trajectory of the EDM model and the Gaussian solution with the same initial condition $\mathbf{x}_T$. **B.** Samples generated by the EDM model, Gaussian solution, and the 'exact' delta mixture scores from the same initial condition. **C.** Image samples from the Gaussian and GMM models are closer to actual diffusion samples than samples from the delta score model.

**Gaussian model predicts low-frequency aspects of diffusion samples.** We visualized the samples generated by the neural and the idealized score models (Fig. 8 A,B), and found that the Gaussian model's samples closely replicated several characteristics of those produced by trained models, including their global color palette, background, the spatial layout of objects, and face shading, among other features.

This observation is consistent with our earlier theoretical results given well-known facts about natural image statistics. These characteristics represent relatively low-spatial-frequency information, which contains far more variance than high-frequency information (Ruderman, 1994). As predicted by our Gaussian model results (Fig. 3C), and empirically shown by Ho et al. (2020), high-variance image features are determined in the early phase of sampling, during which the neural trajectory is accurately described by the Gaussian model. Consequently, the Gaussian model can predict the 'layout' of the final image; given that it does not *fully* describe learned neural scores, especially in the low-noise regime, it is unsurprising that it is less successful at predicting high-frequency details like edges and textures.

**Gaussian model predicts diffusion samples more accurately than exact delta score.** Interestingly, the samples generated by the exact delta score visually deviate even more from the EDM samples than those generated by the Gaussian model (Fig. 8 B). We quantified this observation using the pixelwise mean squared error (MSE) and Perceptual Similarity (LPIPS) metrics, with further details provided in Appendix A.3. Across all metrics, samples from pre-trained diffusion models more closely resembled Gaussian-generated samples than samples produced using the exact delta score (Fig. 8 C). This discrepancy was significant, albeit less pronounced, when using the perceptual similarity metric, except in the case of the AlexNet-based LPIPS distance on the AFHQ dataset, where the difference was not significant (Fig. 19).

These findings indicate, perhaps contrary to expectation, that the Gaussian model in many ways provides a better *quantitative* approximation of the behavior of real diffusion models than the score of the training set. This is true in various senses, and appears to be true whether one uses pixel space or perceptual metrics.

### 4.4 Beyond Gaussian: Low-rank Gaussian mixture scores as a model of learned neural scores

Our previous analyses demonstrated that learned score functions closely match the Gaussian model at high noise levels, but not at low noise levels. To study the behavior of the learned score function at lower noise—or, equivalently—closer to the data manifold, we need to go beyond the single Gaussian. As a next step, we studied the score of a Gaussian mixture model whose covariance was allowed to be low-rank. We asked 1) to what extent does adding Gaussian components help explain the learned neural score? and 2) what is the minimum rank required for good score approximation?

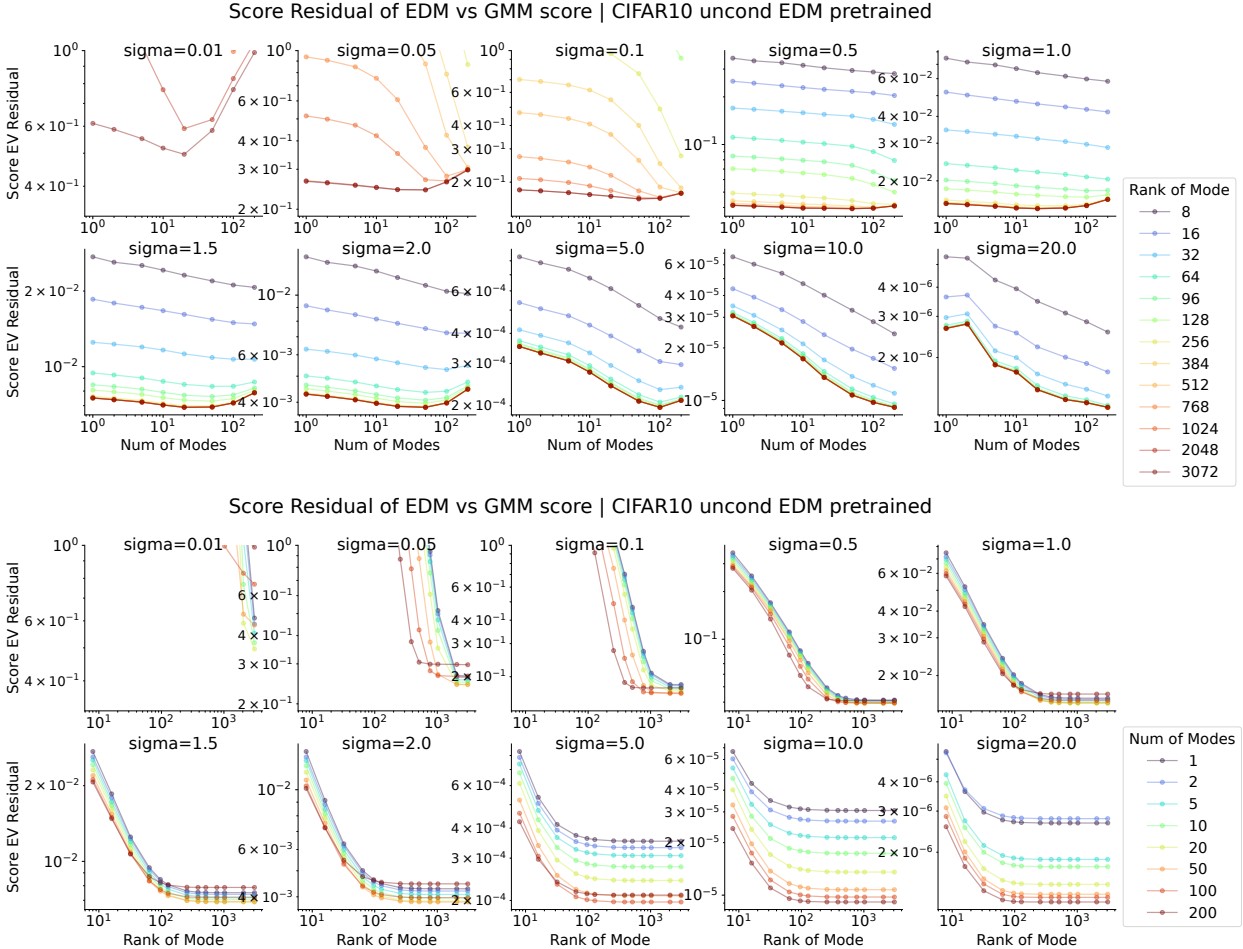

Figure 9: **Deviation of learned neural score from Gaussian mixture model with varying components and ranks**. **Upper:** Residual explained variance (EV) plotted as a function of the number of Gaussian modes on the x-axis, with each colored line representing a different rank. Each panel compares the score at a certain noise scale $\sigma$. **Lower:** An alternative view of the same data plot, with the rank of the covariance matrix on the x-axis. See Fig. 23 for analogous MNIST results.

We systematically compared the structure of neural scores to the scores of GMMs fit to the same training data (Fig. 9). We varied both the number of modes and the ranks of the associated covariance matrices; for the details of the fitting procedure, see Appendix A.5. In brief, we performed k-means clustering on the training set and utilized the empirical mean and covariance of each cluster to define Gaussian components. The fraction of samples within each cluster was used to define the weights of components. To obtain low-rank covariance matrices, we computed the eigendecomposition of the empirical covariances and retained the top $r$ directions.

**Neural score is best explained by Gaussian mixture with moderate number of modes.** First, note that both the Gaussian model and 'exact' score models are special cases of the GMM; the former has $K = 1$ and the latter has $K = N$. This leads us to expect that making $K$ larger than 1 helps approximate the learned neural score, but *only up to a point*.

Indeed, we found that at most noise scales ($\sigma > 0.05$), increasing the number of Gaussian modes reduces the deviation between the neural score and the analytical score. Further, measured by the increase of explained variance, the benefit of adding Gaussian modes increases as we lower the noise level: it rises from $10^{-8}$ to $10^{-2}$ (Fig. 10A). But this trend does not hold indefinitely. At smaller noise scales, augmenting the Gaussian modes beyond a certain number—for instance, 200-500 for MNIST and 100 for CIFAR-10—increases the gap

between the neural and analytical scores, as illustrated by the U-shaped curves in the upper panel of Fig. 9. This divergence highlights that the neural network learned a *coarse-grained approximation of the score field.*

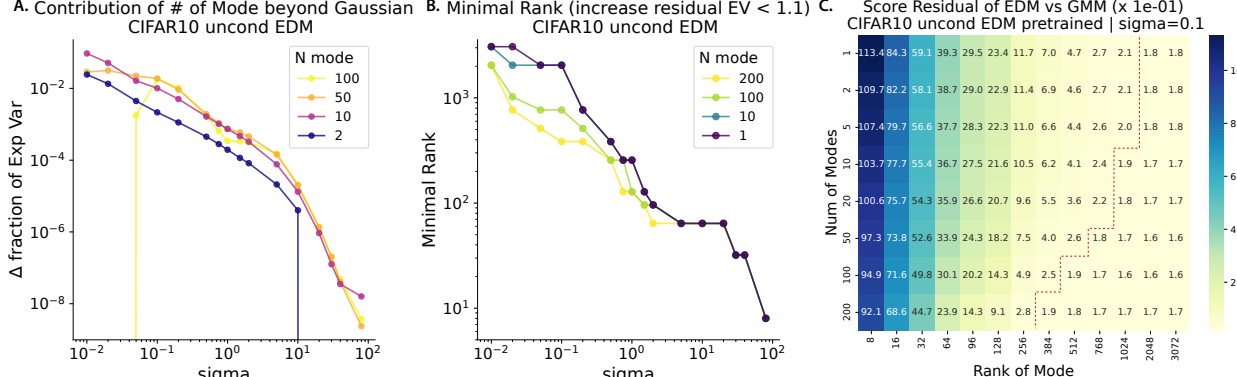

Figure 10: **Impact of number of modes and covariance rank on neural score approximation**. **A.** Contribution of multiple mixture components beyond one Gaussian (full rank). Improvement of the fraction of Exp. Var. is plotted on the y-axis; the vertical line means certain mixture models perform even worse than the single Gaussian. **B.** Minimal rank required for each noise level. Each line shows the minimum rank for different numbers of GMM components. Minimal rank is the smallest rank value that increases the residual Exp. Var. within 10% beyond the full rank model. **C.** Tradeoff between Gaussian rank and mode number. Residual Exp. Var. plotted as a function of mode number and rank, with the red dashed line denoting models with same explained variance.

**Required covariance rank increases with lower noise level.** We found that, in the high-noise regime, a low-rank Gaussian is sufficient to predict the score: for example, when $\sigma = 10$, the explained variance is identical for Gaussian mixtures with full rank and rank 100 (Fig. 9 lower panel; note that the curve has an 'elbow'). As the noise scale decreases, the minimum required rank gradually increases: the elbow of the explained variance curve moves rightward (Fig. 10 B). This phenomenon can be understood through the Gaussian score equation (Eq. 12). In the diagonal matrix $\tilde{\Lambda}_\sigma$, entries for which $\lambda_k \ll \sigma^2$ are rendered negligible, as $\frac{\lambda_k}{\lambda_k + \sigma^2} \approx 0$, i.e., those dimensions become effectively 'invisible' at that noise scale. As the noise scale is reduced, more principal dimensions of the covariance matrix are unveiled and become 'visible' to the score function, so the covariance matrix's rank effectively increases.

**Trade-off between mode number and local rank.** Interestingly, at small noise scales, one can trade off between the number of components and the rank of each component. For the CIFAR-10 dataset, at $\sigma = 0.1$, a GMM with 200 components with rank 384 is as effective at explaining the neural score as a GMM with 10 components with rank 1024 (Fig. 10C, see also horizontal lines in Fig. 9). This suggests an interesting geometric picture of the score. Though globally the data distribution and its score appear to be high rank, locally the data distribution and the score are effectively low rank; this picture is reminiscent of a nonlinear manifold comprised of many glued-together linear manifolds.

**Visual comparison of Gaussian mixture score versus neural score in low noise regime.** The deviation between Gaussian mixture scores and neural scores in the low noise regime raises the question about the spatial nature of this difference. To explore this, we visualized various neural and idealized score fields close to the image manifold. We selected three data samples to establish a 2D plane, and then evaluated and projected each score vector field onto this plane. Various qualitative comparisons of this sort (involving both inter- and intra-category planes, and a mix of training and test examples) are illustrated in Figures 11, 20, 21, and 22.

At a moderate noise level (e.g., $\sigma = 2.0$), the vector field created by the delta point cloud or a large number of Gaussian mixtures could roughly recapitulate the score field structure of the neural score (Fig. 11 upper panel). Intriguingly, at lower noise levels ($\sigma \sim 0.1$), the neural score revealed a complex geometric structure, with the score vector field nearly vanishing inside the lines and triangles (simplexes) interpolating the data

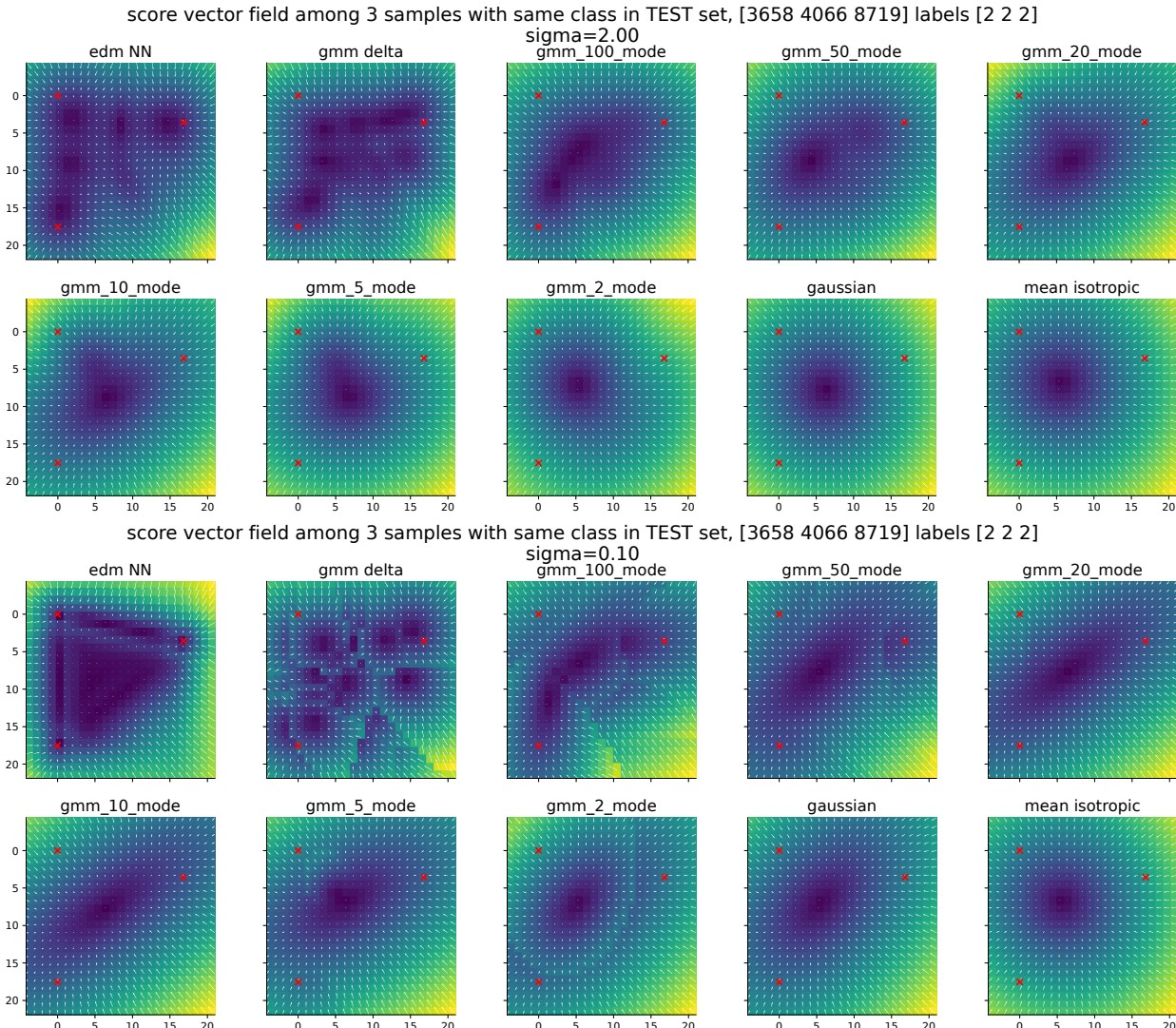

Figure 11: **Visual comparison of the score vector field** $s_\theta(\mathbf{x}, \sigma)$ **of pre-trained diffusion models and the Gaussian mixture model**. All panels visualize score functions on the same 2D domain, namely the plane spanned by three test set examples not seen during training. Red dots mark the 2D coordinates of these three samples. The x-y axes correspond to an orthonormal coordinate system on the plane, with the units denoting L2 distance on the plane. In this case, all three examples are of the MNIST digit '2'. Arrows visualize the projection of score vectors onto this plane, and the heatmap visualizes the L2 norm of the projected vector on the plane. The first panel shows the neural score vector after training; the other panels show idealized score models of decreasing complexity (number of Gaussian modes). See Fig. 20, 21, and 22 for additional similar visualizations.

points, as depicted in the lower panel of Fig. 11. This contrasts with the Gaussian mixture and delta mixture scores, which aligned with theoretical expectations of piece-wise linear vector fields demarcated by linear or quadratic hypersurfaces.

This phenomenon is reminiscent of continuous attractors in the dynamical systems literature (Samsonovich & McNaughton, 1997; Khona & Fiete, 2022), and suggests that the probability flow ODE could converge along these simplices where the vector field vanishes, which may be one mechanism that supports generalization.

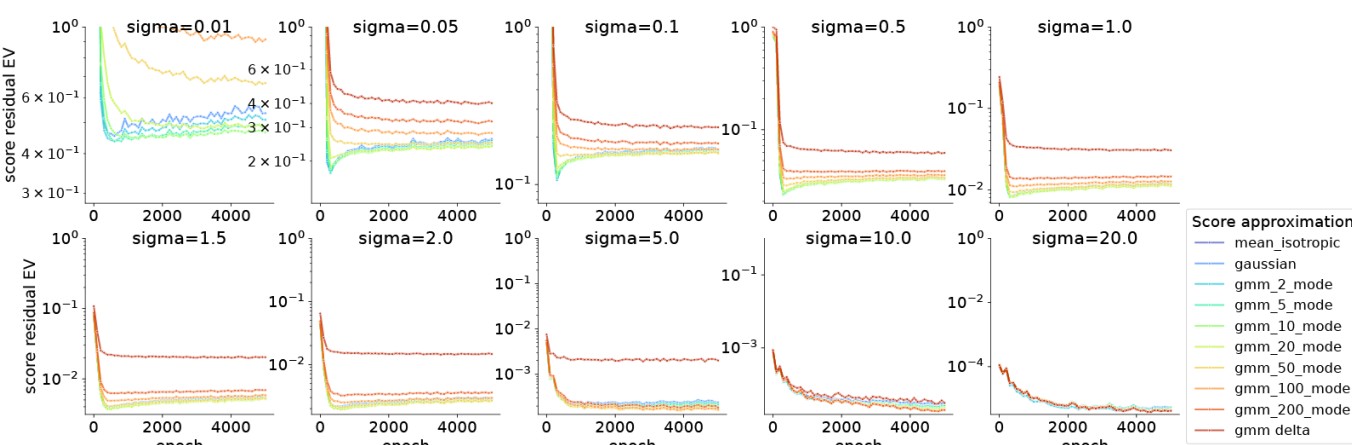

Figure 12: **Learning dynamics of score neural network $s_\theta(\mathbf{x}, \sigma)$ with idealized scores (Gaussian, Gaussian mixture, and Exact) as reference**. Each panel shows a different noise scale $\sigma$ and plots residual explained variance (1 - EV) as a function of training epoch. Each line shows the residual EV of the neural score with respect to one idealized score model. Consistent with Fig. 6, at higher noise scales, all score approximators deviate negligibly from the neural score and from each other. At lower noise scales ($\sigma \leq 1.5$), the deviation between the neural score and the Gaussian score first decreases, and then increases slightly. This indicates that the neural score approaches the score of the Gaussian model before it learns additional structure. Learning dynamics for other datasets and training settings are shown in Fig. 25 and 26.

## 5 Learning of far-field Gaussian score structure

In the preceding sections (Sec. 4.2, 4.3, 4.4), our analyses focused on the structure of the score of trained diffusion models, but did not touch on the question of how that structure emerges during training. Given that overparameterized function approximators like neural networks are thought to learn 'simpler' structure first, it stands to reason that Gaussian/linear score structure may be learned relatively early. Is this true?

To test this idea, we trained neural network score approximators on different datasets using the training procedure described by Karras et al. (2022). Throughout training, we sampled query points $\mathbf{x}_t$ from the noised distribution, and compared the neural score to the three idealized score models from Sec. 2.2.

As we have observed, at higher noise scales ($\sigma \geq 10$), all score approximators are similar to the neural score after convergence. During training, the neural score function steadily converges to these approximators as well (Fig. 12, Fig. 25). Intriguingly, at lower noise scales, the network displays non-monotonic learning dynamics for simpler scores. Specifically, the neural score initially aligns with simpler score models (the Gaussian model and a GMM with few modes) before starting to deviate from them. This pattern was consistently observed across various training sets (refer to Fig. 25 for CIFAR-10, AFHQ, FFHQ). On the other hand, the neural score approached more complex score approximators (the delta mixture) monotonically. This effect suggests an initial tendency to fit the score of simpler distributions.

This tendency manifested slightly differently on the MNIST dataset (Fig. 26). Here, the score network approaches different GMM score approximators in order of increasing complexity. Namely, the deviation between the neural score and simpler Gaussian scores decreased and reached a floor, before moving on to more complex models, such as 2-mode and 5-mode Gaussian mixtures. Although the non-monotonic effect was less pronounced, it supports the idea that the network initially maximizes the explainable variance for simpler distributions before progressing to more complex Gaussian mixtures.

We visually demonstrate this phenomenon by plotting the evolution of the neural score field during training on the MNIST dataset (Fig. 13). At a lower noise scale ($\sigma = 1.0$), the neural score field starts by resembling a Gaussian-like (single basin) vector field, then splits into multiple basins, so that it resembles the score of a multi-modal distribution.

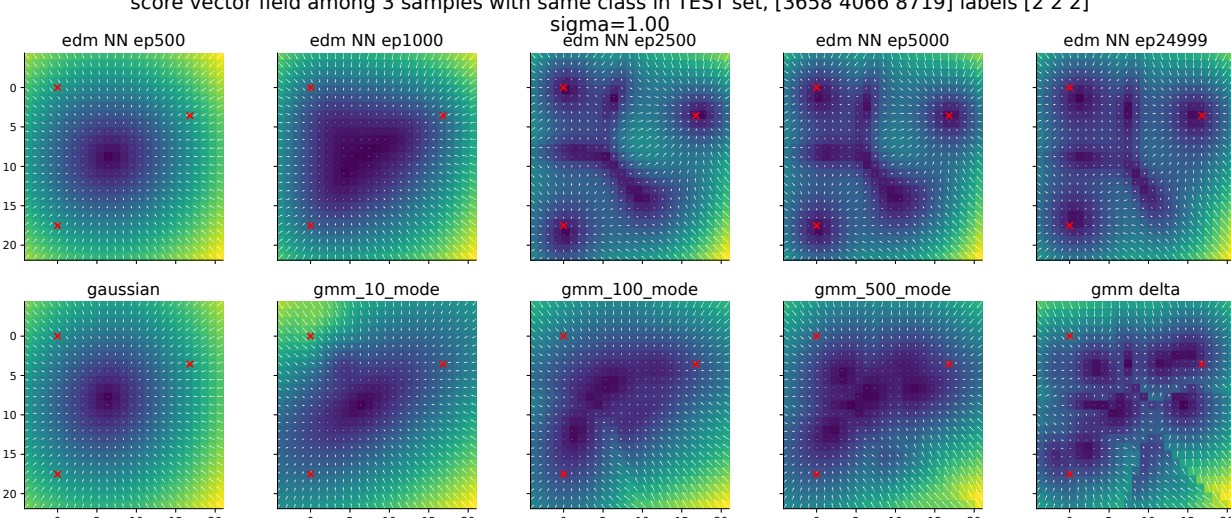

Figure 13: **Visual comparison of neural score vector field** $\mathbf{s}_\theta(\mathbf{x}, \sigma)$ **with idealized scores throughout training**. Layout and score domain are the same as in Fig. 11. The upper row panels depict the neural score vector field at different training epochs; the lower row panels depict the scores of idealized models of increasing complexity. See Fig. 27, 28, 29 for similar plots on the plane spanned by training examples.

Our finding is consistent with the general observation that the learning dynamics of overparameterized neural networks exhibit a 'spectral bias' (Bordelon et al., 2020; Canatar et al., 2021), and tend to capture low-frequency aspects of input-output mappings before high-frequency ones (Rahaman et al., 2019; Xu et al., 2022). In this setting, the relevant mapping is the one defined by the score $\mathbf{s}(\mathbf{x}, \sigma)$ or denoiser $\mathbf{D}(\mathbf{x}, \sigma)$. For lower noise scales $\sigma$, the Gaussian score is smoother and lower-frequency in $\mathbf{x}$ space than the score of GMMs, so it is perhaps unsurprising that Gaussian structure is learned preferentially by the network early in training.

## 6 Application: Accelerating sampling via teleportation

Above, we have shown that the Gaussian model's exact solution provides a surprisingly good approximation to the early sampling trajectory. We can exploit this fact to accelerate diffusion by 'analytical teleportation' (Fig. 14 A). By this, we mean replacing a certain number of initial PF-ODE integration steps with a single evaluation of the Gaussian analytical solution at some intermediate time $t'$ (or equivalently noise scale $\sigma_{skip}$). In this way, one can nontrivially reduce the number of neural function evaluations (NFEs) required to generate a sample. In principle, this speedup can be combined with any deterministic or stochastic sampler.

**Experiment 1.** First, we showcase its effectiveness using the optimized second-order Heun sampler from Karras et al. (2022), which yields near state-of-the-art image quality and efficiency. See Alg. 1 and Appendix A.7, B.2 for method details and additional experiments with DDIM (Song et al., 2020a).

We evaluated our proposed hybrid sampler on unconditional diffusion models trained on CIFAR-10, FFHQ-64, and AFHQv2-64. We sampled 50,000 images using both the Heun sampler and our hybrid sampler with various numbers of skipped steps (or equivalently, different times $t'$), and evaluated the Frechet Inception Distance (FID) in each case. We found that we can consistently save 15-30% of NFES at the cost of a less than 3% increase in the FID score (Fig. 14 B,C), even when competing with the optimized sampling method from EDM (Karras et al., 2022).

For CIFAR-10 and AFHQ, skipping steps can even *reduce* the FID score by 1%, resulting in a highly competitive FID score of 1.93 for CIFAR-10 unconditional generation. (For the full set of results, see Sec. B.8, Fig. 30, and Tab. 4-6.) Comparing Fig. 30 to Fig. 16, we observe that the number of skippable steps roughly corresponds to the noise scale at which the neural score deviates substantially from its Gaussian approximation.

---

**Algorithm 1:** Hybrid sampling using Heun's method and Gaussian model prediction

---

**Require:** Data: mean and covariance of dataset $\boldsymbol{\mu}$, $\boldsymbol{\Sigma}$ and its eigendecomposition $\boldsymbol{\Sigma} = \mathbf{U}\boldsymbol{\Lambda}\mathbf{U}^T$
**Require:** Original Heun's sampler parameters: $\sigma_{min}$, $\sigma_{max}$, $\rho$
**Require:** Parameter: Skip time/noise scale $t' = \sigma_{skip}$
**Input** : $\mathbf{x}_T$

Compute $\mathbf{x}_{t'}$ as the Gaussian solution at $t'$ using $\boldsymbol{\mu}$ and $\boldsymbol{\Sigma}$:;

$$\mathbf{x}_{t'} \leftarrow \boldsymbol{\mu} + \frac{\sigma_{skip}}{\sigma_{max}}(\mathbf{I} - \mathbf{U}\mathbf{U}^T)(\mathbf{x}_T - \boldsymbol{\mu}) + \sum_{k=1}^{r} \sqrt{\frac{\sigma_{skip}^2 + \lambda_k}{\sigma_{max}^2 + \lambda_k}} \mathbf{u}_k \mathbf{u}_k^T (\mathbf{x}_T - \boldsymbol{\mu});$$

Begin numerical integration using Heun's method;
    Use initial time $t' = \sigma_{skip}$ and initial state $\mathbf{x}_{t'}$;
    Integrate until $\sigma_{min}$ to obtain the final sample $\mathbf{x}_0$;

**Return** : $\mathbf{x}_0$

---

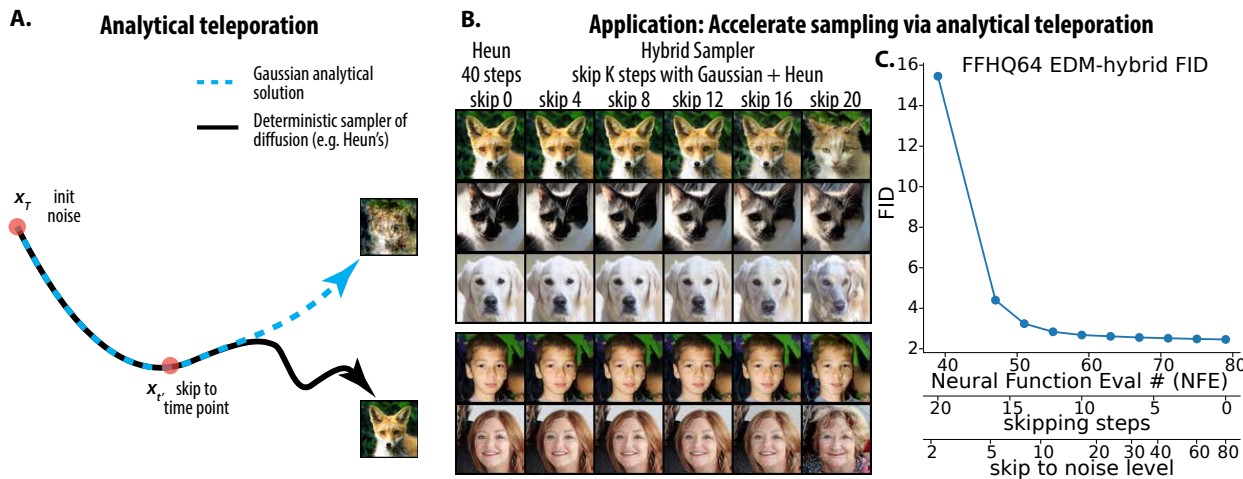

Figure 14: **Leveraging closed-form solution to accelerate sample generation**. **A.** Schematic description of our 'analytical teleportation' method. **B.** Sampled image as a function of number of skipped steps; our hybrid method combines Heun's method with a Gaussian model prediction. **C.** Image quality (FID score) of the hybrid method as a function of NFE and number of skipped steps (see Appendix A.8).

Table 1: **Teleportation improves sampling speed while maintaining FID scores across datasets**. Noise in the table refers to the noise scale that analytical teleportation skips to, i.e., $t' = \sigma_{skip}$ in Alg. 1. See Fig. 30 and Tab. 4-6 for a more complete set of results.

| | FFHQ-64 | | | AFHQ-64 | | | CIFAR-10 | | |
|---|---|---|---|---|---|---|---|---|---|
| | NFE↓ | Noise | FID↓ | NFE↓ | Noise | FID↓ | NFE↓ | Noise | FID↓ |
| EDM baseline | 79 | 80.0 | 2.464 | 79 | 80.0 | 2.043 | 35 | 80.0 | 1.958 |
| *Teleportation* | 67 | 32.7 | 2.561 | 59 | 16.8 | 2.026 | 25 | 12.9 | 1.934 |
| | 55 | 11.7 | 2.841 | 51 | 8.0 | 2.359 | 21 | 5.3 | 2.123 |

**Experiment 2.** To further demonstrate the generality of our method, we evaluated our hybrid sampler with several popular deterministic samplers: `dpm_solver++`, `dpm_solver_v3`, `heun`, `uni_pc_bh1`, `uni_pc_bh2` Lu et al. (2022); Zheng et al. (2023); Zhao et al. (2024) on the same pre-trained EDM models. We systematically varied the skip noise scale $\sigma_{skip}$ and the number of sampling steps $n_{step}$.

We found results consistent with the first experiment across all samplers (Fig. 15): Gaussian teleportation can improve sample generation time (i.e., reduce the number of required neural function evaluations) without

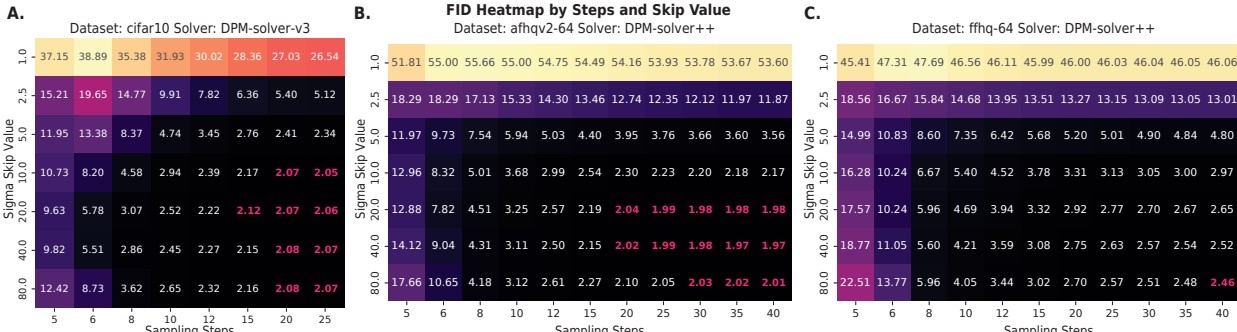

Figure 15: **Image quality as a function of skip noise scale and sampling steps number with DPM-Solver**. for **A.** CIFAR10. **B.** AFHQv2. **C.** FFHQ dataset. FID scores lower than or within 1% increase of the original sampler (without teleportation) are colored in magenta. For full evaluations, see Fig. 31-33.

reducing sample quality. Further, given a fixed budget of sampling steps, using Gaussian teleportation can improve sample quality (i.e., reduce FID score).

Our method compares favorably to the `dpm_solver_v3` approach to speeding up sampling proposed by Zheng et al. (2023), which exploits model-specific statistics. They achieved FIDs of 12.21 (5 NFE) and 2.51 (10 NFE) on unconditional CIFAR-10. Using our teleportation technique, we achieved FIDs of 9.63 (5 NFE, $\sigma_{skip} = 20.0$), 2.45 (10 NFE, $\sigma_{skip} = 40.0$), and 2.22 (12 NFE, $\sigma_{skip} = 20.0$). Similar results hold for the AFHQv2 dataset (see full results in Fig. 31-32). Our intuition is that substituting the easy-to-approximate (i.e., mostly linear) part of the score field with its Gaussian approximation allows the sampler to spend more steps on the harder-to-approximate, nonlinear score field close to the data manifold. If this intuition is correct, it suggests that by spending the neural function evaluation budget more wisely, we can improve sample quality.

For the FFHQ-64 dataset, we observed similar results to Experiment 1: analytical teleportation slightly increased FID given a fixed number of sampling steps (see full results in Fig. 33). For example using `dpm_solver++`, 40 NFE with no skipping yields an FID of 2.46, while skipping to noise scale $\sigma_{skip} = 20.0$ yields FIDs of 2.65 (40 NFE) and 2.77 (25 NFE). Though these changes in FID are tiny, they show that for the face dataset, the neural score model may deviate from the Gaussian approximation in an interesting way in the high noise regime.

# 7    Related work

**Diffusion models and Gaussian mixtures.**    There has been increasing interest in characterizing diffusion models associated with tractable target distributions, and specifically in Gaussian and Gaussian mixture models.  Shah et al. (2023) and Gatmiry et al. (2024) focused on learning to generate samples from Gaussian mixtures using diffusion models, and Shah et al. (2023) found a connection between gradient-based score matching and the expectation-maximization (EM) algorithm and derived convergence guarantees. Concurrently, Pierret & Galerne (2024) derived the exact solution to the reverse SDE and PF-ODE for a Gaussian model, which allowed them to compare Wasserstein errors for any sampling scheme. But to our knowledge, no existing work has provided the in-depth comparisons between idealized score models and neural scores that we present here.

**Consistency of noise-to-image mapping.**    Recently, many researchers have noticed that the mapping from initial noise to images is highly consistent across independently-trained diffusion models, and even across models trained on non-overlapping splits of the same dataset (Zhang et al., 2023; Kadkhodaie et al., 2023). This phenomenon can be partially explained by two of our findings: (1) the noise-to-image mapping is largely determined by the Gaussian structure of the data (Eq. 16), and (2) Gaussian structure seems to be preferentially learned by neural networks (Sec. 5). If different splits of the dataset have almost identical

Gaussian approximations, one expects different networks to learn similar linear score structure, and hence possibly similar noise-to-image mappings.

**Non-isotropic Gaussian initial states.** Though most diffusion models sample initial states from an isotropic Gaussian, Lee et al. (2021) explored sampling from a non-isotropic Gaussian whose mean and covariance depend on the training set. Although our work does not involve any modification to the score matching objective, we also find that one can save computation by choosing a different initial state, i.e., the state $\mathbf{x}_t \sim \mathcal{N}(\alpha_t \boldsymbol{\mu}, \alpha_t^2 \boldsymbol{\Sigma} + \sigma_t^2 \mathbf{I})$ predicted by the Gaussian model. In a sense, during the intial phase of sample generation, diffusion models convert their isotropic/white initial states to preconditioned non-istropic states.

**Score field smoothness and generalization.** Several recent papers have also observed that trained diffusion models learn score fields that are smoother than the scores of their training distributions, which in principle is helpful for generalization (Kadkhodaie et al., 2023; Scarvelis et al., 2023). These findings are consistent with our observation that diffusion models learn smooth score functions that more closely resemble the scores of Gaussian mixture models with a moderate number of modes.

## 8 Discussion

In summary, even for real diffusion models trained on natural images, in the high-noise regime the neural score is well-approximated by a Gaussian model; in the low-noise regime, Gaussian mixture models approximate neural scores better than the score of the training distribution. We mathematically characterized the sampling trajectories of the Gaussian model, and found that it recapitulates various aspects of real sampling trajectories. Finally, we leveraged these insights to accelerate the initial phase of sampling from diffusion models. Below, we mention additional implications of our results for the training and design of diffusion models:

**Noise schedule.** As the early time evolution of the sample distribution is well-predicted by the Gaussian model, we in principle do not need to sample high noise levels to train a denoiser $\mathbf{D}(\mathbf{x}, \sigma)$. Instead, sampling can directly begin (or 'warm start') from the non-isotropic Gaussian distribution $\mathcal{N}(\alpha_t \boldsymbol{\mu}, \alpha_t^2 \boldsymbol{\Sigma} + \sigma_t^2 \mathbf{I})$.

**Model design.** Since it is empirically true that neural scores are dominated by Gaussian/linear structure at high noise levels, we can directly build this structure into the neural network to assist learning. For example, we can add a linear by-pass pathway based on the covariance of training distribution, and let the neural network only learn the nonlinear residual not accounted for by the Gaussian model term.

**Training distribution.** As the neural networks first need to learn the data covariance structure by score matching, we may be able to assist learning by reshaping the training distribution. One hypothesis is that if we pre-condition the target distribution by whitening its spectrum, then the neural score may converge faster. If true, this may explain the higher efficiency of latent diffusion models (Rombach et al., 2022): with KL regularization, the autoencoder not only compresses the state space, but also 'whitens' the image distribution by morphing it to be closer to a Gaussian distribution.

## 9 Limitations and future work

**Higher-resolution and conditional diffusion models.** Our experiments focused on lower-resolution image generative models. Generalizing our results to higher-resolution models, or popular text-to-image conditional diffusion models (Rombach et al., 2022), involves overcoming difficulties related to covariance estimation. For high-dimensional models, estimating covariances is substantially harder given a limited number of training examples; for conditional models, especially for text-to-image models, there may not even exist training data corresponding to a specific prompt, making direct covariance estimation impossible.

**Structure of neural scores close to image manifold.** We focused on characterizing the linear structure that dominates neural scores in the high noise regime, but did not claim to precisely understand neural scores at smaller noise scales, i.e., when the sample is closer to the data manifold. Even though the scores

of Gaussian and Gaussian mixture models with a moderate number of components better explain neural scores than the delta mixture model, both are far from perfect (Fig. 11), and the precise structure of the neural score may be better described by the score of some nonlinear manifold. We leave the elucidation of such structure, which we expect to be closely related to the generalization capabilities of diffusion models, to future work.

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

## A Detailed Methods

### A.1 Image datasets and Pre-trained Models

Table 2: **Specifications of the image datasets and diffusion models**

| Dataset name | Num Samples | Resolution | Pre-trained Model Spec |
|---|---|---|---|
| CIFAR10 | 50000 | 32 | `edm-cifar10-32x32-uncond-vp` |
| FFHQ64 | 70000 | 64 | `edm-ffhq-64x64-uncond-vp` |
| AFHQv2-64 | 15803 | 64 | `edm-afhqv2-64x64-uncond-vp` |

### A.2 Idealized Score Approximations

In the paper, we compared several analytical approximations of the score. We listed their formula below.

**Isotropic score**, only depends on the mean of data, isotropically pointing towards $\mu$.

$$\frac{\mu - \mathbf{x}}{\sigma^2} \tag{29}$$

This is equivalent to approximating the whole training dataset with its mean.

**Gaussian score**, for a Gaussian distribution of $\mathcal{N}(\mathbf{x}; \mu, \boldsymbol{\Sigma})$, its score is

$$\nabla_{\mathbf{x}} \log \mathcal{N}(\mathbf{x}; \mu, \sigma^2 \mathbf{I} + \boldsymbol{\Sigma}) \tag{30}$$

$$= (\sigma^2 I + \Sigma)^{-1}(\mu - \mathbf{x}) \tag{31}$$

$$= \frac{1}{\sigma^2}(I - U\tilde{\Lambda}_\sigma U^T)(\mu - \mathbf{x}) \tag{32}$$

$$= \frac{\mu - \mathbf{x}}{\sigma^2} - \frac{1}{\sigma^2} U\tilde{\Lambda}_\sigma U^T(\mu - \mathbf{x}) \tag{33}$$

$$\tilde{\Lambda}_\sigma = diag\left[\frac{\lambda_k}{\lambda_k + \sigma^2}\right] \tag{34}$$

it depends on the mean and covariance of the data. We can see it added a non-isotropic correction term to the isotropic score.

**Gaussian mixture score**. For a general Gaussian mixture with $k$ component $q(\mathbf{x}) = \sum_i^k \pi_i \mathcal{N}(\mathbf{x}; \mu_i, \boldsymbol{\Sigma}_i)$, it's score function at noise scale $\sigma$ is the following,

$$\nabla_{\mathbf{x}} \log \left( \sum_i^k \pi_i \mathcal{N}(\mathbf{x}; \mu_i, \sigma^2 \mathbf{I} + \boldsymbol{\Sigma}_i) \right)$$

$$= \sum_i -(\sigma^2 \mathbf{I} + \boldsymbol{\Sigma}_i)^{-1}(\mathbf{x} - \mu_i) \frac{\pi_i \mathcal{N}(\mathbf{x}; \mu_i, \sigma^2 \mathbf{I} + \boldsymbol{\Sigma}_i)}{\sum_j^k \pi_j \mathcal{N}(\mathbf{x}; \mu_j, \sigma^2 \mathbf{I} + \boldsymbol{\Sigma}_j))} \tag{35}$$

$$= \sum_i -(\sigma^2 \mathbf{I} + \boldsymbol{\Sigma}_i)^{-1}(\mathbf{x} - \mu_i) w_i(\mathbf{x}, \sigma) \ .$$

Where the covariance matrix for each Gaussian component can be eigen-decomposed and inverted efficiently. It's a weighted average of the score of each Gaussian mode, with a softmax-like weighting function $w_i(\mathbf{x}, \sigma)$.

**Exact point cloud score**. For a set of data point $\{\mathbf{y}_i\}$, the score of $\mathbf{x}$ at noise scale $\sigma$ is

$$\nabla_{\mathbf{x}} \log \Big( \sum_i^N \frac{1}{N} \mathcal{N}(\mathbf{x}; \mathbf{y}_i, \sigma^2 \mathbf{I}) \Big) \tag{36}$$

$$= \frac{1}{\sigma^2} \Big[ -\mathbf{x} + \sum_i w_i(\mathbf{x}, \sigma) \mathbf{y}_i \Big] \tag{37}$$

$$= \frac{1}{\sigma^2} \Big[ -\mathbf{x} + \sum_i \frac{\exp\big( -\frac{1}{2\sigma^2} \|\mathbf{y}_i - \mathbf{x}\|^2 \big)}{\sum_j \exp\big( -\frac{1}{2\sigma^2} \|\mathbf{y}_j - \mathbf{x}\|^2 \big)} \mathbf{y}_i \Big] \ . \tag{38}$$

$$w_i(\mathbf{x}, \sigma) := \frac{\exp\big( -\frac{1}{2\sigma^2} \|\mathbf{y}_i - \mathbf{x}\|^2 \big)}{\sum_j \exp\big( -\frac{1}{2\sigma^2} \|\mathbf{y}_j - \mathbf{x}\|^2 \big)} = \mathrm{softmax}\Big( -\frac{1}{2\sigma^2} \|\mathbf{y}_i - \mathbf{x}\|^2 \Big) \tag{39}$$

### A.3 Measuring Image Similarity

LPIPS is am image distance metric trained to mimic human perceptual judgment of image similarity. We used LPIPS models with all three pretrained backbones: AlexNet, VGG, SqueezeNet. Note that, the images we were measuring has different resolutions, 32 pixels for CIFAR10 and 64 pixels for FFHQ and AFHQ. Consistent with the FID measurement, for all datasets, we resized the image to 224-pixel resolution before sending them to the LPIPS model. We found that, without resizing, the mismatch of image size (e.g. 32 pixel for CIFAR) to the convnet can bias the image distance by the boundary artifact, and obscure the effect.

### A.4 Sampling Diffusion trajectory with Gaussian Mixture scores

For the Gaussian score, we are able to compute the whole sampling trajectory analytically. But for a Gaussian mixture with more than one mode, we need to use numerical integration. We evaluated the numerical score with the Gaussian mixture model defined as above and integrated Eq.2 with off-the-shelf Runge-Kutta 4 integrator (`solve_ivp` from `scipy`) from sigma 80.0 to 0.0. We also chose $\sigma(t) = t$. To compare the trajectory with the one sampled with the Heun method, we evaluated the trajectory at the same discrete time steps as the Karras et al. (2022) paper. The $i$th noise level is the following,

$$\sigma_i = \Big( \sigma_{max}^{1/\rho} + \frac{i}{n_{step} - 1} (\sigma_{min}^{1/\rho} - \sigma_{max}^{1/\rho}) \Big)^\rho \tag{40}$$

we chose the same hyper parameter $\sigma_{min}, \sigma_{max}, \rho$ and $n_{step}$ as the original EDM paper.

### A.5 Fitting Gaussian Mixture Model and Low rank GMM

Given large and high-dimensional image datasets, we used a fast and heuristic method for fitting the Gaussian Mixture model.

We performed mini-batch k-means clustering on the training data (`sklearn.cluster.MiniBatchKMeans`) with batch size 2048 and fixed random seed 0 to cluster the dataset to $K$ cluster. For each cluster $i$, the number of samples belonging to this class is $N_i$. We compute the empirical mean $\tilde{\mu}_i$ and covariance matrix $\tilde{\Sigma}_i$ of all samples belonging to this cluster. Then we define the Gaussian mixture model as

$$\sum_i^K \frac{N_i}{N} \mathcal{N}(\mathbf{x}; \tilde{\mu}_i, \tilde{\Sigma}_i) \tag{41}$$

This fitting procedure is roughly equivalent to one step of Expectation Maximization (EM) iteration, which is not optimal, but suffice our purpose.

For the low-rank Gaussian mixture models used in Sec. 4.4, we performed PCA of the covariance matrices and kept the top $r$ PC to define the low-rank covariance.

## A.6 Training diffusion models

We used the training configuration F from Table 2 in Karras et al. (2022). We note that *non-leaky augmentation* was used in this training configuration.

Specifically, we used the `train.py` and `train_edm.py` function from the code base `https://github.com/NVlabs/edm` and `https://github.com/yuanzhi-zhu/mini_edm/`.

For mini-EDM training runs, hyperparameters are

- MNIST, used batch size 128, Adam optimizer with a learning rate of 2E-4, we densely sampled checkpoints for the first 25000 steps. The channel multiplier is set to "1 2 3 4", with the base channel count 16, no attention, and there is 1 layer per block.

- CIFAR10, used batch size 128, Adam optimizer with a learning rate of 2E-4, checkpoints for the first 50000 steps. The channel multiplier is set to "1 2 2", with the base channel count 96, attention resolution is 16, and there are 2 layers per block.

For EDM training runs, hyperparameters are

- CIFAR10, used batch size 256, Adam optimizer with a learning rate of 0.001, checkpoints for the first 5000K images (around 20000 steps). EDMPrecond class network was used, the channel multiplier was set to "2 2 2", with the base channel count 128. The augmentation probability was 0.12.

- AFHQ and FFHQ, used batch size 256, Adam optimizer with a learning rate of 2E-4, checkpoints for the first 5000K images (around 20000 steps). EDMPrecond class network was used, the channel multiplier was set to "1 2 2 2", with the base channel count 128. augmentation probability was 0.12.

| Dataset | MNIST | CIFAR | CIFAR | AFHQ | FFHQ |
|---|---|---|---|---|---|
| **Batch** | 128 | 128 | 256 | 256 | 256 |
| **Optim.** | Adam | Adam | Adam | Adam | Adam |
| **LR** | 2E-4 | 2E-4 | 0.001 | 2E-4 | 2E-4 |
| **Steps** | 25000 | 50000 | 20000 | 20000 | 20000 |
| **Chan Mult.** | 1 2 3 4 | 1 2 2 | 2 2 2 | 1 2 2 2 | 1 2 2 2 |
| **Base Chan.** | 16 | 96 | 128 | 128 | 128 |
| **Attn Res.** | None | 16 | 16 | 16 | 16 |
| **Lyr per B** | 1 | 2 | 4 | 4 | 4 |
| **Aug.Prob** | None | None | 0.12 | 0.12 | 0.12 |
| **Net. Class** | SongUNet | SongUNet | SongUNet | SongUNet | SongUNet |
| **Code Base** | mini-EDM | mini-EDM | EDM | EDM | EDM |

Table 3: **Hyperparameters for diffusion model (mini-EDM and EDM) training runs** Abbreviations: Batch = Batch Size, Optim. = Optimizer, LR = Learning Rate, Steps = Steps/Images, Chan Mult. = Channel Multiplier, Base Chan = Base Channel Count, Attn Res = Attention Resolution, Lyr per B = Layers per Block, Augm Prob = Augmentation Probability, Net. Class = Network Class

## A.7 Hybrid sampling method

As we stated in the paper, the Hybrid sampling scheme can be combined with any deterministic sampler (e.g. DDIM Song et al. (2020a), PNDM Liu et al. (2022)). In our two main benchmark experiments, we combined it with the Heun sampler, and a few recent samplers in DPMSolver-v3 benchmark.

**Experiment 1** In the first experiment, we used the following strategy for choosing $\sigma_{skip}$, inspired by the baseline Heun method. The Heun method samples a sequence of $n_{step}$ noise levels $[\sigma_0, \sigma_1, \sigma_2, ...\sigma_{n_{step}}]$, where $\sigma_0 = \sigma_{max}$ and $\sigma_{n_{step}} = \sigma_{min}$. For each initial condition $\mathbf{x}_T$, we use the analytical solution to evaluate $\mathbf{x}_{t'}$ or integrate the probability flow ODE to time $t'$, where we chose $t'$ as the $i$-th noise level $\sigma_i$ (Eq. 40). Then we

will skip the first $i$ step in the Heun method and start at initial state $\mathbf{x}_{t'}$. In this manner, when we skip to the $i$th noise level, we will save $2i$ neural function evaluations.

**Experiment 2** In the second experiment, we systematically varied both the skipping noise level $\sigma_{skip}$ and the number of sampling steps $n_{step}$ for a bunch of recent deterministic diffusion samplers: `dpm_solver++`, `dpm_solver_v3`, `heun`, `uni_pc_bh1`, `uni_pc_bh2`. We used the implementation of these samplers in the code base of DPMSolve-v3, used in their benchmark experiments `https://github.com/thu-ml/DPM-Solver-v3/tree/main/codebases/edm`. We used the same models as in Experiment 1, namely the EDM models pre-trained on CIFAR10, AFHQ, FFHQ datasets.

For each initial condition $\mathbf{x}_T$, we use the analytical solution to evaluate $\mathbf{x}_{t'}$ at $t' = \sigma_{skip}$, and then run these diffusion samplers with setting $\sigma_{max} = \sigma_{skip}$ and $n_{step}$ steps. We systematically vary $n_{step}$ and $\sigma_{skip}$ on a grid.

## A.8 FID score computation and baseline

We used the same code for FID score computation as in Karras et al. (2022). For each sampler, we sampled the same initial noise state $x_T$ with random seeds 0-49999. We computed the FID score based on 50,000 samples.

For our baseline, we picked the same model configurations as reported in Tab. 2, specifically, Variance preserving (VP) config F, the unconditional model for CIFAR10, FFHQ 64 and AFHQv2 64. The default sampling steps are 18 steps ($NFE = 35$) for CIFAR10, 79 steps ($NFE = 79$) for FFHQ and AFHQ.

# B Extended Results

## B.1 Detailed validation results

Here we presented the full results for all datasets: approximating neural scores with analytical scores (Fig.6), deviations between sampling trajectory (Fig.7) and denoiser trajectory (Fig.16) guided by neural score and analytical score.

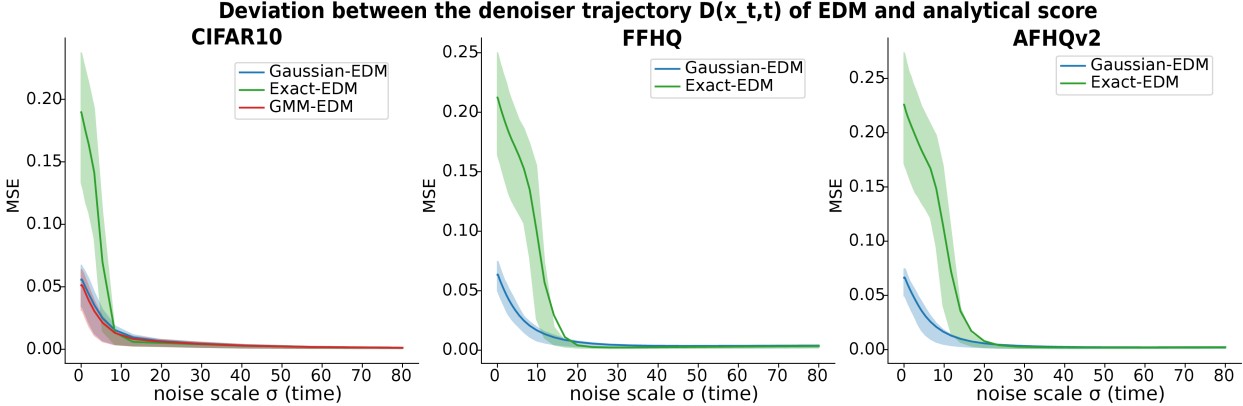

Figure 16: **Deviation between the denoiser $D(\mathbf{x}_t, t)$ of EDM neural network and the analytical score**. The thick line denotes the mean over initial conditions; the shaded area denotes 25%, 75% quantile range over the initial conditions.

## B.2 Additional validation experiments with DDIM

**Teleportation results** For the MNIST and CIFAR-10 models, we can easily skip *40% of the initial steps* with the Gaussian solution without much of a perceptible change in the final sample (Fig.17E). Quantitatively, for CIFAR10 model, we found skipping up to 40% of the initial steps can even slightly decrease the Frechet Inception Distance score (FID), and hence improve the quality of generated samples (Fig.17F). For models of higher resolution datasets like CelebA-HQ, we need to be more careful; skipping more than 20% of the initial steps will induce some perceptible distortions in the generated images (Fig.17E bottom), which suggests that the Gaussian approximation is less effective for larger images. The reason may have to do with a low-quality covariance matrix estimate, which could arise from the number of training images being small compared to the effective dimensionality of the image manifold.

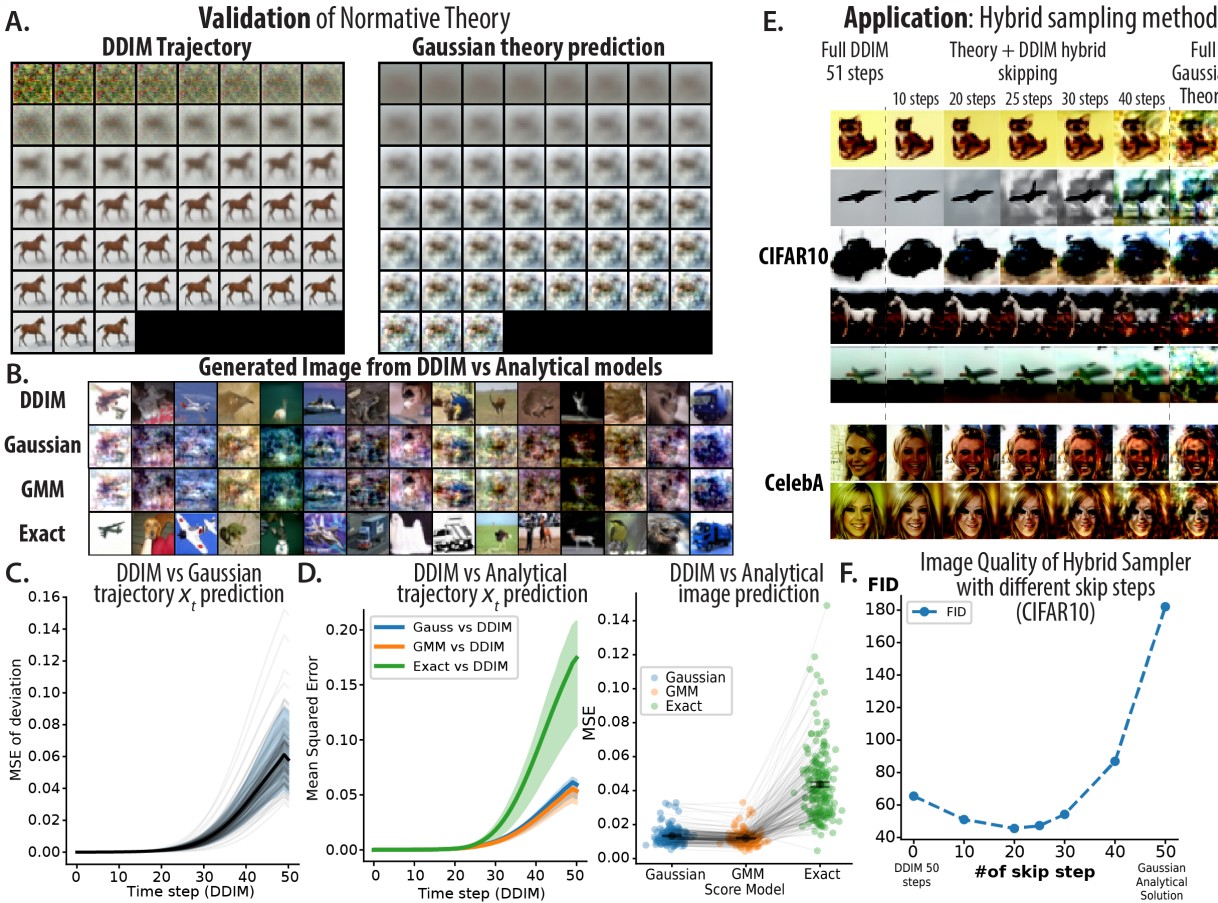

Figure 17: **Comparing analytical solutions of sampling trajectory with DDIM diffusion model for CIFAR-10**. **A.** $\hat{\mathbf{x}}_0(\mathbf{x}_t)$ of a DDIM trajectory and the Gaussian solution with the same initial condition $\mathbf{x}_T$. **B.** Samples generated by DDIM and the analytical theories from the same initial condition. **C.** Mean squared error between the $\mathbf{x}_t$ trajectory of DDIM and Gaussian solution. **D.** Comparing the state trajectory and final sample of three normative models (Gaussian, GMM, exact) with DDIM. **E.** Hybrid sampling method combines Gaussian theory prediction with DDIM. **F.** Image quality of the hybrid method (FID score) as function of different numbers of skipped steps (see Appendix A.8).

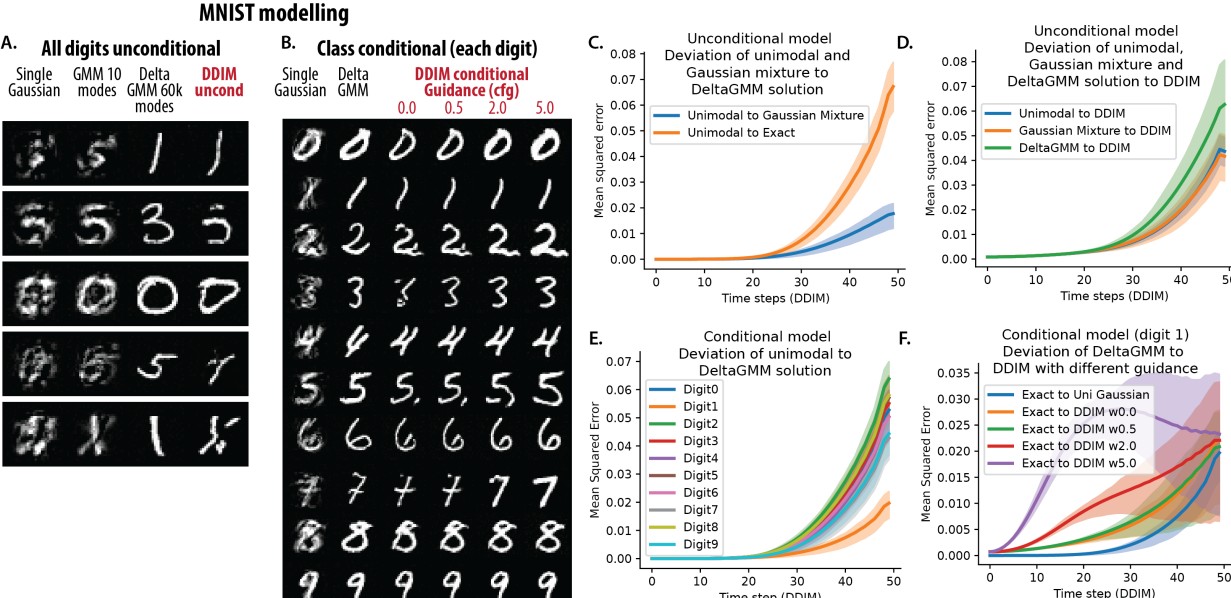

Figure 18: **Comparing diffusion dynamics guided by Gaussian and Gaussian mixture score with the ones learned by a neural network (MNIST)**. **A.** Unconditional model. In each row, it shows the image generated from single-mode Gaussian, 10-mode Gaussian mixture, Delta GMM defined on the whole MNIST dataset, and the unconditional diffusion model.

**B.** Conditional model of the 10 classes. In each row, from left to right, it shows the image generated via single Gaussian, delta GMM defined on all training data of that class, and the conditional diffusion model with different classifier free guidance strength 0.0 - 5.0.

**C.** Unconditional model, Deviation between trajectory predicted by unimodal Gaussian theory and 10-mode GMM and exact delta GMM. It shows that Gaussian mixture the same effect on the diffusion trajectory in the first phase as a matched unimodal Gaussian.

**D.** Unconditional model, Deviation between the trajectory predicted by different GMM theories and that sampled by DDIM. It shows that the trajectory sampled by DDIM is actually closer to that predicted by unimodal or 10 mode GMM, than the exact delta gmm model. This means the model didn't learn the exact score, but some coarse-grained approximation to it.

**E.** Conditional model, Deviation between unimodal Gaussian theory and exact delta GMM for different digits class. It shows some digits are better approximated by Gaussian than others.

**F.** Conditional model, Deviation between the trajectory predicted by exact delta GMM and that sampled by DDIM with different guidance scales. It shows that a larger guidance scale push the sampling result from the exact training distribution further.

## B.3 Extended results for Generated Image Similarity Comparison

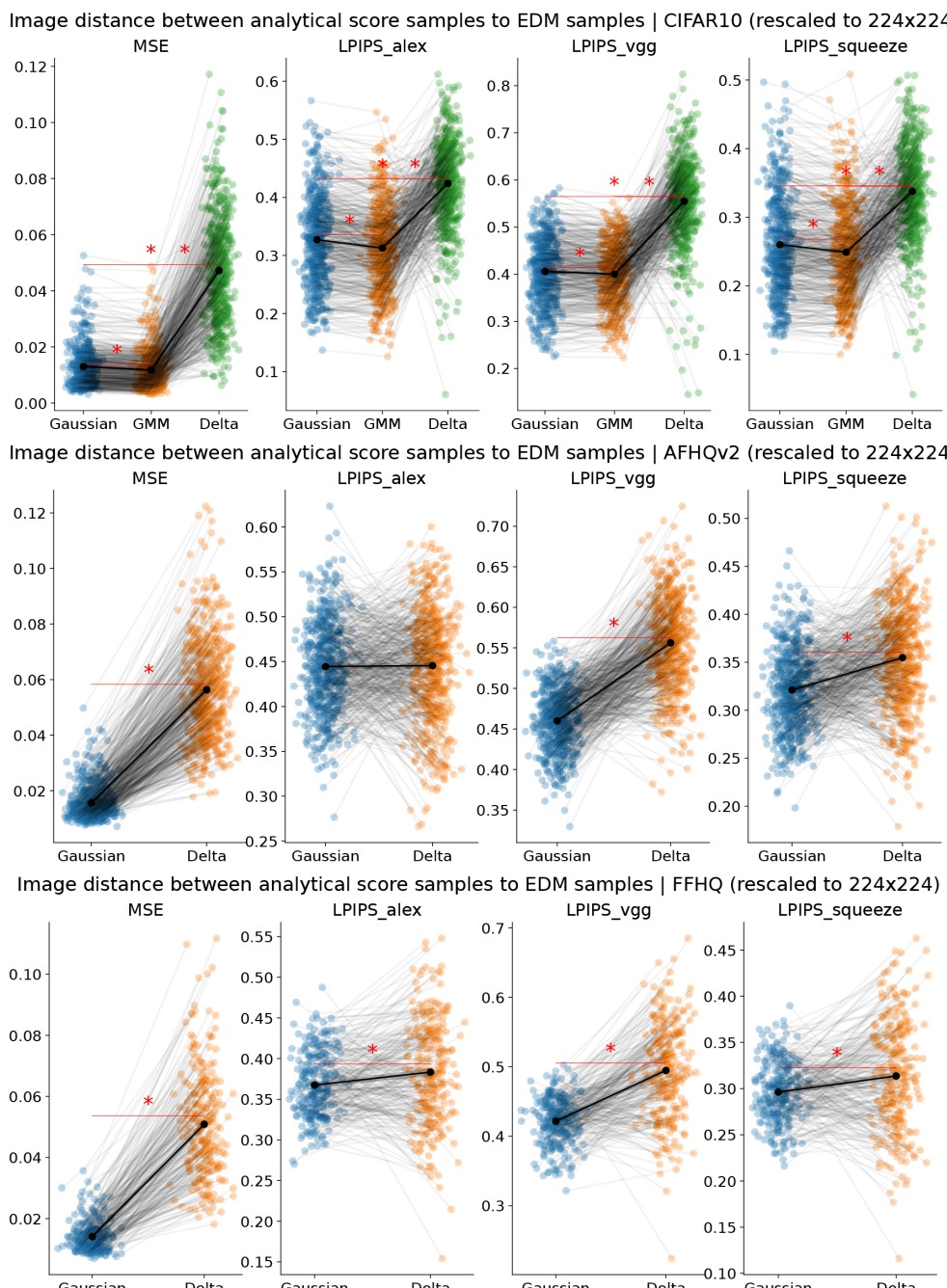

Figure 19: **Comparing image samples generated from pre-trained EDM model versus analytical score models.** Distance between the samples from the EDM and analytical score model $d(\mathbf{x}_{0,EDM}(\mathbf{x}_T), \mathbf{x}_{0,analytical}(\mathbf{x}_T))$ from the same initial noises $\mathbf{x}_T$ were plotted. Each panel showed one type of image metric $d$, MSE or LPIPS with different backbones AlexNet, VGG, SqueezeNet. Gaussian model consistently predicts EDM samples better than the exact delta score model, except for AlexNet based LPIPS in AFHQv2.

## B.4 Extended visualizations of score vector field

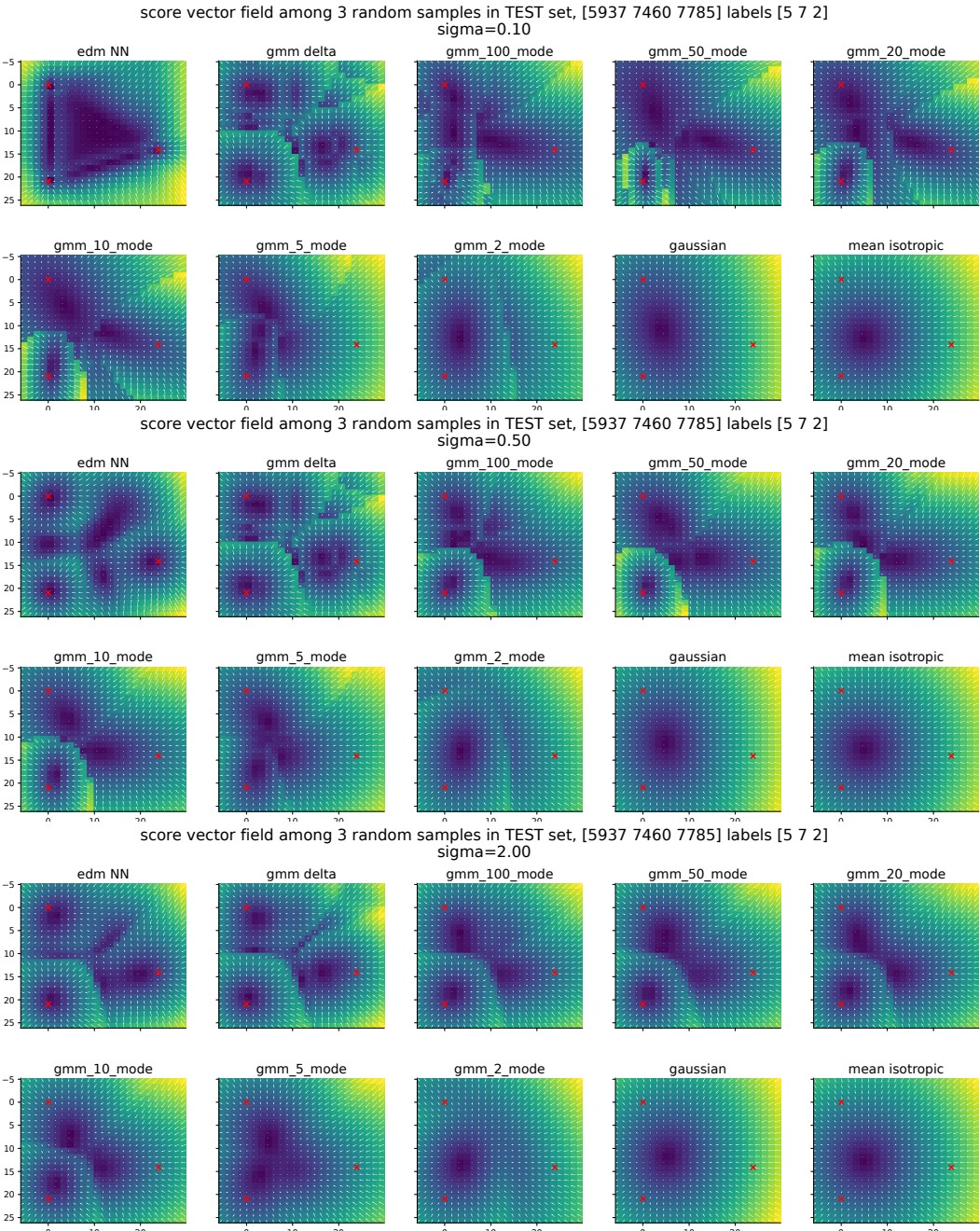

Figure 20: **Visualizing the score vector field** $s_\theta(\mathbf{x}, \sigma)$ **of pretrained EDM network with the analytical scores (Gaussian, Gaussian mixture, and Exact point cloud) as reference**. Same plotting scheme as Fig. 11, evaluated on the plane spanned by three **test set** samples of different classes, (2 5 7).

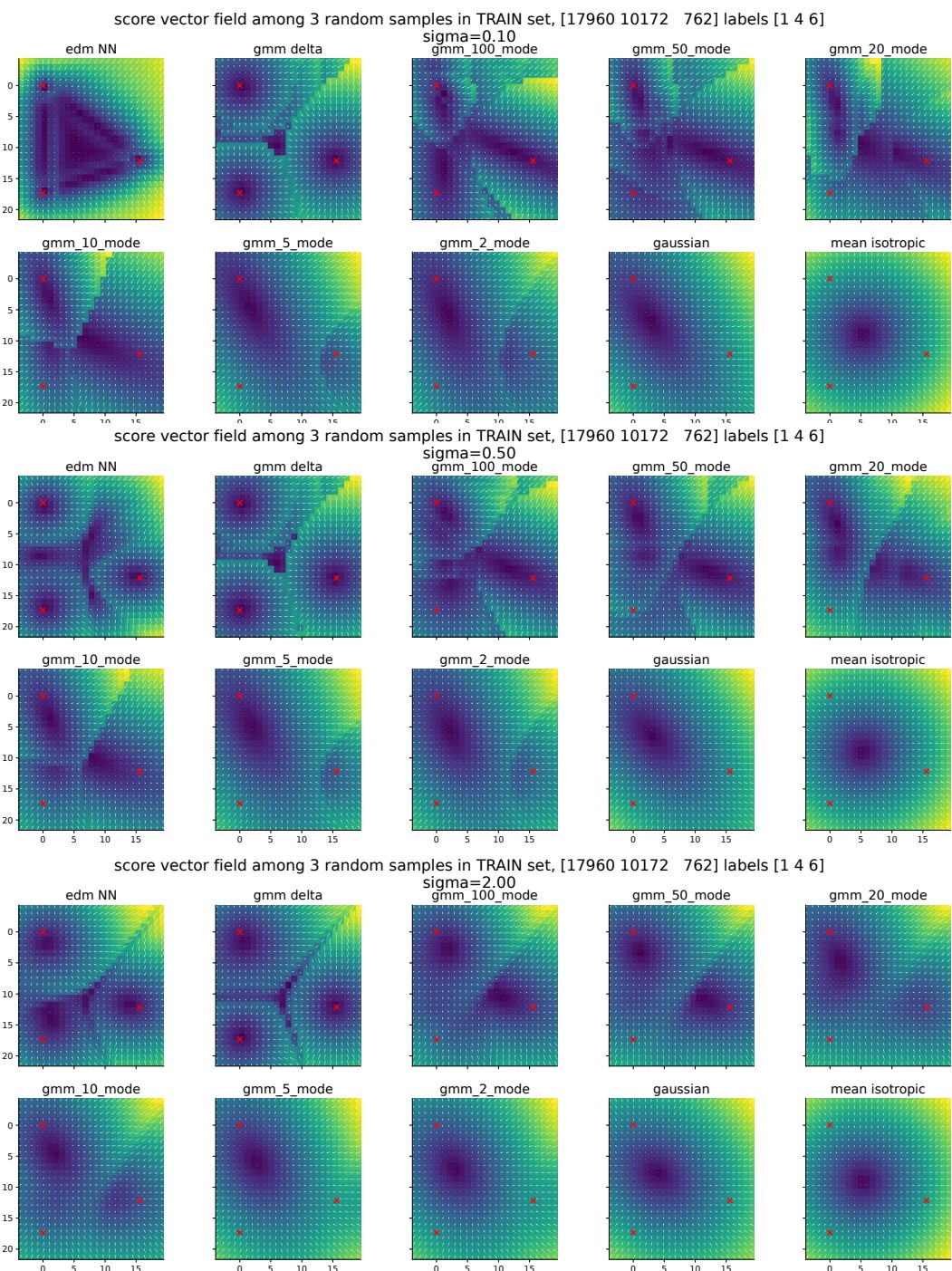

Figure 21: **Visualizing the score vector field $s_\theta(\mathbf{x}, \sigma)$ of pre-trained EDM network with the analytical scores (Gaussian, Gaussian mixture, and Exact point cloud) as reference**. Same plotting scheme as Fig. 11, evaluated on the plane spanned by three **training set** samples of different classes, (1 4 6).

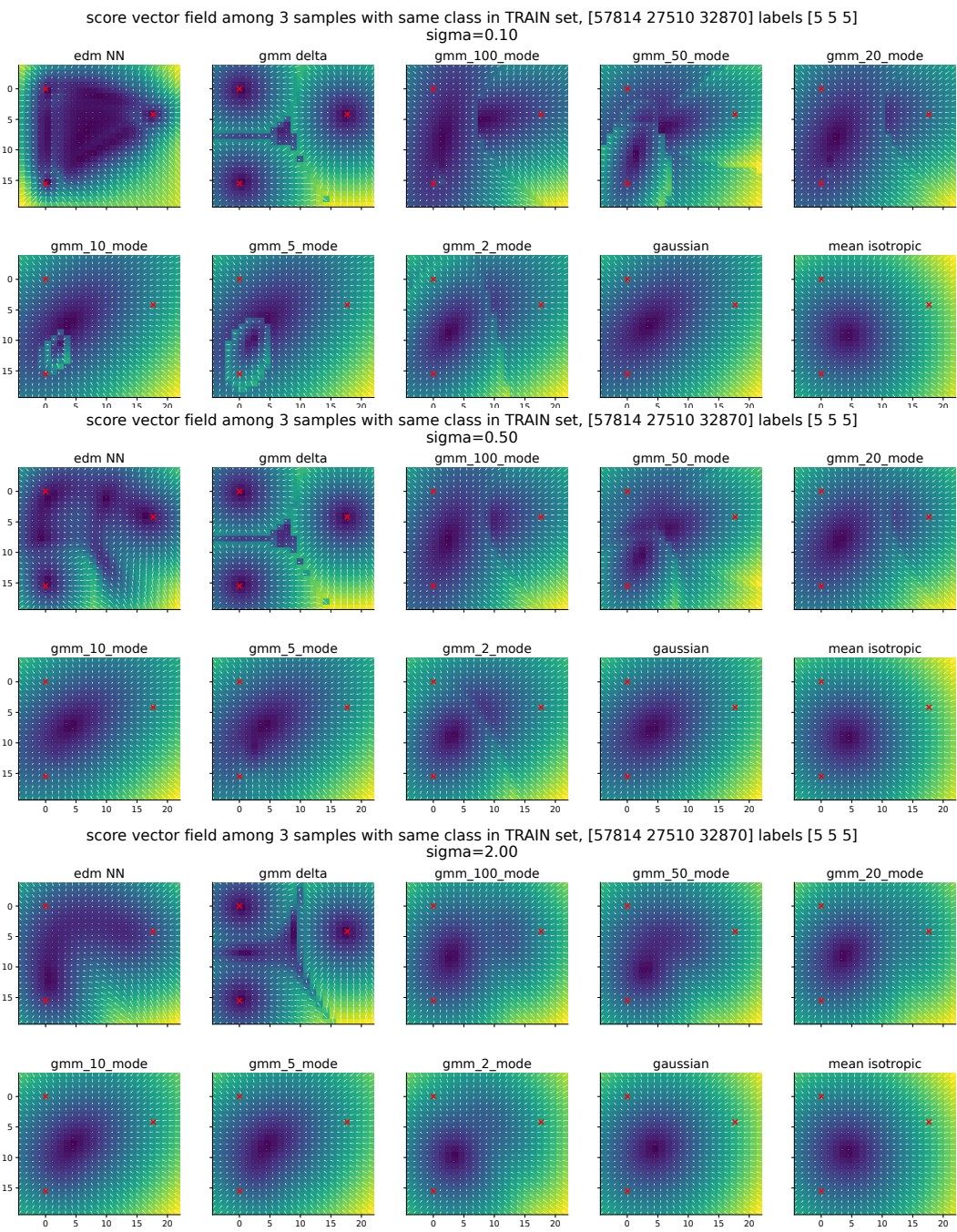

Figure 22: **Visualizing the score vector field** $s_\theta(\mathbf{x}, \sigma)$ **of pre-trained EDM network with the analytical scores (Gaussian, Gaussian mixture, and Exact point cloud) as reference**. Same plotting scheme as Fig. 11, evaluated on the plane spanned by three **training set** samples of the same classes (5).

## B.5 Extended results of comparing neural score with GMM with varying number and rank

Figure 23: **Deviation of the Learned Neural Score from the Score of a Gaussian Mixture Model with Varying Numbers of Components and Ranks, Fitted on Training Data**. Same plotting scheme as Fig. 9 but for MNIST.
**Upper:** Residual explained variance plotted as a function of the number of Gaussian modes on the x-axis, with each colored line representing a different rank. Each panel compares the score at a certain noise scale $\sigma$.
**Lower:** The same data plotted alternatively, with the number of ranks on the x-axis.

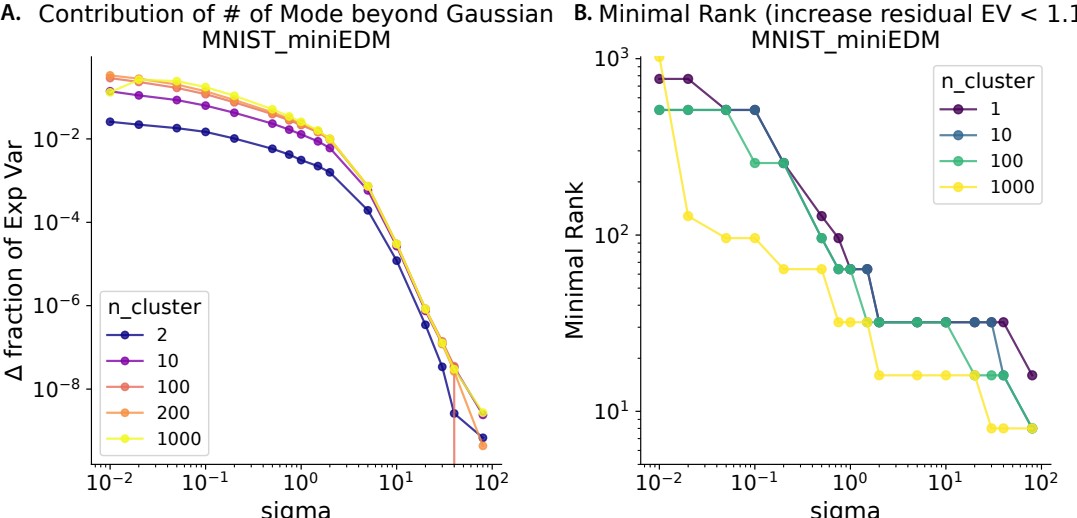

Figure 24: **Comparing Gaussian Mixture Score Model with Gaussian Model for Explaining Neural Score.** Same plotting format as Fig. 10, but for MNIST. **A.** Contribution of multiple mixture components beyond one Gaussian. **B.** Minimal rank required for each noise level.

## B.6 Extended results of neural score structure during diffusion training

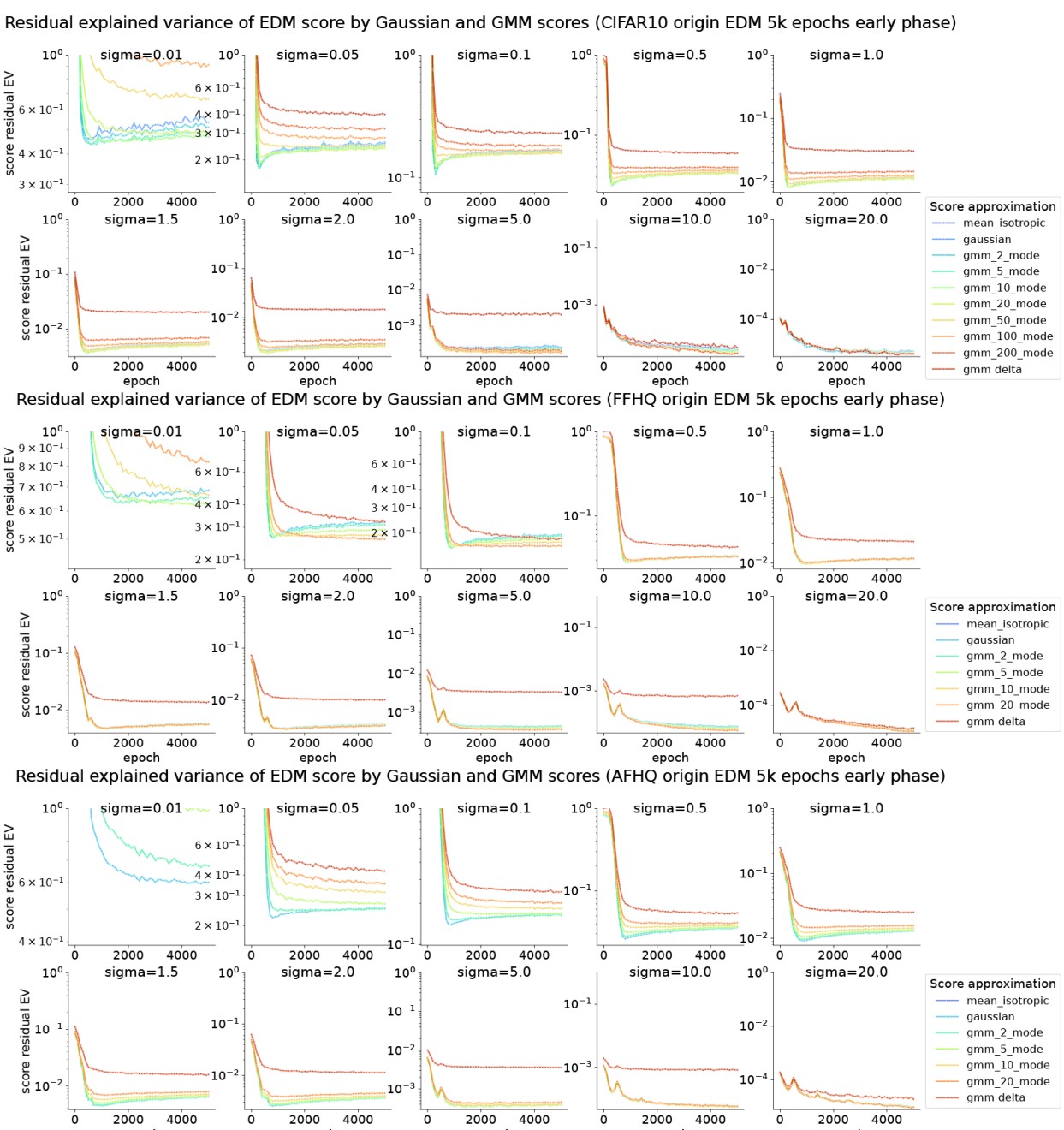

Figure 25: **Deviation between the score $s_\theta(\mathbf{x}_t, t)$ of EDM neural network and the analytical scores during diffusion training process**. Similar to Fig.12, but for CIFAR10, AFHQv2 64, FFHQ 64 datasets.

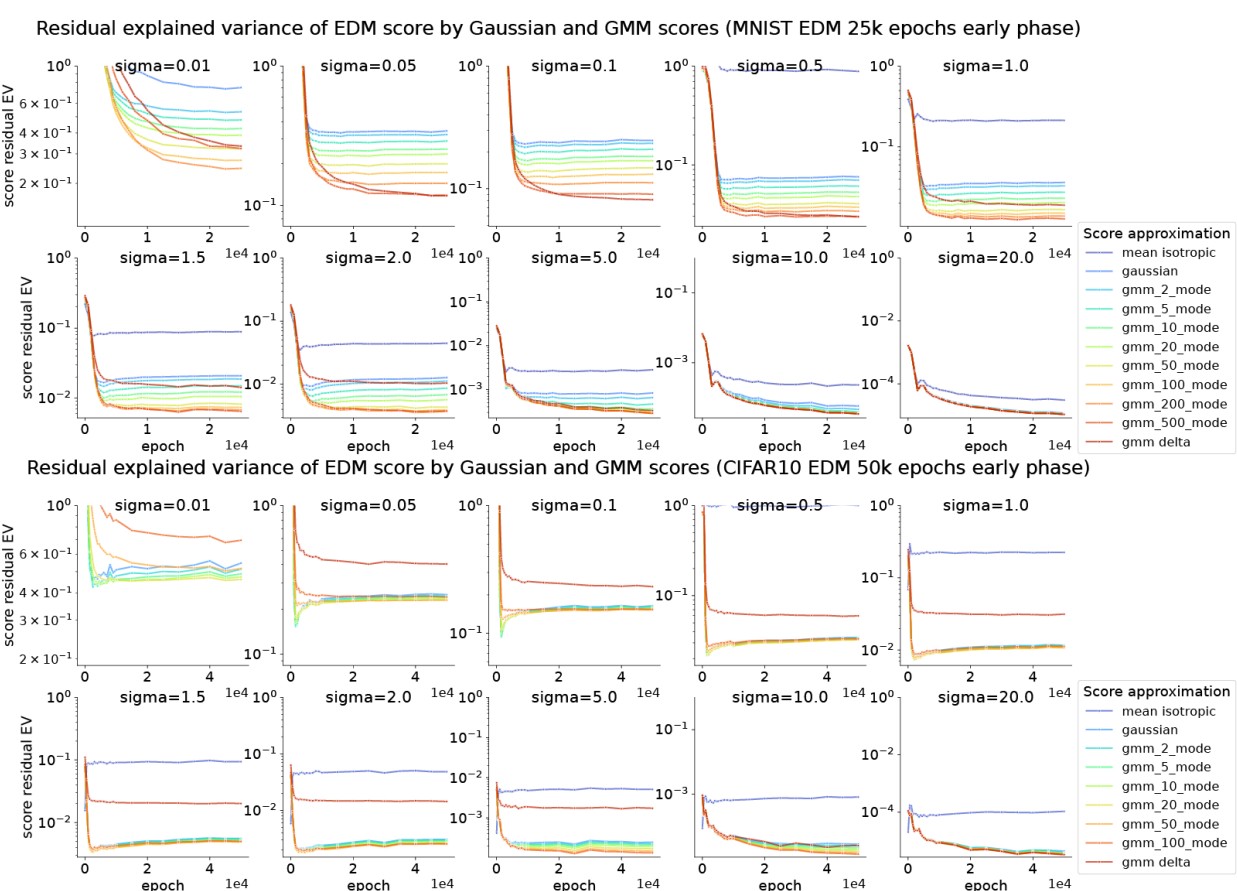

Figure 26: **Deviation between the score** $s_\theta(\mathbf{x}_t, t)$ **of EDM neural network and the analytical scores during diffusion training process**. Similar to Fig.12, but for MNIST and CIFAR datasets, using an alternative code base for EDM training (mini-EDM).

### B.7 Extended visualizations of neural score field during diffusion training

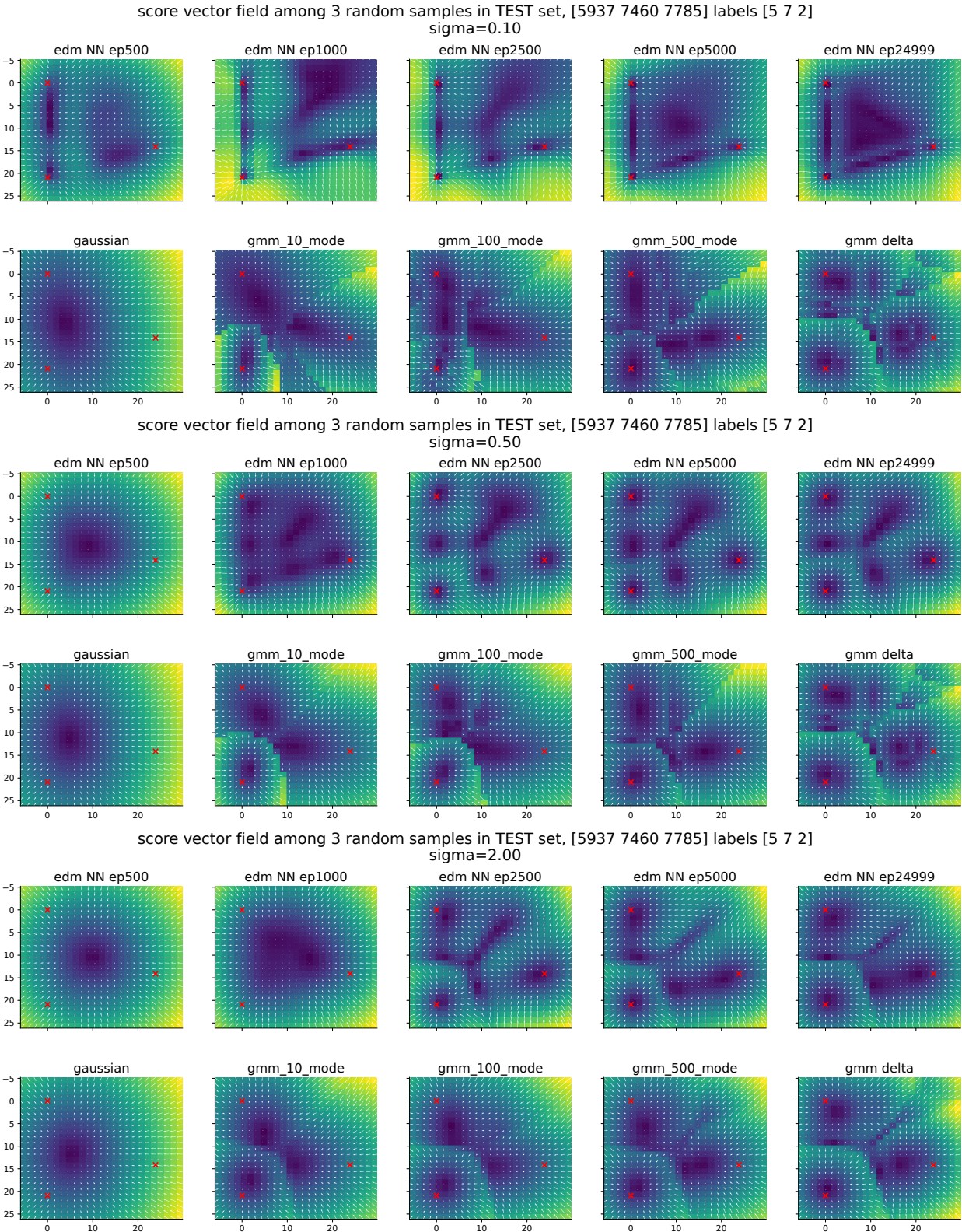

Figure 27: **Visualizing the score vector field** $s_\theta(\mathbf{x}_t, t)$ **of neural network during diffusion training process with the analytical scores as reference**. Same plotting format as Fig. 13. Score functions are evaluated on the plane spanned by three test samples with different classes.

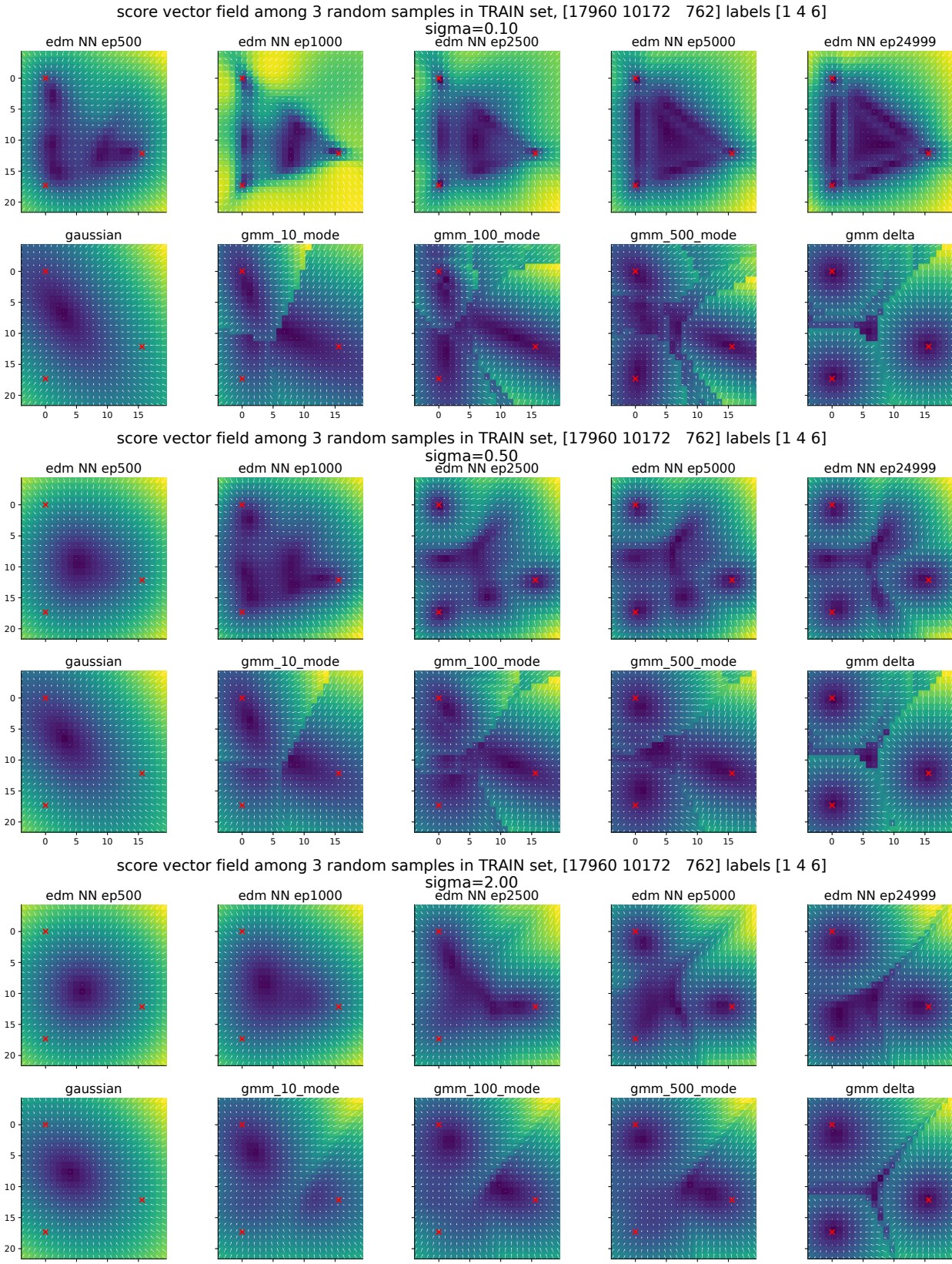

Figure 28: **Visualizing the score vector field** $s_\theta(\mathbf{x}_t, t)$ **of neural network during diffusion training process with the analytical scores as reference**. Same plotting format as Fig. 13. Score functions are evaluated on the plane spanned by three training samples with different classes

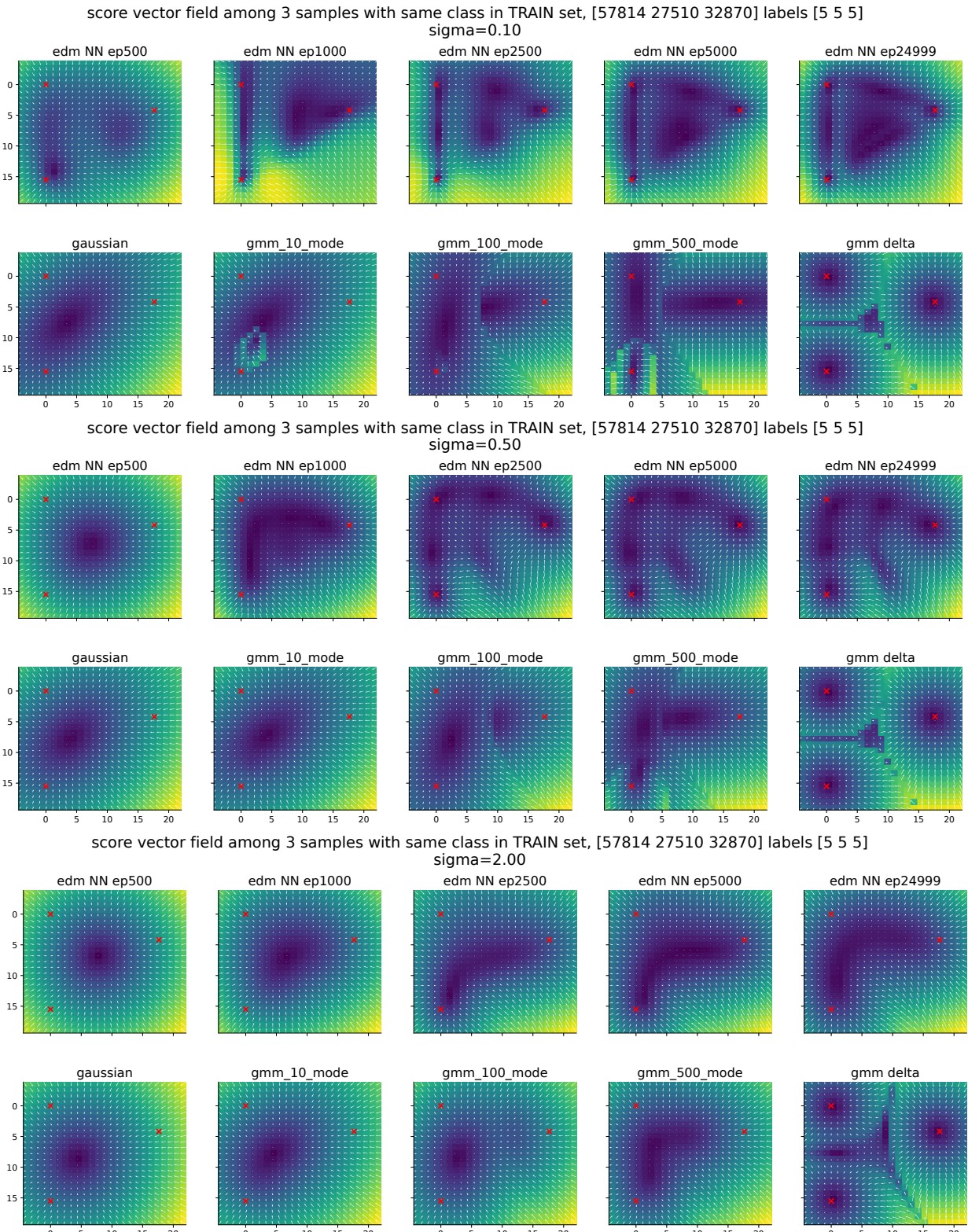

Figure 29: **Visualizing the score vector field** $s_\theta(\mathbf{x}_t, t)$ **of neural network during diffusion training process with the analytical scores as reference**. Same plotting format as Fig. 13. Score functions are evaluated on the plane spanned by three training samples with same classes (5).

## B.8 Extended results of FID scores for Gaussian analytical teleportation

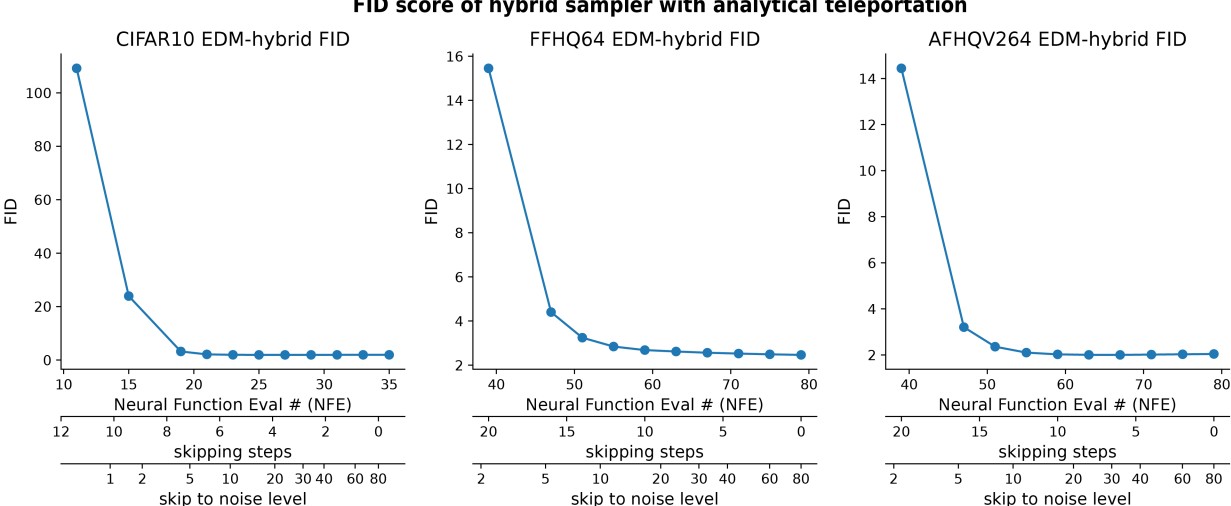

Figure 30: **Image quality as a function of skipping steps for hybrid sampling approach.** Note the main x-axes are the number of Neural Function Evaluation (NFE); the secondary x-axes are the number of skipping steps from the Heun sampler; the tertiary-axes are the time or noise level $\sigma_{skip}$ at which we evaluate the Gaussian solution. See Tab.6,4,5 for numbers.

The effects of applying the Gaussian analytical teleportation on the FID scores are presented below (Fig.30, Tab.5,4,6).

For CIFAR-10 model, we found teleportation not only reduces the number of required neural function calls, but also lowers the FID score. Admittedly, the FID score effect is quite small on the CIFAR-10 model we tried: it goes from 1.958 (no teleportation) to 1.933 (5 steps replaced by teleportation), which is around a 1% decrease. Visually, the difference in sample images is mostly negligible; there are some changes, but they are plausible changes in image details. Hence, teleportation saves 29% of NFE and improves the sample quality, without changing the model or the sampler. Further, skipping 6 steps will save 34% NFE but increase the FID score by less than .2%.

For AFHQv2-64, the teleportation can save 25-30% of NFE (10-12 steps skipped) or reduce the FID by 2% (6 steps skipped). For FFHQ64, the teleportation saves 10-15% of NFE (4-6 steps skipped) with around 3% increase in FID.

The full table of FID scores as a function of the number of skipped steps is shown below.

The fact that replacing NN evaluations with the analytical solution can even improve the already low FID score (albeit very slightly in the tests we just ran) is intriguing, and it is not immediately obvious why this is true. One possibility is that, early in reverse diffusion, the neural network might be a worse or noisier version of the Gaussian score.

Table 4: FFHQ64 FID with analytical teleportation

| Nskip | NFE | time/noise scale | FID |
|---|---|---|---|
| 0 | 79 | 80.0 | 2.464 |
| 2 | 75 | 60.1 | 2.489 |
| 4 | 71 | 44.6 | 2.523 |
| 6 | 67 | 32.7 | 2.561 |
| 8 | 63 | 23.6 | 2.617 |
| 10 | 59 | 16.8 | 2.681 |
| 12 | 55 | 11.7 | 2.841 |
| 14 | 51 | 8.0 | 3.243 |
| 16 | 47 | 5.4 | 4.402 |
| 20 | 39 | 2.2 | 15.451 |

Table 5: AFHQV264 FID with analytical teleportation

| Nskip | NFE | time/noise scale | FID |
|---|---|---|---|
| 0 | 79 | 80.0 | 2.043 |
| 2 | 75 | 60.1 | 2.029 |
| 4 | 71 | 44.6 | 2.016 |
| 6 | 67 | 32.7 | 2.003 |
| 8 | 63 | 23.6 | 2.005 |
| 10 | 59 | 16.8 | 2.026 |
| 12 | 55 | 11.7 | 2.102 |
| 14 | 51 | 8.0 | 2.359 |
| 16 | 47 | 5.4 | 3.206 |
| 20 | 39 | 2.2 | 14.442 |

Table 6: CIFAR10 FID with analytical teleportation

| Nskip | NFE | time/noise scale | FID |
|---|---|---|---|
| 0 | 35 | 80.0 | 1.958 |
| 1 | 33 | 57.6 | 1.955 |
| 2 | 31 | 40.8 | 1.949 |
| 3 | 29 | 28.4 | 1.940 |
| 4 | 27 | 19.4 | 1.932 |
| 5 | 25 | 12.9 | 1.934 |
| 6 | 23 | 8.4 | 1.963 |
| 7 | 21 | 5.3 | 2.123 |
| 8 | 19 | 3.3 | 3.213 |
| 10 | 15 | 1.1 | 23.947 |
| 12 | 11 | 0.3 | 109.178 |

## B.9 Extended results of FID scores for Gaussian analytical teleportation: Experiment 2

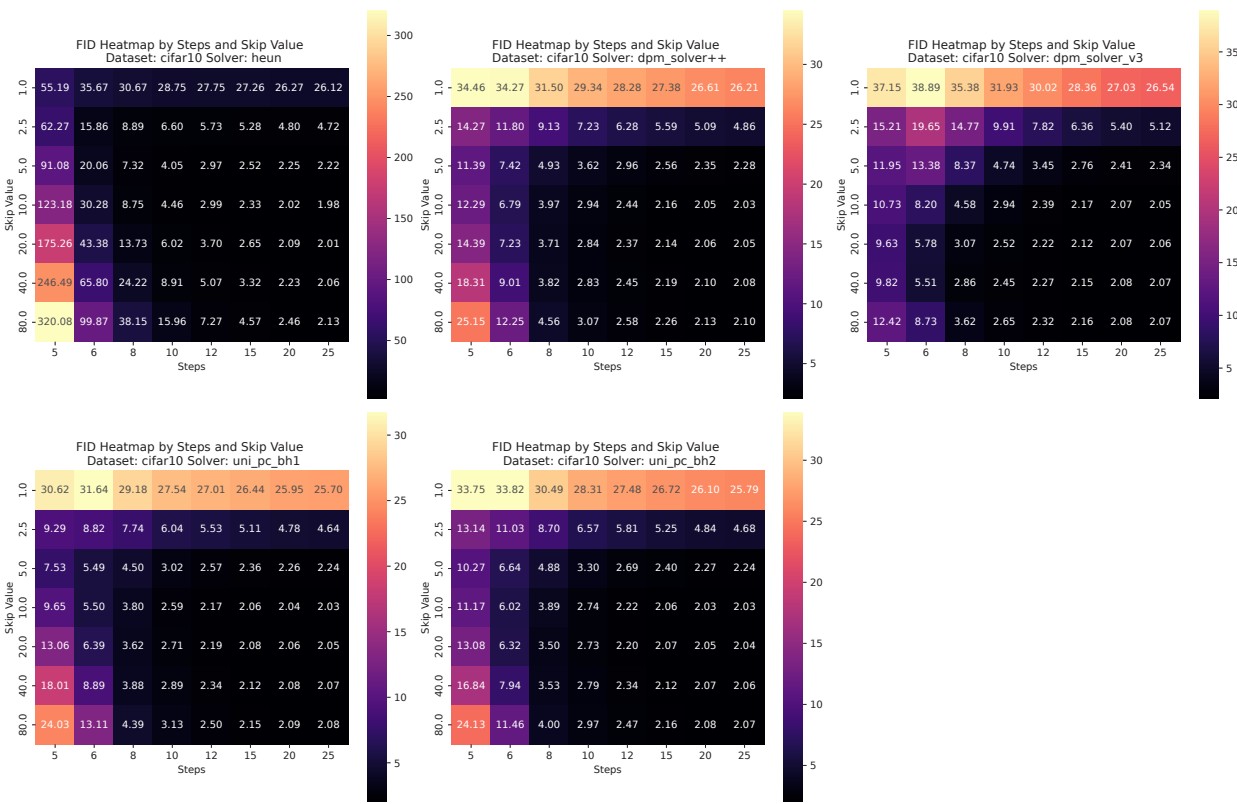

Figure 31: **Image quality as a function of skipping noise scale $\sigma_{skip}$ and sampling step number for hybrid sampling approach (CIFAR10).** y-axis of heatmap (*Skip Value*) denotes the $\sigma_{skip}$ (edm convention, $\sigma_{min} = 0.002, \sigma_{max} = 80$); x-axis denotes sampling steps. Each panel features hybrid sampler with one deterministic sampler: *Heun, DPM_solver++, DPM_solver_v3, UniPC-bh1, UniPC-bh2.*

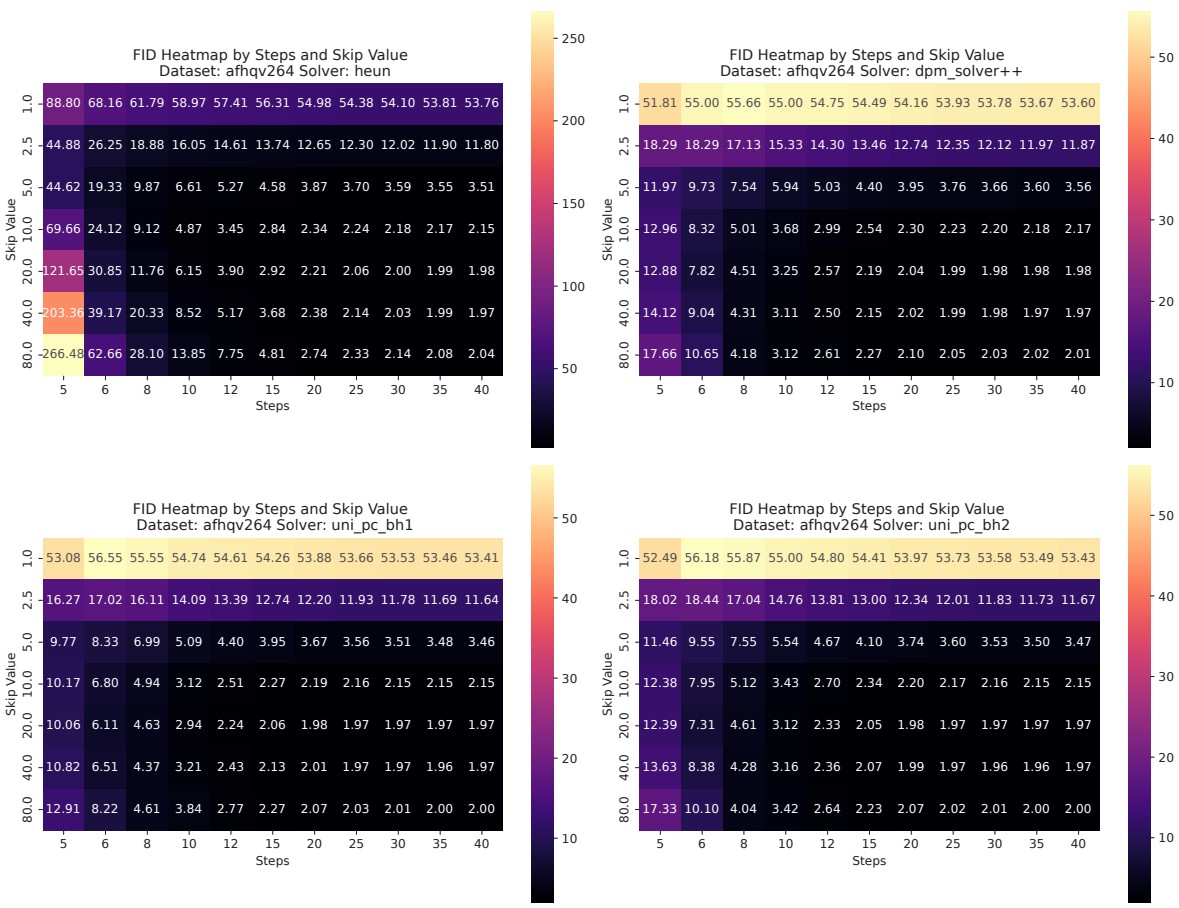

Figure 32: **Image quality as a function of skipping noise scale $\sigma_{skip}$ and sampling step number for hybrid sampling approach (AFHQv2).** Same plotting format as Fig. 31

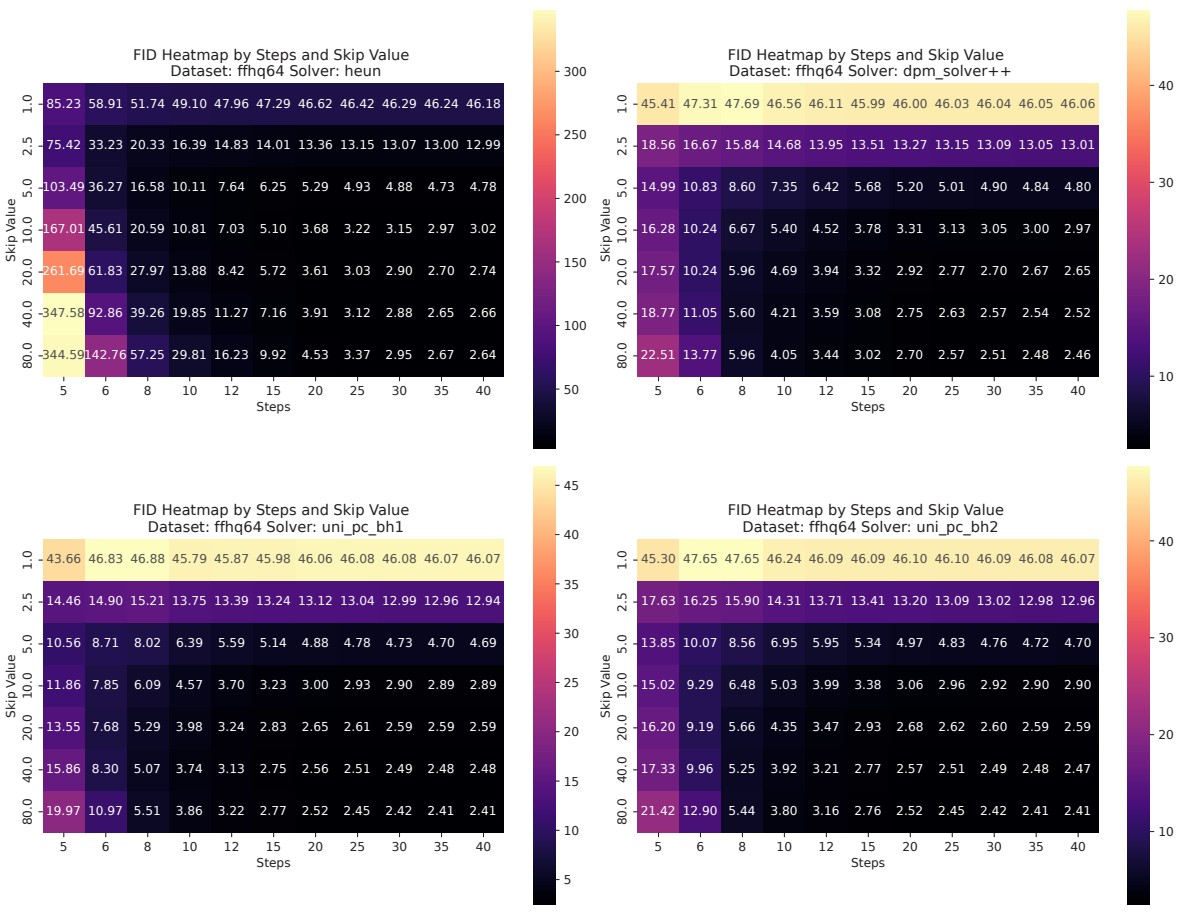

Figure 33: **Image quality as a function of skipping noise scale $\sigma_{skip}$ and sampling step number for hybrid sampling approach (FFHQ).** Same plotting format as Fig. 31

## C  Notation correspondence

Diffusion models usually have forward processes whose conditional probabilities are

$$p(\mathbf{x}_t|\mathbf{x}_0) = \mathcal{N}(A_t\mathbf{x}_0, B_t\mathbf{I}) \tag{42}$$

for all $t \in [0, T]$. In the limit of small time steps, the transition probability distribution can be captured by the SDE

$$\dot{\mathbf{x}}_t = -C_t\mathbf{x}_t + D_t\eta(t) \tag{43}$$

where $\eta(t)$ is a vector of independent Gaussian white noise terms.

Papers discussing these models may use slightly different notation. In the table below, we briefly indicate how various choices of notation correspond to one another. To make comparing discrete and continuous models easier, we assume the time step size is $\Delta t = 1$.

Table 7: **Comparison of notation for diffusion model parameters**.
†: EDM formulation does not explicitly specify a forward SDE in the paper, so we left out these notations.

| Paper | Citation | $A_t$ | $B_t$ | $C_t$ | $D_t$ |
|---|---|---|---|---|---|
| DDPM | Ho et al. (2020) | $\sqrt{\bar{\alpha}_t}$ | $1 - \bar{\alpha}_t$ | $1 - \sqrt{1 - \beta_t}$ | $\sqrt{\beta_t}$ |
| DDIM | Song et al. (2020a) | $\sqrt{\alpha_t}$ | $1 - \alpha_t$ | $1 - \sqrt{\alpha_t/\alpha_{t-1}}$ | $\sqrt{1 - \alpha_t/\alpha_{t-1}}$ |
| Stable Diff. | Rombach et al. (2022) | $\alpha_t$ | $\sigma_t^2$ | $1 - \alpha_t/\alpha_{t-1}$ | $\sqrt{\sigma_t^2 - (\alpha_t/\alpha_{t-1})^2 \sigma_{t-1}^2}$ |
| VP SDE | Song et al. (2021) | $\exp\left[-\frac{1}{2}\int_0^t \beta(s)ds\right]$ | $1 - \exp\left[-\int_0^t \beta(s)ds\right]$ | $\beta(t)/2$ | $\sqrt{\beta(t)}$ |
| EDM | Karras et al. (2022) | $1$ | $\sigma(t)^2$ | $0$ | NA† |
| EDM with scaling | Karras et al. (2022) | $s(t)$ | $s^2(t)\sigma^2(t)$ | NA† | NA† |
| Ours | | $\alpha_t$ | $\sigma_t^2$ | $\beta(t)$ | $g(t)$ |

In the popular huggingface `diffusers` library implementation of diffusion models, the function `alphas_cumprod` corresponds to our $\alpha_t^2$.

# D  Detailed Derivations

## D.1  Detailed derivation of the solution of the Gaussian Diffusion

In this section, we derive the analytic solution $\mathbf{x}_t$ for the probability flow ODE in EDM formulation (Eq.2) with Gaussian score. We will assume that the score function corresponds to a Gaussian distribution whose mean is $\mu$ and covariance matrix is $\mathbf{\Sigma}$. We will also assume, given that images data are usually thought of as residing on a low-dimensional manifold within pixel space, that the rank $r$ of the covariance matrix may be less than the dimensionality $D$ of state space. Let $\mathbf{\Sigma} = \mathbf{U}\mathbf{\Lambda}\mathbf{U}^T$ be the eigendecomposition or compact SVD of the covariance matrix, where $\mathbf{U}$ is a $D \times r$ semi-orthogonal matrix whose columns are normalized (i.e. $\mathbf{U}^T\mathbf{U} = \mathbf{I}_r$), and $\mathbf{\Lambda}$ is the $r \times r$ diagonal eigenvalue matrix. Denote the $k$th column of $\mathbf{U}$ by $\mathbf{u}_k$ and the $k$th diagonal element of $\mathbf{\Lambda}$ by $\lambda_k$.

The probability flow ODE used for the EDM model reads,

$$d\mathbf{x} = -\sigma(\sigma^2 I + \Sigma)^{-1}(\mu - \mathbf{x})d\sigma \tag{44}$$

$$d\mathbf{x} = -\frac{1}{\sigma}(I - \mathbf{U}\tilde{\Lambda}_\sigma\mathbf{U}^T)(\mu - \mathbf{x})d\sigma \ , \tag{45}$$

$$\tilde{\Lambda}_\sigma = diag\Big[\frac{\lambda_k}{\lambda_k + \sigma^2}\Big] \tag{46}$$

We can see it's a linear ODE with respect to $\mathbf{x} - \mu$, with disentangled dynamics along each eigenvector $\mathbf{u}_k$. Choosing our dynamic variable to be the projection coefficient on each axes $c_k(\sigma) = \mathbf{u}_k^T(\mathbf{x} - \mu)$. Then its dynamics could be written as

$$dc_k(\sigma) = \frac{\sigma}{\lambda_k + \sigma^2}c_k(\sigma)d\sigma \tag{47}$$

Integrating this, we get the following, with integral constant $K$.

$$\frac{dc_k(\sigma)}{c_k(\sigma)} = d\log\sqrt{\lambda_k + \sigma^2} \tag{48}$$

$$d\log c_k(\sigma) = d\log\sqrt{\lambda_k + \sigma^2} \tag{49}$$

$$c_k(\sigma) = \sqrt{\lambda_k + \sigma^2}K \tag{50}$$

Using the initial condition $c_k(T) = \mathbf{u}_k^T(\mathbf{x}_T - \mu)$, we have

$$\frac{c_k(\sigma)}{c_k(T)} = \sqrt{\frac{\lambda_k + \sigma^2}{\lambda_k + \sigma_T^2}} =: \psi(\lambda_k, \sigma) \tag{51}$$

Thus, we arrive at the solution Eq.15.

$$\mathbf{x}_\sigma - \mu = \sum_k c_k(\sigma)\mathbf{u}_k \tag{52}$$

$$\mathbf{x}_\sigma = \mu + \sum_k \psi(\lambda_k, \sigma)c_k(T)\mathbf{u}_k \tag{53}$$

$$\mathbf{x}_\sigma = \mu + \sum_k \sqrt{\frac{\lambda_k + \sigma^2}{\lambda_k + \sigma_T^2}}\mathbf{u}_k\mathbf{u}_k^T(\mathbf{x}_T - \mu) \tag{54}$$

When the $\mathbf{U}$ is rank deficient, we will recover the off-manifold term as in Eq. 15 by setting $\lambda_k = 0$.

For the denoising outcome, we have Eq. 3,

$$D(\mathbf{x}, \sigma) = \mathbf{x} + \sigma^2\nabla\log p(\mathbf{x}, \sigma) \tag{55}$$

$$= \mathbf{x} + (I - \mathbf{U}\tilde{\Lambda}_\sigma\mathbf{U}^T)(\mu - \mathbf{x}) \tag{56}$$

$$= \mu + \mathbf{U}\tilde{\Lambda}_\sigma\mathbf{U}^T(\mathbf{x} - \mu) \tag{57}$$

Bring in the solution of $\mathbf{x}$ above, we have

$$D(\mathbf{x}_\sigma, \sigma) = \mu + \sum_k \frac{\lambda_k}{\sqrt{(\lambda_k + \sigma^2)(\lambda_k + \sigma_T^2)}} \mathbf{u}_k \mathbf{u}_k^T (\mathbf{x}_T - \mu) \tag{58}$$

If we define

$$\xi(\lambda_k, \sigma) := \frac{\lambda_k}{\sqrt{(\lambda_k + \sigma^2)(\lambda_k + \sigma_T^2)}} \tag{59}$$

and use the definition of $c_k(T)$, then we get the trajectory for denoiser image as Eq.16.

$$D(\mathbf{x}_\sigma, \sigma) = \mu + \sum_k \xi(\lambda_k, \sigma) c_k(T) \mathbf{u}_k \tag{60}$$

## D.2 Deriving solution to PF-ODE with data scaling term

If we have a data scaling term $\alpha_t$, we can define the scaled state $\tilde{\mathbf{x}}_t = \alpha_t \mathbf{x}_t$, and the effective noise scale $\tilde{\sigma}_t = \alpha_t \sigma_t$. Equivalent to Eq.2, the PF-ODE governing the dynamics of this state will be

$$d\tilde{\mathbf{x}}_t = \alpha_t d\mathbf{x}_t + d\alpha_t \mathbf{x}_t \tag{61}$$

$$= [-\alpha_t \dot{\sigma}_t \sigma_t \nabla_\mathbf{x} \log p(\mathbf{x}_t, \sigma_t) + \dot{\alpha}_t \mathbf{x}_t] dt \tag{62}$$

$$= [\frac{\dot{\alpha}_t}{\alpha_t} \tilde{\mathbf{x}}_t - \alpha_t^2 \dot{\sigma}_t \sigma_t \nabla_{\tilde{\mathbf{x}}} \log p(\tilde{\mathbf{x}}_t/\alpha_t, \sigma_t)] dt \tag{63}$$

Without solving this ODE directly, we can directly translate the solutions of Eq. 15, 16 by subsituting $\mathbf{x}_t \to \mathbf{x}_t/\alpha_t$ and $\sigma_t \to \sigma_t/\alpha_t$. Using this simple rule, we get the solutions for general probability flow ODE with data scaling.

$$\mathbf{x}_\sigma/\alpha_t = \mu + \sum_k \sqrt{\frac{\lambda_k + \sigma^2/\alpha_t^2}{\lambda_k + \sigma_T^2/\alpha_T^2}} \mathbf{u}_k \mathbf{u}_k^T (\mathbf{x}_T/\alpha_T - \mu) \tag{64}$$

$$\mathbf{x}_\sigma = \alpha_t \mu + \sum_k \sqrt{\frac{\lambda_k \alpha_t^2 + \sigma_t^2}{\lambda_k \alpha_T^2 + \sigma_T^2}} \mathbf{u}_k \mathbf{u}_k^T (\mathbf{x}_T - \alpha_T \mu) \tag{65}$$

$$D(\mathbf{x}_\sigma, \sigma) = \mu + \sum_k \frac{\alpha_t \lambda_k}{\sqrt{(\alpha_t^2 \lambda_k + \sigma_t^2)(\alpha_T^2 \lambda_k + \sigma_T^2)}} \mathbf{u}_k \mathbf{u}_k^T (\mathbf{x}_T - \alpha_T \mu) \tag{66}$$

If we set $\alpha_t = 1$, these solution recovers the form in Eq. 15, 16. If we set $\alpha_t^2 + \sigma_t^2 = 1$, it will recover the solution of VP-SDE or DDIM.

## D.3 Detailed derivation of the solution of the Gaussian Diffusion with VP-ODE

In the following section, we present an *ab initio* derivation of the solution to VP-SDE. We consider forward processes defined by the stochastic differential equation (SDE)

$$\dot{\mathbf{x}} = -\beta(t)\mathbf{x} + g(t)\eta(t) \tag{67}$$

where $\beta(t)$ controls the decay of signal, $g(t)$ is a time-dependent noise amplitude, $\eta(t)$ is a vector of independent Gaussian white noise terms, and time runs from $t = 0$ to $T$. Its reverse process is

$$\dot{\mathbf{x}} = -\beta(t)\mathbf{x} - g(t)^2 \mathbf{s}(\mathbf{x}, t) + g(t)\eta(t) \tag{68}$$

where $\mathbf{s}(\mathbf{x}, t) := \nabla_\mathbf{x} \log p(\mathbf{x}, t)$ is the score function, and where we use the standard convention that time runs *backward*, i.e. from $t = T$ to 0. The variance-preserving SDE, which enforces the constraint $\beta(t) = \frac{1}{2}g^2(t)$. The marginal probabilities of this process are

$$p(\mathbf{x}_t|\mathbf{x}_0) = \mathcal{N}(\mathbf{x}_t|\alpha_t \mathbf{x}_0, \sigma_t^2 \mathbf{I}) \qquad \alpha_t := e^{-\int_0^t \beta(t')dt'} \qquad \sigma_t^2 := 1 - e^{-2\int_0^t \beta(t')dt'} \tag{69}$$

where $\alpha_t$ and $\sigma_t$ represent the signal and noise scale, satisfying $\alpha_t^2 + \sigma_t^2 = 1$. Normally, as $t$ goes from $0 \to T$, signal scale $\alpha_t$ monotonically decreases from $1 \to 0$ and $\sigma$ increases from $0 \to 1$. The equivalent *probability flow ODE* with the same marginal probabilities Song et al. (2021) is

$$\dot{\mathbf{x}} = -\beta(t)\mathbf{x} - \frac{1}{2}g(t)^2\mathbf{s}(\mathbf{x}, t) \tag{70}$$

where time again runs backward from $t = 1$ to $t = 0$.

For a Gaussian score, the probability flow ODE that reverses a VP-SDE forward process Song et al. (2021) is

$$\dot{\mathbf{x}} = -\beta(t)\mathbf{x} - \frac{1}{2}g^2(t)(\sigma_t^2\mathbf{I} + \alpha_t^2\mathbf{\Sigma})^{-1}(\alpha_t\mu - \mathbf{x}) . \tag{71}$$

Using the eigen decomposition of $\mathbf{\Sigma}$ described in the main text,

$$\dot{\mathbf{x}} = -\beta(t)\mathbf{x} - \frac{1}{2}g^2(t)\frac{1}{\sigma_t^2}(\mathbf{I} - \mathbf{U}\tilde{\mathbf{\Lambda}}_t\mathbf{U}^T)(\alpha_t\mu - \mathbf{x}) \tag{72}$$

where $\tilde{\mathbf{\Lambda}}_t$ is defined to be the time-dependent diagonal matrix

$$\tilde{\mathbf{\Lambda}}_t = \mathrm{diag}\left[\frac{\alpha_t^2\lambda_k}{\alpha_t^2\lambda_k + \sigma_t^2}\right] . \tag{73}$$

Consider the dynamics of the quantity $\mathbf{x}_t - \alpha_t\mu$. Using the relationship between $\beta_t$ and $\alpha_t$, we have

$$\frac{d}{dt}(\mathbf{x}_t - \alpha_t\mu) = \dot{\mathbf{x}}_t - \mu\dot{\alpha}_t \tag{74}$$

$$= \dot{\mathbf{x}}_t + \beta_t\alpha_t\mu \tag{75}$$

$$= \beta_t(\alpha_t\mu - \mathbf{x}) - \frac{1}{2}g^2(t)\frac{1}{\sigma_t^2}(\mathbf{I} - \mathbf{U}\tilde{\mathbf{\Lambda}}_t\mathbf{U}^T)(\alpha_t\mu - \mathbf{x}) \tag{76}$$

$$= \left[\frac{1}{2}g^2(t)\frac{1}{\sigma_t^2}(\mathbf{I} - \mathbf{U}\tilde{\mathbf{\Lambda}}_t\mathbf{U}^T) - \beta_t\mathbf{I}\right](\mathbf{x} - \alpha_t\mu) . \tag{77}$$

If we assume that the forward process is a variance-preserving SDE, then $\beta_t = \frac{1}{2}g^2(t)$, which implies $\alpha_t^2 = 1 - \sigma_t^2$. Using this, we obtain

$$\frac{d}{dt}(\mathbf{x}_t - \alpha_t\mu) = \beta_t\left[\frac{1}{\sigma_t^2}(\mathbf{I} - \mathbf{U}\tilde{\mathbf{\Lambda}}_t\mathbf{U}^T) - \mathbf{I}\right](\mathbf{x} - \alpha_t\mu) \tag{78}$$

$$= \beta_t\left[(\frac{1}{\sigma_t^2} - 1)\mathbf{I} - \frac{1}{\sigma_t^2}\mathbf{U}\tilde{\mathbf{\Lambda}}_t\mathbf{U}^T\right](\mathbf{x} - \alpha_t\mu) . \tag{79}$$

Define the variable $\mathbf{y}_t := \mathbf{x}_t - \alpha_t\mu$. We have just shown that its dynamics are fairly 'nice', in the sense that the above equation is well-behaved separable linear ODE. As we are about to show, it is exactly solvable.

Write $\mathbf{y}_t$ in terms of the orthonormal columns of $\mathbf{U}$ and a component that lies entirely in the orthogonal space $\mathbf{U}^\perp$:

$$\mathbf{y}_t = \mathbf{y}^\perp(t) + \sum_{k=1}^{r} c_k(t)\mathbf{u}_k , \ \mathbf{y}^\perp(t) \in \mathbf{U}^\perp . \tag{80}$$

The dynamics of the coefficient $c_k(t)$ attached to the eigenvector $\mathbf{u}_k$ are

$$\dot{c}_k(t) = \frac{d}{dt}(\mathbf{u}_k^T\mathbf{y}_t) = \beta_t\left[(\frac{1}{\sigma_t^2} - 1) - \frac{1}{\sigma_t^2}\frac{\alpha_t^2\lambda_k}{\alpha_t^2\lambda_k + \sigma_t^2}\right](\mathbf{u}_k^T\mathbf{y}_t) \tag{81}$$

$$= \frac{\beta_t}{\sigma_t^2}\left(1 - \sigma_t^2 - \frac{\alpha_t^2\lambda_k}{\alpha_t^2\lambda_k + \sigma_t^2}\right)c_k(t) \tag{82}$$

$$= \frac{\beta_t\alpha_t^2}{\sigma_t^2}\left(1 - \frac{\lambda_k}{\alpha_t^2\lambda_k + \sigma_t^2}\right)c_k(t) . \tag{83}$$

Using the constraint that $\alpha_t^2 + \sigma_t^2 = 1$, this becomes

$$\dot{c}_k(t) = \frac{\beta_t \alpha_t^2 (1 - \lambda_k)}{\alpha_t^2 \lambda_k + \sigma_t^2} c_k(t) \ . \tag{84}$$

For the orthogonal space component $\mathbf{y}^\perp(t)$, it will stay in the orthogonal space $\mathbf{U}^\perp$, and more specifically the 1D space spanned by the initial $\mathbf{y}^\perp(t)$—so, when going backward in time, its dynamics is simply a downscaling of $\mathbf{y}^\perp(T)$.

$$\dot{\mathbf{y}}^\perp(t) = \beta_t \left( \frac{1}{\sigma_t^2} - 1 \right) \mathbf{y}^\perp(t) = \beta_t \frac{1 - \sigma_t^2}{\sigma_t^2} \mathbf{y}^\perp(t) = \frac{\beta_t \alpha_t^2}{\sigma_t^2} \mathbf{y}^\perp(t) \ . \tag{85}$$

Combining these two results and solving the ODEs in the usual way, we have the trajectory solution

$$\mathbf{y}_t = d(t)\mathbf{y}^\perp(T) + \sum_{k=1}^{r} c_k(t)\mathbf{u}_k \tag{86}$$

$$d(t) = \exp \left( \int_T^t d\tau \frac{\beta_\tau \alpha_\tau^2}{\sigma_\tau^2} \right) \tag{87}$$

$$c_k(t) = c_k(T) \exp \left( \int_T^t d\tau \frac{\beta_\tau \alpha_\tau^2 (1 - \lambda_k)}{\alpha_\tau^2 \lambda_k + \sigma_\tau^2} \right) \ . \tag{88}$$

The initial conditions are

$$c_k(T) = \mathbf{u}_k^T \mathbf{y}_T \tag{89}$$

$$\mathbf{y}^\perp(T) = \mathbf{y}_T - \sum_{k=1}^{r} c_k(T)\mathbf{u}_k \ , \ \mathbf{y}^\perp(T) \in \mathbf{U}^\perp \ . \tag{90}$$

To solve the ODEs, it is helpful to use a particular reparameterization of time. In particular, consider a reparameterization in terms of $\alpha_t$ using the relationship $-\beta_t \alpha_t dt = d\alpha_t$. The integral we must do is

$$\int_T^t d\tau \frac{\beta_\tau \alpha_\tau^2 (1 - \lambda_k)}{\alpha_\tau^2 \lambda_k + \sigma_\tau^2} = \int_T^t d\tau \frac{\beta_\tau \alpha_\tau^2 (1 - \lambda_k)}{1 + \alpha_\tau^2 (\lambda_k - 1)} \tag{91}$$

$$= \int_{\alpha_T}^{\alpha_t} d\alpha_\tau \frac{\alpha_\tau (\lambda_k - 1)}{1 + \alpha_\tau^2 (\lambda_k - 1)} \tag{92}$$

$$= \frac{1}{2} \log(1 + \alpha_\tau^2 (\lambda_k - 1)) \Big|_{\alpha_T}^{\alpha_t} \tag{93}$$

$$= \frac{1}{2} \log \left( \frac{1 + (\lambda_k - 1)\alpha_t^2}{1 + (\lambda_k - 1)\alpha_T^2} \right) \ . \tag{94}$$

Note that taking $\lambda_k = 0$ gives us the solution to dynamics in the directions orthogonal to the manifold. We have

$$c_k(t) = c_k(T) \sqrt{\frac{1 + (\lambda_k - 1)\alpha_t^2}{1 + (\lambda_k - 1)\alpha_T^2}} \tag{95}$$

$$d(t) = \sqrt{\frac{1 - \alpha_t^2}{1 - \alpha_T^2}} \ . \tag{96}$$

The time derivatives of these coefficients are

$$\dot{c}_k(t) = c_k(T) \frac{-(\lambda_k - 1)\alpha_t^2 \beta_t}{\sqrt{(1 + (\lambda_k - 1)\alpha_T^2)(1 + (\lambda_k - 1)\alpha_t^2)}} \tag{97}$$

$$\dot{d}(t) = \frac{\alpha_t^2 \beta_t}{\sqrt{(1 - \alpha_T^2)(1 - \alpha_t^2)}} \ . \tag{98}$$

Finally, we can write out the explicit solution for the trajectory $\mathbf{x}_t$:

$$\mathbf{x}_t = \alpha_t \mu + d(t)\mathbf{y}^\perp(T) + \sum_{k=1}^{r} c_k(t)\mathbf{u}_k \ . \tag{99}$$

We can see that there are three terms: 1) $\alpha_t \mu$, an increasing term that scales up to the mean $\mu$ of the distribution; 2) $d(t)\mathbf{y}^\perp(T)$, a decaying term downscaling the residual part of the initial noise vector, which is orthogonal to the data manifold; and 3) the $c_k(t)\mathbf{u}_k$ sum, each term of which has independent dynamics.

We also now have the analytical solution for the projected outcome:

$$\begin{aligned}
\hat{\mathbf{x}}_0(\mathbf{x}_t) - \mu &= \frac{1}{\alpha_t}\mathbf{U}\tilde{\mathbf{\Lambda}}_t\mathbf{U}^T(\mathbf{x}_t - \alpha_t\mu) \\
&= \sum_{k=1}^{r} c_k(t)\frac{\alpha_t \lambda_k}{\alpha_t^2 \lambda_k + \sigma_t^2}\mathbf{u}_k \\
&= \sum_{k=1}^{r} c_k(T)\frac{\alpha_t \lambda_k}{\sqrt{(\alpha_t^2 \lambda_k + \sigma_t^2)(\alpha_T^2 \lambda_k + \sigma_T^2)}}\mathbf{u}_k \ .
\end{aligned} \tag{100}$$

### D.4 Derivation of rotational dynamics

In this section, we derive various results quantifying how reverse diffusion trajectories are rotation-like. In particular, under certain assumptions, we will show that the dynamics of the state $\mathbf{x}_t$ looks like a rotation within a 2D plane spanned by $\mathbf{x}_0$ (the reverse diffusion endpoint) and $\mathbf{x}_T$ (the initial noise). We will derive the formula by assuming that the training set consists of a single Gaussian mode, but will explain why this assumption may not be strictly necessary.

#### D.4.1 Derivation of rotation formula and correction terms

Assume that reverse diffusion begins at time $T$ with $\alpha_T \approx 0$, and ends at time $t = 0$ with $\alpha_0 = 1$. Using our exact solution for $\mathbf{x}_t$ (Eq. 15), at some intermediate time $t$ we have that

$$\mathbf{x}_t = \alpha_t \boldsymbol{\mu} + \sqrt{\frac{1 - \alpha_t^2}{1 - \alpha_T^2}} \mathbf{y}^\perp(T) + \sum_{k=1}^{r} \sqrt{\frac{1 + (\lambda_k - 1)\alpha_t^2}{1 + (\lambda_k - 1)\alpha_T^2}} c_k(T) \boldsymbol{u}_k \ . \tag{101}$$

It is also true, by substituting $t = 0$ and $t = T$, that

$$\boldsymbol{\mu} = \mathbf{x}_0 - \sum_{k=1}^{r} \sqrt{\frac{\lambda_k}{1 + (\lambda_k - 1)\alpha_T^2}} c_k(T) \boldsymbol{u}_k$$

$$\mathbf{y}^\perp(T) = \mathbf{x}_T - \alpha_T \boldsymbol{\mu} - \sum_{k=1}^{r} c_k(T) \boldsymbol{u}_k \ . \tag{102}$$

Using these two equations, we can rewrite Eq. 101 as

$$\mathbf{x}_t = \alpha_t \boldsymbol{\mu} + \sqrt{\frac{1 - \alpha_t^2}{1 - \alpha_T^2}} \left[ \mathbf{x}_T - \alpha_T \boldsymbol{\mu} - \sum_{k=1}^{r} c_k(T) \boldsymbol{u}_k \right] + \sum_{k=1}^{r} \sqrt{\frac{1 + (\lambda_k - 1)\alpha_t^2}{1 + (\lambda_k - 1)\alpha_T^2}} c_k(T) \boldsymbol{u}_k$$

$$= \left[ \alpha_t - \alpha_T \sqrt{\frac{1 - \alpha_t^2}{1 - \alpha_T^2}} \right] \boldsymbol{\mu} + \sqrt{\frac{1 - \alpha_t^2}{1 - \alpha_T^2}} \mathbf{x}_T + \sum_{k=1}^{r} \left\{ \sqrt{\frac{1 + (\lambda_k - 1)\alpha_t^2}{1 + (\lambda_k - 1)\alpha_T^2}} - \sqrt{\frac{1 - \alpha_t^2}{1 - \alpha_T^2}} \right\} c_k(T) \boldsymbol{u}_k \tag{103}$$

$$= \left[ \alpha_t - \alpha_T \sqrt{\frac{1 - \alpha_t^2}{1 - \alpha_T^2}} \right] \mathbf{x}_0 + \sqrt{\frac{1 - \alpha_t^2}{1 - \alpha_T^2}} \mathbf{x}_T + \boldsymbol{R}_t$$

where the remainder term $\boldsymbol{R}_t$ is equal to

$$\boldsymbol{R}_t = \sum_{k=1}^{r} \left\{ \sqrt{\frac{1 + (\lambda_k - 1)\alpha_t^2}{1 + (\lambda_k - 1)\alpha_T^2}} - \sqrt{\frac{1 - \alpha_t^2}{1 - \alpha_T^2}} - \left[ \alpha_t - \alpha_T \sqrt{\frac{1 - \alpha_t^2}{1 - \alpha_T^2}} \right] \sqrt{\frac{\lambda_k}{1 + (\lambda_k - 1)\alpha_T^2}} \right\} c_k(T) \boldsymbol{u}_k \ . $$

The expression simplifies somewhat if we take $\alpha_T \approx 0$. Doing so, we obtain the equation seen in the main text:

$$\mathbf{x}_t \approx \alpha_t \mathbf{x}_0 + \sqrt{1 - \alpha_t^2} \, \mathbf{x}_T + \sum_{k=1}^{r} \left\{ \sqrt{\sigma_t^2 + \lambda_k \alpha_t^2} - \alpha_t \sqrt{\lambda_k} - \sigma_t \right\} c_k(T) \boldsymbol{u}_k \ . $$

Let's examine the correction terms more closely. Define the function

$$J(\alpha_t; \lambda) := \sqrt{\sigma_t^2 + \lambda \alpha_t^2} - \alpha_t \sqrt{\lambda} - \sigma_t$$

$$= \sqrt{1 + (\lambda - 1)\alpha_t^2} - \alpha_t \sqrt{\lambda} - \sqrt{1 - \alpha_t^2} \ . \tag{104}$$

Note that we can rewrite $J$ as

$$J(\alpha_t; \lambda) = \frac{\left( \sqrt{\sigma_t^2 + \lambda \alpha_t^2} - \alpha_t \sqrt{\lambda} - \sigma_t \right) \left( \sqrt{\sigma_t^2 + \lambda \alpha_t^2} + \alpha_t \sqrt{\lambda} + \sigma_t \right)}{\sqrt{\sigma_t^2 + \lambda_k \alpha_t^2} + \alpha_t \sqrt{\lambda} + \sigma_t}$$

$$= -2 \frac{\sigma_t \alpha_t \sqrt{\lambda}}{\sqrt{\sigma_t^2 + \lambda \alpha_t^2} + \alpha_t \sqrt{\lambda} + \sigma_t} \ . \tag{105}$$

From this, it is immediately clear that the correction term is not completely arbitrary. First, $J$ is always negative. Second, its time course is bowl-shaped: it begins close to zero (since $\alpha_T \approx 0$), becomes more negative, then ends close to zero (since $\sigma_0 \approx 0$). It achieves its most negative value roughly when $\sigma_t$ and $\alpha_t \sqrt{\lambda}$ are comparable, i.e. when

$$\alpha_t \approx \sqrt{\frac{1}{\lambda + 1}} \ , \tag{106}$$

in which case

$$J(\alpha_t; \lambda) \approx -2 \left(1 - \frac{\sqrt{2}}{2}\right) \sqrt{\frac{\lambda}{\lambda + 1}} \geq -2 \left(1 - \frac{\sqrt{2}}{2}\right) \approx -0.586 \ . \tag{107}$$

Notice that $|J| < 1$, regardless of $\lambda$.

Although we derived this formula by assuming a single Gaussian mode, its form does not actually depend on any properties of the mode. This suggests that, as long as the score function landscape looks *locally* Gaussian, the formula may still be applicable. For example, suppose the learned image distribution is a Gaussian mixture. Even though the mean and the covariance of the nearest mode—the one which we expect to dominate the score function—may regularly change throughout reverse diffusion, even in a discontinuous way, the rotation equation should stay the same.

### D.4.2 Low-rank image distribution sufficient for small correction terms

Suppose that the rank $r$ of the covariance matrix $\boldsymbol{\Sigma}$ is much less than $D$, the dimensionality of state space. The error in the rotation formula is

$$\left\| \mathbf{x}_t - \alpha_t \mathbf{x}_0 - \sqrt{1 - \alpha_t^2} \ \mathbf{x}_T \right\|_2^2 = \sum_{k=1}^{r} J(\alpha_t; \lambda_k)^2 c_k(T)^2 \ . \tag{108}$$

Recall that $c_k(T)$ is the coefficient of the original noise seed $\mathbf{x}_T \sim \mathcal{N}(\mathbf{0}, \boldsymbol{I})$ along the direction $\boldsymbol{u}_k$. Assuming $D$ is large, the norm of the noise seed is approximately 1. Since there is a priori no relationship between $\mathbf{x}_T$ and $\boldsymbol{u}_k$, we expect that $\mathbf{x}_T \cdot \boldsymbol{u}_k \approx 1/\sqrt{D}$. (Suppose we express $\mathbf{x}_T$ in terms of a set of $D$ orthonormal basis vectors. Given that its norm is 1, and that it has no special relationship with any basis vector, the overlap between $\mathbf{x}_T$ and each vector must be about $1/\sqrt{D}$.) The error becomes

$$\left\| \mathbf{x}_t - \alpha_t \mathbf{x}_0 - \sqrt{1 - \alpha_t^2} \ \mathbf{x}_T \right\|_2^2 \approx \sum_{k=1}^{r} J(\alpha_t; \lambda_k)^2 \frac{1}{D} \leq \sum_{k=1}^{r} 4 \left(1 - \frac{\sqrt{2}}{2}\right)^2 \frac{1}{D} = 4 \left(1 - \frac{\sqrt{2}}{2}\right)^2 \frac{r}{D}$$

where we have used the bound from the previous subsection. Since $r \ll D$, this error is small.

It is worth noting, however, that the $r \ll D$ assumption is not *necessary* for the rotation formula correction terms to be negligible. Another case in which this is true is when the image distribution is isotropic, i.e. $\lambda_k = \lambda$ for all $k$. Then the error is

$$\left\| \mathbf{x}_t - \alpha_t \mathbf{x}_0 - \sqrt{1 - \alpha_t^2} \ \mathbf{x}_T \right\|_2^2 \approx 4 \left(1 - \frac{\sqrt{2}}{2}\right)^2 \frac{\lambda}{\lambda + 1} \frac{r}{D} < \frac{\lambda}{\lambda + 1} \ .$$

This is somewhat smaller than the typical scale of $\mathbf{x}_t$, since $\|\mathbf{x}_t\|_2^2$ remains roughly between 1 and $\lambda$.

### D.4.3 Can the rotation formula be used to predict the trajectory endpoint?

Naively, since any two vectors are mathematically sufficient to define a plane, the rotation plane should be completely determined from the first two steps—or if not the first two steps, one might naively expect the first *several* steps to be sufficient. In particular, since

$$\mathbf{x}_t \approx \alpha_t \mathbf{x}_0 + \sqrt{1 - \alpha_t^2} \mathbf{x}_T + \boldsymbol{R}_t \ , \tag{109}$$

where $\boldsymbol{R}_t$ is the correction term we derived earlier, we can approximate $\mathbf{x}_0$ as

$$\mathbf{x}_0 \approx \frac{\mathbf{x}_t - \sqrt{1 - \alpha_t^2} \ \mathbf{x}_T}{\alpha_t} \ . \tag{110}$$

The problem is that the correction term along each feature direction is 'large' until that feature has been 'committed to'. Concretely, the correction term along direction $\boldsymbol{u}_k$ is proportional to

$$\frac{J(\alpha_t; \lambda_k)}{\alpha_t} = -2\frac{\sigma_t\sqrt{\lambda}}{\sqrt{\sigma_t^2 + \lambda\alpha_t^2} + \alpha_t\sqrt{\lambda} + \sigma_t} \ . \tag{111}$$

This function has saturating behavior, and remains high (in the sense of being $\sim \sqrt{\lambda_k}$) until around the time when $\alpha_t\sqrt{\lambda_k} \approx \sqrt{1 - \alpha_t^2}$. But consistent with our other results (see Figure 3 and the associated formulas), this is roughly around the time of 'feature commitment', or more specifically when the sigmoid-shaped coefficients of $\hat{\mathbf{x}}_0$ begin to transition to their final value. So the rotation formula only becomes useful for determining the endpoint after enough time has passed that one is sufficiently close to the endpoint.

Using multiple $\mathbf{x}_t$ does not help reduce the error in applying the rotation formula, since the error equation above is monotonic in time. In other words, averaging over multiple recent $\mathbf{x}_t$ is *strictly worse* than just using the rotation formula and the most recent (i.e. greatest number of reverse diffusion time steps) $\mathbf{x}_t$.

As a final note: since we have the form of the correction terms, why not use that as additional information? We *could* do this, but this only works for the Gaussian case, where we already have access to the full solution! And knowing these terms at all times is also roughly equivalent to knowing the full trajectory. So in summary, viewing reverse diffusion trajectories as 2D 'rotations' is a useful geometric picture, but it is less quantitatively useful than the full analytical solution to the Gaussian model, e.g. for accelerating sampling.

### D.4.4 Rotation-like dynamics beyond the Gaussian model

In the more general non-Gaussian setting, there is an alternative line of evidence that reverse diffusion dynamics are rotation-like.

**Derivation from EDM formulation with state scaling** We can express the PF-ODE in EDM framework (Eq.2,) using the ideal denoiser (Eq.4)

$$\frac{d\mathbf{x}_t}{dt} = -\dot{\sigma}_t\sigma_t\mathbf{s}(\mathbf{x}, \sigma_t) \tag{112}$$

$$= -\frac{\dot{\sigma}_t}{\sigma_t}(\mathbf{D}(\mathbf{x}_t, \sigma_t) - \mathbf{x}_t) \tag{113}$$

$$= -\frac{d}{dt}(\ln\sigma_t)(\mathbf{D}(\mathbf{x}_t, \sigma_t) - \mathbf{x}_t) \tag{114}$$

When an arbitrary time-dependent state scaling function $\alpha_t$ is introduced, we can substitute $\mathbf{x}_t \mapsto \mathbf{x}_t/\alpha_t$, $\sigma_t \mapsto \sigma_t/\alpha_t$, which yields the equation for the PF-ODE with the scaled states.

$$\frac{d}{dt}\left(\frac{\mathbf{x}_t}{\alpha_t}\right) = -\frac{d}{dt}\ln\left(\frac{\sigma_t}{\alpha_t}\right)\left(\mathbf{D}\left(\frac{\mathbf{x}_t}{\alpha_t}, \frac{\sigma_t}{\alpha_t}\right) - \frac{\mathbf{x}_t}{\alpha_t}\right) \tag{115}$$

When we assume the denoiser target $\mathbf{D}(\frac{\mathbf{x}_t}{\alpha_t}, \frac{\sigma_t}{\alpha_t}) = \mathbf{D}$ is fixed, then this equation can be solved directly,

$$\frac{d}{dt}\left(\frac{\mathbf{x}_t}{\alpha_t}\right) = \frac{d}{dt}\ln\left(\frac{\sigma_t}{\alpha_t}\right)\left(\frac{\mathbf{x}_t}{\alpha_t} - \mathbf{D}\right) \tag{116}$$

$$\frac{\mathbf{x}_t}{\alpha_t} - \mathbf{D} = \exp\left(\int_{t_0}^t d\ln\frac{\sigma_t}{\alpha_t}\right)\left(\frac{\mathbf{x}_{t_0}}{\alpha_{t_0}} - \mathbf{D}\right) \tag{117}$$

$$= \exp\left(\ln\frac{\sigma_t}{\alpha_t} - \ln\frac{\sigma_{t_0}}{\alpha_{t_0}}\right)\left(\frac{\mathbf{x}_{t_0}}{\alpha_{t_0}} - \mathbf{D}\right) \tag{118}$$

$$= \frac{\sigma_t\alpha_{t_0}}{\alpha_t\sigma_{t_0}}\left(\frac{\mathbf{x}_{t_0}}{\alpha_{t_0}} - \mathbf{D}\right) \tag{119}$$

where $t_0$ is a reference time where we start solving this. This solution shows an invariant quantity over the trajectory

$$\frac{\mathbf{x}_t - \alpha_t \mathbf{D}}{\sigma_t} = \frac{\mathbf{x}_{t_0} - \alpha_{t_0} \mathbf{D}}{\sigma_{t_0}} =: \text{Const} \tag{120}$$

$$\mathbf{x}_t = \alpha_t \mathbf{D} + \sigma_t \text{Const} \tag{121}$$

In the specific case of VP-SDE, the scaling term satisfies $\alpha_t^2 + \sigma_t^2 = 1$, then

$$\mathbf{x}_t = \alpha_t \mathbf{D} + \sqrt{1 - \alpha_t^2} \text{Const} \tag{122}$$

This means the solution of $\mathbf{x}_t$ can be interpreted as a spherical interpolation between the hypothetical initial state Const and denoiser state $\mathbf{D}$. We used the word rotation in a very loose sense: this trajectory will be an arc on spherical surface when $\mathbf{D}$ and const have the same norms and orthogonal to each other. These criteria are not always true in actual diffusion trajectories.

**Derivation from VP-ODE formulation** Similarly, we can also re-express the VP-ODE as a rotation. Using the formula of the projected outcome

$$\hat{\mathbf{x}}_0(\mathbf{x}) = \frac{\mathbf{x} + \sigma_t^2 \nabla_{\mathbf{x}} \log p(\mathbf{x}, t)}{\alpha_t} \tag{123}$$

we can write the probability flow ODE (Eq. 70) as

$$\dot{\mathbf{x}} = -\beta(t)\mathbf{x} - \frac{1}{2}g^2(t)\nabla_{\mathbf{x}} \log p(\mathbf{x}, t) = -\beta(t)\mathbf{x} - \frac{1}{2}\frac{g^2(t)}{\sigma_t^2}\left[\alpha_t \hat{\mathbf{x}}_0(\mathbf{x}) - \mathbf{x}\right] . \tag{124}$$

Notice that

$$\alpha(t) = \exp\left(-\int_0^t \beta(\tau)d\tau\right) \qquad \frac{d}{dt}\alpha(t) = -\beta(t)\alpha(t) , \tag{125}$$

which allows us to write

$$\frac{d}{dt}\left(\frac{\mathbf{x}_t}{\alpha_t}\right) = \frac{\dot{\mathbf{x}}_t}{\alpha_t} - \frac{\dot{\alpha}_t \mathbf{x}_t}{\alpha_t^2} = \frac{\dot{\mathbf{x}}_t}{\alpha_t} + \frac{\beta_t \mathbf{x}_t}{\alpha_t} . \tag{126}$$

Equivalently,

$$\frac{d}{dt}\left(\frac{\mathbf{x}_t}{\alpha_t}\right) = -\frac{1}{2}\frac{g^2(t)}{\sigma_t^2}\left(\hat{\mathbf{x}}_0(\mathbf{x}_t) - \frac{\mathbf{x}_t}{\alpha_t}\right) = -\frac{\beta_t}{\sigma_t^2}\left(\hat{\mathbf{x}}_0(\mathbf{x}_t) - \frac{\mathbf{x}_t}{\alpha_t}\right) . \tag{127}$$

From this, we can see that the quantity $\mathbf{x}_t/\alpha_t$, i.e. the state scaled by the signal scale, is isotropically attracted towards the moving target $\hat{\mathbf{x}}_0(\mathbf{x}_t)$ at a rate determined by $\frac{1}{2}\frac{g^2(t)}{\sigma_t^2}$.

Suppose that the endpoint estimates $\hat{\mathbf{x}}_0$ change slowly compared to the state $\mathbf{x}_t$; we will show that this gives us rotation-like dynamics.

First, note that we can evaluate the integral

$$\int_T^t \frac{\beta_t}{\sigma_t^2} \, dt = \int_T^t \frac{\beta_t}{1 - \alpha_t^2} \, dt \tag{128}$$

using a change of variables $\alpha := \alpha_t$ with $d\alpha = -\beta\alpha \, dt$. The integral becomes

$$-\int_{\alpha_T}^{\alpha_t} \frac{d\alpha}{\alpha(1 - \alpha^2)} = \log \frac{\sqrt{1 - \alpha^2}}{\alpha}\bigg|_{\alpha_T}^{\alpha_t} . \tag{129}$$

Using this integral, we can find that the solution to Eq. 127 under the assumption that $\hat{\mathbf{x}}_0$ remains constant is

$$
\log\left(\frac{\frac{\mathbf{x}_t}{\alpha_t} - \hat{\mathbf{x}}_0}{\frac{\mathbf{x}_T}{\alpha_T} - \hat{\mathbf{x}}_0}\right) = \log\left(\frac{\sqrt{1 - \alpha_t^2}}{\alpha_t}\frac{\alpha_T}{\sqrt{1 - \alpha_T^2}}\right)
$$
$$
\implies \quad \frac{\mathbf{x}_t - \alpha_t\hat{\mathbf{x}}_0}{\sqrt{1 - \alpha_t^2}} = \frac{\mathbf{x}_T - \alpha_T\hat{\mathbf{x}}_0}{\sqrt{1 - \alpha_T^2}} \ . \tag{130}
$$

Interestingly, this indicates that there is a conserved quantity

$$
\frac{\mathbf{x}_t - \alpha_t\hat{\mathbf{x}}_0}{\sqrt{1 - \alpha_t^2}} = \text{const.} \tag{131}
$$

along the reverse diffusion trajectory, under this approximation. Since $\alpha_T \approx 0$, $\mathbf{x}_T \approx \text{const.}$, i.e. the value of the constant roughly matches the initial noise seed $\mathbf{x}_T$. Given any $\hat{\mathbf{x}}_0$ the solution to the ODE at time $t$ can be written as

$$
\mathbf{x}_t = \alpha_t\hat{\mathbf{x}}_0 + \sqrt{1 - \alpha_t^2}\,\text{const.} \approx \alpha_t\hat{\mathbf{x}}_0 + \sqrt{1 - \alpha_t^2}\,\mathbf{x}_T \ . \tag{132}
$$

In words: through a rotation, $\mathbf{x}_t$ interpolates between $\hat{\mathbf{x}}_0$ and const. This solution paints the picture that the state is constantly rotating towards the estimated outcome $\hat{\mathbf{x}}_0$, with Eq. 132 describing that hypothetical trajectory's shape. But as the target $\hat{\mathbf{x}}_0$ is moving, the actual trajectory will be similar to Eq. 132 only on short time scales, and not on longer time scales. This idea is visualized by the circular dashed curves in Fig. 4.

Beyond the constant $\hat{\mathbf{x}}_0$ approximation, there will be some correction terms to the above rotation formula, whose precise form depends on the (generally not Gaussian) score function.

### D.5 Score function of general Gaussian mixture models

Let

$$q(\mathbf{x}) = \sum_i \pi_i \mathcal{N}(\mathbf{x}; \mu_i, \mathbf{\Sigma}_i) \tag{133}$$

be a Gaussian mixture distribution, where the $\pi_i$ are mixture weights, $\mu_i$ is the $i$-th mean, and $\mathbf{\Sigma}_i$ is the $i$-th covariance matrix. The score function for this distribution is

$$
\begin{aligned}
\nabla_{\mathbf{x}} \log q(\mathbf{x}) &= \frac{\sum_i \pi_i \nabla_{\mathbf{x}} \mathcal{N}(\mathbf{x}; \mu_i, \mathbf{\Sigma}_i)}{q(\mathbf{x})} \\
&= \sum_i -\Sigma_i^{-1}(\mathbf{x} - \mu_i) \frac{\pi_i \mathcal{N}(\mathbf{x}; \mu_i, \mathbf{\Sigma}_i)}{q(\mathbf{x})} \\
&= \sum_i \frac{\pi_i \mathcal{N}(\mathbf{x}; \mu_i, \mathbf{\Sigma}_i)}{q(\mathbf{x})} \nabla_{\mathbf{x}} \log \mathcal{N}(\mathbf{x}; \mu_i, \mathbf{\Sigma}_i) \\
&= \sum_i w_i(\mathbf{x}) \nabla_{\mathbf{x}} \log \mathcal{N}(\mathbf{x}; \mu_i, \mathbf{\Sigma}_i)
\end{aligned}
\tag{134}
$$

where we have defined the mixing weights

$$w_i(\mathbf{x}) := \frac{\pi_i \mathcal{N}(\mathbf{x}; \mu_i, \mathbf{\Sigma}_i)}{q(\mathbf{x})} . \tag{135}$$

Thus, the score of the Gaussian mixture is a weighted mixture of the score fields of each of the individual Gaussians.

In the context of diffusion, we are interested in the *time-dependent* score function. Given a Gaussian mixture initial condition, the end result of the VP-SDE forward process will also be a Gaussian mixture:

$$p_t(\mathbf{x}) = \sum_i \pi_i \mathcal{N}(\mathbf{x}; \mu_i, \sigma_t^2 \mathbf{I} + \mathbf{\Sigma}_i) . \tag{136}$$

The corresponding time-dependent score is

$$
\begin{aligned}
\mathbf{s}(\mathbf{x}, t) &= \nabla_{\mathbf{x}} \log p_t(\mathbf{x}) \\
&= \sum_i -(\sigma_t^2 \mathbf{I} + \mathbf{\Sigma}_i)^{-1}(\mathbf{x} - \mu_i) \frac{\pi_i \mathcal{N}(\mathbf{x}; \mu_i, \sigma_t^2 \mathbf{I} + \mathbf{\Sigma}_i)}{p_t(\mathbf{x})} \\
&= \sum_i -(\sigma_t^2 \mathbf{I} + \mathbf{\Sigma}_i)^{-1}(\mathbf{x} - \mu_i) w_i(\mathbf{x}, t) .
\end{aligned}
\tag{137}
$$

Note that we have a formula for $(\sigma_t^2 \mathbf{I} + \mathbf{\Sigma}_i)^{-1}$ (derived using the Woodbury matrix inversion identity; see Eq.5) in terms of the (compact) SVD of $\mathbf{\Sigma}_i$. We can use it to write

$$\mathbf{s}(\mathbf{x}, t) = \frac{1}{\sigma_t^2} \sum_i -(\mathbf{I} - \mathbf{U}_i \tilde{\mathbf{\Lambda}}_{i_t} \mathbf{U}_i^T)(\mathbf{x} - \mu_i) w_i(\mathbf{x}, t) \tag{138}$$

where

$$\mathbf{\Sigma}_i = \mathbf{U}_i \mathbf{\Lambda}_i \mathbf{U}_i^T \qquad\qquad \tilde{\mathbf{\Lambda}}_t := \mathrm{diag}\left[\frac{\lambda_k}{\lambda_k + \sigma_t^2}\right] . \tag{139}$$

This representation of the score function is numerically convenient, since (once the SVDs of each covariance matrix have been obtained), it can be evaluated using a relatively small number of matrix multiplications, which are cheaper than the covariance matrix inversions that a naive implementation of the Gaussian mixture score function would require.

We used this formula (Eq.138) and an off-the-shelf ODE solver to simulate the reverse diffusion trajectory of a 10-mode Gaussian mixture score model (Fig. 17).

### D.6 Score function of Gaussian mixture with identical and isotropic covariance

Assume each Gaussian mode has covariance $\boldsymbol{\Sigma}_i = \sigma^2 \mathbf{I}$ and that every mixture weight is the same (i.e. $\pi_i = \pi_j, \forall i, j$). Then the score for this kind of specific Gaussian mixture is

$$
\begin{aligned}
\nabla_{\mathbf{x}} \log q(\mathbf{x}) &= \frac{\sum_i \pi_i \nabla_{\mathbf{x}} \mathcal{N}(\mathbf{x}; \mu_i, \sigma^2 \mathbf{I})}{\sum_i \pi_i \mathcal{N}(\mathbf{x}; \mu_i, \sigma^2 \mathbf{I})} \\
&= \frac{\sum_i -\frac{1}{\sigma^2}(\mathbf{x} - \mu_i) \exp\left(-\frac{1}{2\sigma^2}\|\mathbf{x} - \mu_i\|_2^2\right)}{\sum_i \exp\left(-\frac{1}{2\sigma^2}\|\mathbf{x} - \mu_i\|_2^2\right)} \\
&= \frac{1}{\sigma^2} \sum_i w_i(\mathbf{x})(\mu_i - \mathbf{x}) ,
\end{aligned}
\tag{140}
$$

where the weight $w_i(\mathbf{x})$ is a softmax of the negative squared distance to all the means, with $\sigma^2$ functioning as a temperature parameter:

$$
w_i(\mathbf{x}) = Softmax(\{-\frac{1}{2\sigma^2}\|\mathbf{x} - \mu_i\|_2^2\}) = \frac{\exp\left(-\frac{1}{2\sigma^2}\|\mathbf{x} - \mu_i\|_2^2\right)}{\sum_i \exp\left(-\frac{1}{2\sigma^2}\|\mathbf{x} - \mu_i\|_2^2\right)} .
\tag{141}
$$

Since the weights $w_i(\mathbf{x})$ sum to 1, we can also write the score function in the suggestive form

$$
\nabla_{\mathbf{x}} \log q(\mathbf{x}) = \frac{(\sum_i w_i(\mathbf{x})\mu_i) - \mathbf{x}}{\sigma^2} .
\tag{142}
$$

This has a form analogous to the score of a single Gaussian mode—but instead of $\mathbf{x}$ being 'attracted' towards a single mean $\mu$, it is attracted towards a weighted combination of all of the means, with modes closer to the state $\mathbf{x}$ being more highly weighted.

### D.7 Score function of exact (delta mixture) score model

A particularly interesting special case of the Gaussian mixture model is the delta mixture model used in the main text, whose components are vanishing-width Gaussians centered on the training images. In particular, consider a dataset $\{\mathbf{y}_i\}$ with $i = 1, ..., N$, so that the starting distribution is

$$
p(\mathbf{x}) = \frac{1}{N} \sum_i \delta(\mathbf{x} - \mathbf{y}_i) .
\tag{143}
$$

At time $t$, the marginal distribution will be a Gaussian mixture

$$
p_t(\mathbf{x}_t) = \frac{1}{N} \sum_i \mathcal{N}(\mathbf{x}_t; \mathbf{y}_i, \sigma_t^2 \mathbf{I}) .
\tag{144}
$$

Then using the Eq. 140 above we have

$$
\begin{aligned}
s(\mathbf{x}_t, t) = \nabla \log p_t(\mathbf{x}_t) &= \frac{1}{\sigma_t^2} \sum_i w_i(\mathbf{x}_t)(\mathbf{y}_i - \mathbf{x}_t) \\
&= \frac{1}{\sigma_t^2}\left[-\mathbf{x}_t + \sum_i w_i(\mathbf{x}_t)\mathbf{y}_i\right] \\
&= \frac{1}{\sigma_t^2}\left[-\mathbf{x}_t + \sum_i Softmax(\{-\frac{1}{2\sigma_t^2}\|\mathbf{y}_i - \mathbf{x}_t\|^2\})\mathbf{y}_i\right] .
\end{aligned}
\tag{145}
$$

The endpoint estimate of the distribution is

$$
\begin{aligned}
\hat{\mathbf{x}}_0(\mathbf{x}_t) &= \frac{\mathbf{x}_t + \sigma_t^2 \nabla \log p(\mathbf{x}_t)}{} \\
&= \sum_i Softmax(\{-\frac{1}{2\sigma_t^2}\|\mathbf{y}_i - \mathbf{x}_t\|^2\})\mathbf{y}_i \\
&= \sum_i w_i(\mathbf{x}_t)\mathbf{y}_i .
\end{aligned}
\tag{146}
$$

Thus, the endpoint estimate is a weighted average of training data, with the softmax of negative squared distance as weights and $\sigma_t^2$ as a temperature parameter.

### D.8 Arguments of the approximation of score field of Gaussian and point cloud

In this section, we will argue from a theoretical perspective, the score field of a bounded point cloud is equivalent to a Gaussian with matching mean and covariance, when the noise level is high and when the query point is far from a bounded point cloud. We are going to use a technique inspired by multi-pole expansion in electrodynamics Griffiths (2005).

**Set up**   Assume we have a set of data points $\{\mathbf{y}_i\}, i = 1...N$, their mean and covariance are defined as

$$\mu = \frac{1}{N} \sum_i \mathbf{y}_i \tag{147}$$

$$\Sigma = \frac{1}{N} \sum_i \mathbf{y}_i \mathbf{y}_i^T - \mu\mu^T \tag{148}$$

these are the first moment and the second central moment of the distribution. Let us assume this point cloud is bounded by a sphere of radius $r$ around $\mu$, i.e. $\|\mathbf{y_i} - \mu\| \leq r$. Since $\sum_j^N \|y_j - \mu\| = N\text{tr}\Sigma$, this bounds the covariance spectrum $\text{tr}\Sigma \leq r^2$. The Gaussian distribution with matching first two moments is $\mathcal{N}(\mu, \Sigma)$. We are going to show that when the noise level $\sigma$ and $\|\mathbf{x} - \mu\|$ are both much larger than the standard deviation of the distribution, the score at noise level $\sigma$ of the original dataset (point cloud) is equivalent to the score of this Gaussian distribution.

**Score expansion for Gaussian distribution**   Consider the Gaussian distribution, at noise level $\sigma$, the distribution is $\mathcal{N}(\mu, \Sigma + \sigma^2 I)$. Then its score can be written as

$$\log p(\mathbf{x}; \sigma) = (\sigma^2 I + \Sigma)^{-1}(\mu - \mathbf{x}) \tag{149}$$

$$= \frac{1}{\sigma^2}(I - U\tilde{\Lambda}_\sigma U^T)(\mu - \mathbf{x}) \tag{150}$$

in which the

$$\tilde{\Lambda}_\sigma = diag\left[\frac{\lambda_k}{\lambda_k + \sigma^2}\right] \tag{151}$$

$$= diag\left[\frac{\lambda_k}{\sigma^2} - \frac{\lambda_k^2}{\sigma^2(\sigma^2 + \lambda_k)}\right] \tag{152}$$

$$= diag\left[\frac{\lambda_k}{\sigma^2} - \frac{\lambda_k^2}{\sigma^4} + \frac{\lambda_k^3}{\sigma^4(\sigma^2 + \lambda_k)}\right] \tag{153}$$

$$= \frac{1}{\sigma^2}\Lambda - \frac{1}{\sigma^4}\Lambda^2 + \frac{1}{\sigma^6}\Lambda^3 - \frac{1}{\sigma^8}\Lambda^4 + ... \tag{154}$$

$$= \frac{1}{\sigma^2}\Lambda - \Delta \tag{155}$$

where the residual term $\Delta$ is the 2nd order term of $\lambda/\sigma^2$, $\Delta \sim (\lambda/\sigma^2)^2$.

Using this expansion, the score field can be expressed as

$$\log p(\mathbf{x}; \sigma) = \frac{1}{\sigma^2}(I - \frac{1}{\sigma^2}U\Lambda U^T + U\Delta U^T)(\mu - \mathbf{x}) \tag{156}$$

$$= \frac{1}{\sigma^2}(\mu - \mathbf{x}) - \frac{1}{\sigma^4}\Sigma(\mu - \mathbf{x}) + \frac{1}{\sigma^2}U\Delta U^T(\mu - \mathbf{x}) \tag{157}$$

$$= \frac{1}{\sigma^2}(\mu - \mathbf{x}) - \frac{1}{\sigma^4}\Sigma(\mu - \mathbf{x}) + \frac{1}{\sigma^6}U\Lambda^2 U^T(\mu - \mathbf{x}) - \frac{1}{\sigma^8}U\Lambda^3 U^T(\mu - \mathbf{x}) + ... \tag{158}$$

$$= \frac{1}{\sigma^2}(\mu - \mathbf{x}) - \frac{1}{\sigma^4}\Sigma(\mu - \mathbf{x}) + \frac{1}{\sigma^6}\Sigma^2(\mu - \mathbf{x}) - \frac{1}{\sigma^8}\Sigma^3(\mu - \mathbf{x}) + ... \tag{159}$$

From this result, we can express the score field of a Gaussian distribution by a series of terms: the first-order term is the isotropic attraction to the mean, the second-order term is the anisotropic term conditioned by the

covariance matrix; the higher-order terms are increasingly anisotropic with higher power of the covariance matrix, i.e. larger condition number. Note that this series is only convergent when $\sigma^2 > \lambda_{max}$. Note also that, this series is always linear with respect to $\mu - \mathbf{x}$, and it doesn't contain any quadratic term of $\mathbf{x}$.

**Score expansion for delta point distribution** Consider the data points, at noise level $\sigma$, the noised distribution is a Gaussian mixture with $N$ components, $q(\mathbf{x}; \sigma) = \frac{1}{N}\sum_i \mathcal{N}(\mathbf{x}; \mathbf{y}_i, \sigma^2 I)$. As we have derived above, the score of a Gaussian mixture with identical variance at each mode can be expressed as

$$\nabla_x \log q(\mathbf{x}; \sigma) = \frac{1}{\sigma^2}\sum_i w_i(\mathbf{x})(\mathbf{y}_i - \mathbf{x}) \tag{160}$$

$$= \frac{1}{\sigma^2}(\sum_i w_i(\mathbf{x})\mathbf{y}_i - \mathbf{x}) \tag{161}$$

$$= \frac{1}{\sigma^2}(\mu - \mathbf{x}) + \frac{1}{\sigma^2}\sum_i w_i(\mathbf{x})(\mathbf{y}_i - \mu) \tag{162}$$

Where the weighting function is

$$w_i(\mathbf{x}) = \frac{\exp\left(-\frac{1}{2\sigma^2}\|\mathbf{x} - \mathbf{y}_i\|_2^2\right)}{\sum_j^N \exp\left(-\frac{1}{2\sigma^2}\|\mathbf{x} - \mathbf{y}_j\|_2^2\right)} \tag{163}$$

Here we rewrite all the distances using the distributional mean as a reference

$$\|\mathbf{x} - \mathbf{y}_i\|_2^2 = \|\mathbf{x} - \mu + \mu - \mathbf{y}_i\|_2^2 \tag{164}$$

$$= \|\mathbf{x} - \mu\|_2^2 + \|\mu - \mathbf{y}_i\|_2^2 + 2(\mathbf{x} - \mu)^T(\mu - \mathbf{y}_i) \tag{165}$$

Multiply the same term $\exp\left(\frac{1}{2\sigma^2}\|\mathbf{x} - \mu\|_2^2\right)$ at numerator and denominator, we get

$$w_i(\mathbf{x}) = \frac{\exp\left(-\frac{1}{2\sigma^2}(\|\mu - \mathbf{y}_i\|_2^2 + 2(\mathbf{x} - \mu)^T(\mu - \mathbf{y}_i))\right)}{\sum_j^N \exp\left(-\frac{1}{2\sigma^2}(\|\mu - \mathbf{y}_j\|_2^2 + 2(\mathbf{x} - \mu)^T(\mu - \mathbf{y}_j))\right)} \tag{166}$$

**Order of terms** Note the identities

$$\sum_j^N (\mu - \mathbf{y}_j) = 0 \tag{167}$$

$$\sum_j^N \|\mu - \mathbf{y}_j\|_2^2 = Ntr\Sigma \tag{168}$$

$$\mathbb{E}_{\mathbf{y}\sim\{\mathbf{y}_i\}}\|\mu - \mathbf{y}\|_2^2 = tr\Sigma \tag{169}$$

So $\|\mu - \mathbf{y}_j\|_2^2$ is on the order of $tr\Lambda$.

The query points are sampled from the noised distribution, which convolves the data points cloud with an isotropic Gaussian, i.e. $\mathbf{x} \sim 1/N \sum_i \mathcal{N}(\mathbf{y}_i, \sigma^2 I)$. This is approximately $\mathcal{N}(\mu, \sigma^2 I)$, so when $\sigma$ is large, $\|\mathbf{x} - \mu\|^2$ is on the order of $d\sigma^2$.

When we assume the query point is sampled from Gaussian with width $\sigma$, we can estimate the norm of cross term.

$$\mathbb{E}_{\mathbf{x}\sim\mathcal{N}(\mu,\sigma^2 I)}[\|(\mathbf{x} - \mu)^T(\mu - \mathbf{y}_i)\|] = \sqrt{\frac{2}{\pi}}\sigma\|\mu - \mathbf{y}_i\| \tag{170}$$

Thus, the cross term $(\mathbf{x} - \mu)^T(\mu - \mathbf{y}_i)$ is on the order of $\sigma\sqrt{tr\Lambda}$.

When the noise scale $\sigma \gg \sqrt{tr\Lambda}$, the cross-term $(\mathbf{x} - \mu)^T(\mu - \mathbf{y}_i)$ dominates $\|\mu - \mathbf{y}_j\|_2^2$ . In this case, when we expand the weighting function $w_i$ by orders of $\frac{r}{\sigma}$, the cross term is one order higher. Thus, on the 1st order of $\frac{r}{\sigma}$ , we get

$$
\begin{aligned}
w_i(\mathbf{x}) &\approx \frac{1 + \frac{1}{\sigma^2}(\mathbf{x} - \mu)^T(\mathbf{y}_i - \mu)}{\sum_j^N 1 + \frac{1}{\sigma^2}(\mathbf{x} - \mu)^T(\mathbf{y}_j - \mu)} \\
&= \frac{1 + \frac{1}{\sigma^2}(\mathbf{x} - \mu)^T(\mathbf{y}_i - \mu)}{N + \frac{1}{\sigma^2}(\mathbf{x} - \mu)^T \sum_j^N (\mathbf{y}_j - \mu)} \\
&= \frac{1}{N}\left(1 + \frac{1}{\sigma^2}(\mathbf{x} - \mu)^T(\mathbf{y}_i - \mu)\right)
\end{aligned}
$$

Using this approximation, the GMM score expansion has the same first-two terms as the Gaussian expansion

$$
\begin{aligned}
\nabla_{\mathbf{x}} \log q(\mathbf{x}; \sigma) &= \frac{1}{\sigma^2}(\mu - \mathbf{x}) + \frac{1}{\sigma^2} \sum_i w_i(\mathbf{x})(\mathbf{y}_i - \mu) \\
&\approx \frac{1}{\sigma^2}(\mu - \mathbf{x}) + \frac{1}{\sigma^4}\frac{1}{N} \sum_i (\mathbf{x} - \mu)^T(\mathbf{y}_i - \mu)(\mathbf{y}_i - \mu) \\
&= \frac{1}{\sigma^2}(\mu - \mathbf{x}) + \frac{1}{\sigma^4}\Sigma(\mathbf{x} - \mu)
\end{aligned}
$$

