# OpenReview forum: "The Unreasonable Effectiveness of Gaussian Score Approximation for Diffusion Models and its Applications"
_TMLR — Accepted by TMLR_

### Review · Reviewer_ii8C · 2024-08-05

**Summary Of Contributions:**

This paper evaluates diffusion models with a score consisting of tractable distributions (Gaussian and Gaussian mixtures). They analyze the Gaussian score model, including characterizing the properties of its sampling trajectories by exactly solving the associated probability flow ordinary differential equation (PF-ODE). They theoretically and empirically support their claim that, in the high-noise regime, the score field of real diffusion models is dominated by its Gaussian approximation by proving that the score function of the (noise-corrupted) point cloud is dominated by the score function of the Gaussian model. They use these findings to speed up image sampling by 15-30% by skipping the initial phase (teleportation).

**Audience:**

Yes

**Claims And Evidence:**

Yes

**Requested Changes:**

* Consider rearranging terms in Eq. 1 so that the marginal likelihood is clearer

\begin{align}
p(\mathbf{x} | \sigma) &= \int p(\mathbf{x} | \mathbf{x}’, \sigma) p(\mathbf{x}’) d\mathbf{x} \newline
&= \int \mathcal{N}(\mathbf{x} | \mathbf{x}’, \sigma^2 \mathbf{I}) p(\mathbf{x}’) d\mathbf{x}.
\end{align}

**Strengths And Weaknesses:**

Strengths:
* The paper is clearly written and of interest to the TMLR audience.
* The proof that the score function of the (noise-corrupted) point cloud is dominated by the score function of the Gaussian model is clear and easy to follow.
* Figure 1 and 6 does a great job of showing the theoretical results that the score field of real diffusion models is dominated by its Gaussian approximation. Figure 14 does a great job of showing the effect or teleportation and how skipping K steps by using the Gaussian approximation has little impact on the final sample.

Weaknesses:
* Several term definitions could be included in Sec. 2 so that the introduction is clear to readers without an understanding of diffusion. For example, unless I missed them, I do not think that $\dot{\sigma}$, $t$, and $\mathbf{y}$ are defined.
* The biggest weakness of the current work is the lack of standard deviations and comparison to other works in Tab. 1. It is unclear how good the improved sampling speed and maintained performance are without a comparison to current state-of-the-art methods.

---

> ### Author Response · Authors · 2024-10-16
> **Thanks for your review!**
>
> We thank the reviewer for their helpful comments and suggestions.
>
> > Several term definitions could be included in Sec. 2 so that the introduction is clear to readers without an understanding of diffusion. For example, unless I missed them, I do not think that $\dot{\sigma}$, $t$, and $\mathbf{y}$ are defined.
>
> Good catch! We will carefully look over the manuscript to ensure all relevant symbols are defined before they are used. One possible source of confusion, which we will be more careful about, is that we sometimes parameterize the reverse process in terms of noise scale (in which case one considers values of $\sigma$), and sometimes parameterize it in terms of time (in which case one considers the function $\sigma_t$ evaluated at time $t$).
>
> $\dot{\sigma}$ is just shorthand for the time derivative of $\sigma_t$, but we will be clearer about this. $\mathbf{y}$ is defined as being a sample from $p_{data}$ in Eq. 3, the first time it is used, but we can be more explicit.
>
> > Consider rearranging terms in Eq. 1 so that the marginal likelihood is clearer
>
> Good suggestion, we will implement it.
>
> > The biggest weakness of the current work is the lack of standard deviations and comparison to other works in Tab. 1. It is unclear how good the improved sampling speed and maintained performance are without a comparison to current state-of-the-art methods.
>
> We are currently running additional benchmarks and will post them soon.

---

### Review · Reviewer_n5h4 · 2024-08-15

**Summary Of Contributions:**

This work provides a theoretical analysis of generative modeling by considering the associated ODE. They discuss both the high noise regime and the low noise regime for the approximation of real diffusion models. They propose methods to accelerate sampling process.

**Audience:**

Yes

**Broader Impact Concerns:**

N.A.

**Claims And Evidence:**

Yes

**Requested Changes:**

If possible, include more baselines for the sampling speed comparison with prior work.

**Strengths And Weaknesses:**

Strengths :The authors apply their findings on the algorithm to speed up the sampling process and evaluate it on several datasets.

---

> ### Author Response · Authors · 2024-10-16
> **Thanks for your review!**
>
> Thanks!
>
> > If possible, include more baselines for the sampling speed comparison with prior work.
>
> We are running additional benchmarks now and will post them soon.

---

### Review · Reviewer_Rrvc · 2024-09-15

**Summary Of Contributions:**

This paper puts forth the claim that in diffusion models, a large portion of the reverse process can be replaced by its Gaussian approximation, whereby the score function at a given noise scale is approximated by the (linear) score function of the Gaussian with mean and covariance matching the noisy data. The paper's contributions towards this claim are the following empirical findings on real image datasets:
1) For most of the reverse process, the Gaussian score explains a large fraction of the posterior variance, and the reverse process with Gaussian score stays close to the true reverse process. Moreover, the resulting samples retain low-frequency features of the true generated images
2) Near the end of the reverse process, the true score is somewhat better approximated by a mixture of low-dimensional Gaussians
3) Neural networks trained on the score matching objective initially seem to converge towards the Gaussian score, but at lower noise scales, at some point they start to deviate. This suggests that in early phases of training, the network is learning low-frequency aspects of the distribution.
4) They use finding 1) to "shortcut" the reverse process and reduce the number of NFEs needed to sample compared to baselines like Karras et al. '22

**Audience:**

Yes

**Broader Impact Concerns:**

There are some broader impact aspects that are inherent to any generative modeling paper, but I'm not so concerned about this here as they are actually working towards more transparent interpretations of the dynamics of the reverse process.

**Claims And Evidence:**

Yes

**Requested Changes:**

- I would be interested to see how the score error achieved by the Gaussian score looks at the noise scale corresponding to sqrt(*median* eigenvalue)
- The accelerated sampling results are somewhat far from current state of the art as the baseline of EDM is a bit dated, but this is maybe not a concern because the teleportation trick might be a useful orthogonal improvement on top of state of the art methods? For instance, instead of speeding up inference for diffusions, can this be used to speed up training for consistency models?
- It would be helpful to include some images generated by using Gaussian score for most of the reverse process and then the best possible Gaussian mixture near the end of the reverse process, just to compare against Figure 8.

**Strengths And Weaknesses:**

### Strengths
The experiments are quite comprehensive and address quite a large number of facets of this phenomenon. Beyond the initial finding that the score is approximately linear for much of the reverse process, they consider other analytical approximations (e.g. the "exact" score of the empirical distribution, Gaussian mixture approximation), they study ways in which failure of the Gaussian approximation can be corrected for with a suitable mixture model approximation (by modulating # components, rank of components, etc.), and they study the training dynamics of denoising networks and provide useful visualizations of the evolution of the score estimate over the course of training.

### Weaknesses
It's unclear how surprising the Gaussian approximation finding is. The "unreasonable effectiveness" aspect of their claim largely stems from the finding that the approximation is good even at noise scales where the distribution is not statistically close to Gaussian, because there are eigendirections of the covariance along which the distribution has not yet mixed. But this would happen even if the data were a product of an arbitrary $(n-1)$-dimensional isotropic distribution and a high variance 1D Gaussian: even if the one bad eigendirection takes much longer to mix, the $(n-1)$-dimensional part mixes much earlier and an overwhelming fraction of the variance is already explained by that part. The authors do consider the mean eigenvalue in their experiments, but at the scales considered in this paper, this is considered to be close to the end of the reverse process, e.g. even under the zoomed-in plots in Figure 6.

Additionally, the finding that the Gaussian model predicts low-frequency features of the samples is perhaps not that surprising as we expect that matching the mean and covariance of any natural image distribution with a Gaussian will result in images that look very roughly like true images. The fact that the trajectories of the true reverse process and Gaussian reverse process is arguably also not that surprising: by Girsanov's theorem we know that score error does not compound exponentially.

---

> ### Author Response · Authors · 2024-10-17
> **Thanks for your review!**
>
> We thank the reviewer for their detailed comments and suggestions, and think their insight has helped us improve the clarity of our paper.
>
> > It's unclear how surprising the Gaussian approximation finding is. The "unreasonable effectiveness" aspect of their claim largely stems from the finding that the approximation is good even at noise scales where the distribution is not statistically close to Gaussian, because there are eigendirections of the covariance along which the distribution has not yet mixed. But this would happen even if the data were a product of an arbitrary $(n-1)$-dimensional isotropic distribution and a high variance 1D Gaussian: even if the one bad eigendirection takes much longer to mix, the $(n-1)$-dimensional part mixes much earlier and an overwhelming fraction of the variance is already explained by that part. The authors do consider the mean eigenvalue in their experiments, but at the scales considered in this paper, this is considered to be close to the end of the reverse process, e.g. even under the zoomed-in plots in Figure 6.
> >
>
> The reviewer is essentially correct and hits upon an extremely important point: one does not expect the Gaussian approximation to well-approximate reverse diffusion sampling from *arbitrary* distributions, but only distributions for which a substantial amount of variance is captured by Gaussian-like directions. From this point of view, the "unreasonable effectiveness" claim reflects an *empirical* property of many real image distributions.
>
> What it means for directions to be 'Gaussian-like' is hard to define precisely, but certain examples are easy to classify, and analyzing them helps build intuition. 'Off-manifold' directions are one interesting example. Under the 'image manifold' hypothesis, the effective dimensionality of some set of images is much lower than the dimensionality of pixel space, and hence most state space directions are 'off-manifold' in the sense that the image distribution to good approximation does not vary along them. The corresponding noise-corrupted image distribution is then approximately Gaussian along these directions.
>
> Consider, on the other hand, a simple example for which our Gaussian approximation may seem quite bad: an equally-weighted mixture of two well-separated, isotropic Gaussians in $D$ dimensions, i.e.,
> $$
> p(\mathbf{x}, t) := \frac 12 \left[ \mathcal{N}(\mathbf{x}; \boldsymbol{\mu},(q^2 + t^2) \mathbf{I}) + \mathcal{N}(\mathbf{x}; -\boldsymbol{\mu}, (q^2 + t^2) \mathbf{I} ) \right] \ .
> $$
> Let $q^2 > 0$ denote the variance of each mode, and $\pm \boldsymbol{\mu}$ denote the centers of each mode. The score function (assuming the EDM parameterization) at noise scale $\sigma = t$ is
> $$
> \mathbf{s}(\mathbf{x}, t) := \frac{1}{q^2 + t^2}\left[ \boldsymbol{\mu} \tanh(\frac{\mathbf{x} \cdot \boldsymbol{\mu}}{q^2 + t^2}) - \mathbf{x}  \right] \ .
> $$
> Suppose that $\Vert \boldsymbol{\mu} \Vert \gg q$. If this is true, the axis along which the two modes are most separated (which is parallel to $\boldsymbol{\mu}$) is the highest-variance direction. Moreover, for all $t > 0$, the sign of $\mathbf{x} \cdot \boldsymbol{\mu}$ alone determines whether the score function points more towards the $+\boldsymbol{\mu}$ mode or the $- \boldsymbol{\mu}$ mode along this axis. The corresponding Gaussian approximation to this score is
> $$
> \mathbf{s}_G(\mathbf{x}, t) := - \left[ (q^2 + t^2) \mathbf{I} + \boldsymbol{\mu} \boldsymbol{\mu}^T \right]^{-1} \mathbf{x} \ ,
> $$
> and always points towards the origin $\mathbf{x} = \mathbf{0}$. Hence, for $\mathbf{x} \neq \mathbf{0}$ it is **antiparallel** to the true score along the direction of highest variance. (On the other hand, the Gaussian approximation is exactly correct for all other state space directions.) At noise scales where one can 'resolve' this high-variance direction from noise (i.e., $t$ comparable to or smaller than $\Vert\boldsymbol{\mu}\Vert$), the Gaussian approximation fails badly for the 'on-manifold' direction.
>
> Our first and second examples are not unrelated. In the mixture model case, if there are *sufficiently many* other directions, the other directions may explain a large fraction of the overall variance of the score, and hence the Gaussian approximation will be good, even though the highest-variance direction is decidedly not Gaussian. This is true more generally early in reverse diffusion, assuming there are many more off-manifold than on-manifold directions. As reverse diffusion proceeds, off-manifold directions explain an ever smaller amount of the overall variance, and the Gaussian approximation eventually ceases to be good.
>
> Since the conceptual point the reviewer brings up is important, we will write a new subsection that addresses it explicitly. In this subsection, we will have a more extensive discussion of when the Gaussian approximation works and when it doesn’t, using examples like the aforementioned Gaussian mixture model to illustrate this point.

---

> > ### Author Response · Authors · 2024-10-17
> > **Response, part 2**
> >
> > (continued)
> >
> > > Additionally, the finding that the Gaussian model predicts low-frequency features of the samples is perhaps not that surprising as we expect that matching the mean and covariance of any natural image distribution with a Gaussian will result in images that look very roughly like true images. The fact that the trajectories of the true reverse process and Gaussian reverse process is arguably also not that surprising: by Girsanov's theorem we know that score error does not compound exponentially.
> > >
> >
> > The reviewer is correct that Girsanov's theorem guarantees that bounded score error produces bounded trajectory errors. We will comment on this connection and cite some relevant papers when we compare the trajectories of different models to those of their Gaussian approximations.

---

### Author Response · Authors · 2024-10-17
**Additional figure and evaluation of analytical teleportation combined with other diffusion samplers (DPM-solver-v3, DPM++, Heun, Uni-PC)**

### Additional evaluation results

In response to the concerns raised by multiple reviewers that there are not enough evaluations regarding our results, and our baseline methods (Heun’s sampler) are somewhat dated, we performed more extensive experiments combining our method with many more recent deterministic sampling methods: `dpm_solver++`, `dpm_solver_v3`, `heun`, `uni_pc_bh1`, `uni_pc_bh2`, with the same pretrained model (EDM).

We basically find results consistent with those stated in the paper across all these samplers, i.e. with the same sampling quality, we can improve the number of neural function evaluations i.e. sampling speed; Further, we found, with a fixed amount of sampling budget, Gaussian teleportation will help improve the sampling quality. We think the intuition is that, as we substitute the easy-to-approximate part with Gaussian analytical solution, the sampler can now spend more steps on the hard-to-approximate, nonlinear score field around the data manifold. By spending the neural function evaluation budget more wisely, we can improve the sample quality in the end.

Specifically, comparing to the recent method DPM-solver-v3 (NeurIPS 2023) which leveraged model-specific statistics, they found “We achieve FIDs of 12.21 (5 NFE), 2.51 (10 NFE) on unconditional CIFAR10”. In comparison, using the teleportation technique, we achieved FID of 9.63 (5 NFE, skip to sigma=20.0) and 2.45 (10 NFE, skip to sigma=40.0) and 2.22 (12 NFE, skip to sigma=20.0). Similar results hold for AFHQv2 dataset.

**Tab. 1 Analytical teleportation combined with DPM-Solver-v3 for CIFAR10**

| NFE / Steps | 5 | 6 | 8 | 10 | 12 | 15 | 20 | 25 |
| --- | --- | --- | --- | --- | --- | --- | --- | --- |
| skip time / sigma |  |  |  |  |  |  |  |  |
| **1.0** | 37.15 | 38.89 | 35.38 | 31.93 | 30.02 | 28.36 | 27.03 | 26.54 |
| **2.5** | 15.21 | 19.65 | 14.77 | 9.91 | 7.82 | 6.36 | 5.40 | 5.12 |
| **5.0** | 11.95 | 13.38 | 8.37 | 4.74 | 3.45 | 2.76 | 2.41 | 2.34 |
| **10.0** | 10.73 | 8.20 | 4.58 | 2.94 | 2.39 | 2.17 | 2.07 | 2.05 |
| **20.0** | 9.63 | 5.78 | 3.07 | 2.52 | 2.22 | 2.12 | 2.07 | 2.06 |
| **40.0** | 9.82 | 5.51 | 2.86 | 2.45 | 2.27 | 2.15 | 2.08 | 2.07 |
| **80.0** | 12.42 | 8.73 | 3.62 | 2.65 | 2.32 | 2.16 | 2.08 | 2.07 |

**Tab. 2 Analytical teleportation combined with DPM-Solver++ for AFHQv2**

| skip_value | 5 | 6 | 8 | 10 | 12 | 15 | 20 | 25 | 30 | 35 | 40 |
| --- | --- | --- | --- | --- | --- | --- | --- | --- | --- | --- | --- |
| **1.0** | 51.81 | 55.00 | 55.66 | 55.00 | 54.75 | 54.49 | 54.16 | 53.93 | 53.78 | 53.67 | 53.60 |
| **2.5** | 18.29 | 18.29 | 17.13 | 15.33 | 14.30 | 13.46 | 12.74 | 12.35 | 12.12 | 11.97 | 11.87 |
| **5.0** | 11.97 | 9.73 | 7.54 | 5.94 | 5.03 | 4.40 | 3.95 | 3.76 | 3.66 | 3.60 | 3.56 |
| **10.0** | 12.96 | 8.32 | 5.01 | 3.68 | 2.99 | 2.54 | 2.30 | 2.23 | 2.20 | 2.18 | 2.17 |
| **20.0** | 12.88 | 7.82 | 4.51 | 3.25 | 2.57 | 2.19 | 2.04 | 1.99 | 1.98 | 1.98 | 1.98 |
| **40.0** | 14.12 | 9.04 | 4.31 | 3.11 | 2.50 | 2.15 | 2.02 | 1.99 | 1.98 | 1.97 | 1.97 |
| **80.0** | 17.66 | 10.65 | 4.18 | 3.12 | 2.61 | 2.27 | 2.10 | 2.05 | 2.03 | 2.02 | 2.01 |

We note that for FFHQ-64 dataset, similar to the results in the paper, analytical teleportation slightly increased FID given fixed sampling steps. For example using DPM-solver++, 40 NFE with no skipping yield FID of 2.46, while skipping to sigma 20.0 gives FID of 2.65 (40 NFE), and FID of 2.77 (25 NFE). Though these changes in FID are tiny, this shows that for face dataset, the neural score model may deviate from the Gaussian approximation in some interesting way at the high noise regime.

**Tab. 3 Analytical teleportation combined with DPM-Solver++ for FFHQ**

| skip_value | 5 | 6 | 8 | 10 | 12 | 15 | 20 | 25 | 30 | 35 | 40 |
| --- | --- | --- | --- | --- | --- | --- | --- | --- | --- | --- | --- |
| **1.0** | 45.41 | 47.31 | 47.69 | 46.56 | 46.11 | 45.99 | 46.00 | 46.03 | 46.04 | 46.05 | 46.06 |
| **2.5** | 18.56 | 16.67 | 15.84 | 14.68 | 13.95 | 13.51 | 13.27 | 13.15 | 13.09 | 13.05 | 13.01 |
| **5.0** | 14.99 | 10.83 | 8.60 | 7.35 | 6.42 | 5.68 | 5.20 | 5.01 | 4.90 | 4.84 | 4.80 |
| **10.0** | 16.28 | 10.24 | 6.67 | 5.40 | 4.52 | 3.78 | 3.31 | 3.13 | 3.05 | 3.00 | 2.97 |
| **20.0** | 17.57 | 10.24 | 5.96 | 4.69 | 3.94 | 3.32 | 2.92 | 2.77 | 2.70 | 2.67 | 2.65 |
| **40.0** | 18.77 | 11.05 | 5.60 | 4.21 | 3.59 | 3.08 | 2.75 | 2.63 | 2.57 | 2.54 | 2.52 |
| **80.0** | 22.51 | 13.77 | 5.96 | 4.05 | 3.44 | 3.02 | 2.70 | 2.57 | 2.51 | 2.48 | 2.46 |

These additional results show that our approximation and acceleration method is still relevant and improved upon the recent more SOTA diffusion sampling method.

For more comprehensive results and FID tables please see https://figshare.com/s/95327f9b23d514566fb7

---

### Decision · Action_Editor_uxqT · 2024-11-04

**Recommendation:** Accept with minor revision

**Comment:**

Overall, the proposed analysis of the learned score model based on the analytical Gaussian score seems to be reasonable and supported by empirical validation on various datasets. Also, most concerns raised by the reviewers are sufficiently addressed in the revision with better description and additional experiments. Therefore, I would recommend the paper to be accepted.

More explanations, better descriptions and corrections, and additional experimental results in the replies need to be included in the final version.

**Audience:**

Comprehensive analysis of current diffusion models in terms of both theoretical and empirical aspects would be interesting as well as important to the current diffusion-based generative modeling. Especially the understanding of the whole process from the perspective of the analytical Gaussian score function and the corresponding acceleration of the reverse process can receive some interest from the community including TMLR's audience.

**Claims And Evidence:**

This paper shows both theoretically and empirically that the learned neural scores of diffusion models can be more accurately approximated by the single Gaussian scores at the initial high noise levels and the Gaussian mixture scores at the final low noise levels, compared to the delta scores of training examples. This finding can explain the generalization ability of the diffusion models. Also, based on this analysis, it accelerates the image sampling 15-30% by skipping the initial phase using the prediction from the exact Gaussian solution.

Overall, the proposed analysis seems to be reasonable and supported by empirical validation on various datasets. Moreover, the authors clarifies why their Gaussian score approximation could be especially effective in the early reverse process and shows more acceleration results combined with recent diffusion samplers in the revision.